# Improving Soil Moisture Prediction of a High–Resolution Land Surface Model by Parameterising Pedotransfer Functions through Assimilation of SMAP Satellite Data

Ewan Pinnington[1], Javier Amezcua[1], Elizabeth Cooper[2], Simon Dadson[2,3], Rich Ellis[2], Jian Peng[3], Emma Robinson[2], Ross Morrison[2], Simon Osborne[4], and Tristan Quaife[1]

[1]National Center for Earth Observation, Department of Meteorology, University of Reading, Reading, UK
[2]UK Centre for Ecology and Hydrology, Wallingford, UK
[3]University of Oxford, Oxford, UK
[4]Met Office Field Site, Cardington Airfield, Shortstown, Bedford, UK

**Correspondence:** Ewan Pinnington (e.pinnington@reading.ac.uk)

**Abstract.** Pedotransfer functions are used to relate gridded databases of soil texture information to the soil hydraulic and thermal parameters of land surface models. The parameters within these pedotransfer functions are uncertain and calibrated through analyses of point soil samples. How these calibrations relate to the soil parameters at the spatial scale of modern land surface models is unclear, because gridded databases of soil texture represent an area average. We present a novel approach for calibrating such pedotransfer functions to improve land surface model soil moisture prediction by using observations from the Soil Moisture Active Passive (SMAP) satellite mission within a data assimilation framework. Unlike traditional calibration procedures data assimilation always takes into account the relative uncertainties given to both model and observed estimates to find a maximum likelihood estimate. After performing the calibration procedure we find improved estimates of soil moisture and heat flux for the JULES land surface model (run at a 1 km resolution) when compared to estimates from a cosmic-ray soil moisture monitoring network (COSMOS–UK) and 3 flux tower sites. The spatial resolution of the COSMOS probes is much more representative of the 1 km model grid than traditional point based soil moisture sensors. For 11 cosmic–ray neutron soil moisture probes located across the modelled domain we find an average 22% reduction in root-mean squared error, a 16% reduction in unbiased root-mean squared error and a 16% increase in correlation after using data assimilation techniques to retrieve new pedotransfer function parameters.

*Copyright statement.* TEXT

## 1 Introduction

Land surface models are important tools for translating meteorological forecasts and reanalyses into real-world impacts by providing schemes for how energy, water and other matter will interact with the Earth's surface, outputting relevant diagnostics and variables and understanding the role of variability in the terrestrial hydrological cycle in the Earth system. As the spatial

resolution of available meteorological information has become increasingly fine (Clark et al., 2016) it is necessary to ensure land surface models can utilise this information at its native resolution in order to provide outputs that are as accurate as possible for local populations. In this paper our focus is on soil moisture which plays an essential role in agriculture (Asfaw et al., 2018), weather and climate prediction (Hauser et al., 2017) and land surface energy partitioning (Beljaars et al., 1996; Bateni and Entekhabi, 2012). The modelling of soil moisture is highly sensitive to driving precipitation and model parameterisations (Pitman et al., 1999). Typically, models of soil moisture will determine parameters based on spatial datasets of soil texture information using pedotransfer functions such as those defined by Cosby et al. (1984) for the Brooks and Corey (1964) soil model. The majority of pedotransfer relationships are calibrated for point samples of soil for a specific geographic location (Cosby et al., 1984; Wösten et al., 1999; Schaap et al., 2004; Tóth et al., 2015). Selecting the appropriate set of pedotransfer functions for the modelled area will allow for more representative results. It is unclear how these calibrations of pedotransfer functions and their resulting soil model parameters relate to the varying spatial scales of modern land surface models, and indeed the use of additional streams of information from remote sensing and in-situ observations is seen as increasingly important for calibration and validation (Van Looy et al., 2017). Pedotransfer functions can be continuous or discrete (setting predefined model parameters for different ranges of soil texture). Discrete examples of pedotransfer functions can be found in Wösten et al. (1999) for the van Genuchten (1980) soil model. Continuous versions of these functions may be preferential as they provide greater heterogeneity for resulting soil model parameter maps which may be more realistic. Tóth et al. (2015) provide more recent examples of continuous pedotransfer functions for the van Genuchten (1980) model. For this paper continuous functions will also allow us to seek updated parameter values that improve the prediction of a land surface model at a given spatial scale and properly account for uncertainty in both the soils information and resulting model predictions.

There now exists a large amount of information from different satellite missions relating to the spatial and temporal variability of soil moisture. These can be based on either active (*e.g.* The Advanced Scatterometer (ASCAT) (Wagner et al., 2013)) or passive (*e.g.* The Soil Moisture Ocean Salinity (SMOS) mission (Kerr et al., 2001)) observing instruments with good results found when combining both (*e.g.* the Soil Moisture Active Passive (SMAP) mission (Entekhabi et al., 2010)). The NASA SMAP mission was originally designed with both an active and passive sensor on board, soon after launch in January 2015 the active sensor malfunctioned. Sentinel 1 is now used as the active component in the SMAP soil moisture retrieval. Recent validation studies have shown SMAP to perform well in comparison with other satellite estimates (Montzka et al., 2017; Chen et al., 2018; Peng et al., 2020). These remotely sensed products are available at scales comparable to current land surface models from 50 km down to 9 km. Traditional in-situ observations of soil moisture are made at a single point using a variety of different methods (Walker et al., 2004). These in-situ measurements provide accurate estimates to the true state of the amount of water contained within the soil. However, the scale of such measurements can be unrepresentative to the scales of the model, even when land surface models are run at a high resolution ($\sim$1 km). The recent developments of cosmic–ray neutron sensing soil moisture probes (Zreda et al., 2008) somewhat alleviates this issue. Cosmic–ray neutron probe observations have a variable spatial footprint dependent on atmospheric air density (130 m - 240 m (Köhli et al., 2015) with some studies quoting a diameter of $\sim$600 m Desilets and Zreda (2013)) that is much more representative of land surface model estimates than that of traditional soil moisture probes. There are now good networks of cosmic–ray probes within several countries (Zreda et al., 2012). This

is true in the UK where the COsmic–ray Soil Moisture Observing System United Kingdom (COSMOS–UK) network (Evans et al., 2016) has been established by The UK Center for Ecology and Hydrology (UKCEH) and returning observations since 2013 (Stanley et al., 2019). These observations can act as valuable validation data of both satellite and land surface model soil moisture estimates (Duygu and Akyürek, 2019).

Data assimilation provides methods for combining new observations with land surface models in order to improve predictions. These techniques can either be used for state-estimation to update soil moisture values of the model in real-time as new observations are available (Liu et al., 2011; Draper et al., 2012; De Lannoy and Reichle, 2016; Kolassa et al., 2017) or for model parameter-estimation to find improved calibrations which better represent the observations (Rasmy et al., 2011; Sawada and Koike, 2014; Yang et al., 2016; Pinnington et al., 2018). Unlike traditional calibration procedures data assimilation and other associated Bayesian optimisation methods always take into account the relative uncertainties given to both model and observed estimates to find a maximum-a-posteriori estimate (Beven and Binley, 1992; Thiemann et al., 2001; Vrugt et al., 2003; Moradkhani et al., 2005; Nearing et al., 2010; Mizukami et al., 2017). Previous studies have used data assimilation to update the soil parameters of land surface models (Rasmy et al., 2011; Sawada and Koike, 2014; Yang et al., 2016; Han et al., 2014). However, we are unaware of any studies using data assimilation to update the parameters of pedotransfer functions to improve land surface model predictions. Updating the parameters of these pedotransfer functions by combining them with observations from satellites addresses a key uncertainty within their calibration with respect to land surface models, adding additional information about spatial heterogeneity and the larger scales of both satellite and land surface model estimates. Many previous studies optimising model soil parameters have taken a filtering DA approach (Moradkhani et al., 2005; Montzka et al., 2011; Han et al., 2014; Baatz et al., 2017; Botto et al., 2018) leading to the recovery of a time-series of parameter values as additional data is assimilated through time. In this study we use a smoother method, i.e. one that uses all observations in the spatial domain within a time window of a given length. Then, the static parameters are obtained by a single minimization process (which can contain iterative steps). Smoothers can be used in a sequence of 'analysis windows' (as it is done in operational numerical weather prediction), but in this study we only use one of these windows since the parameters we search for do not vary in time.

We have used the Land Variational Ensemble Data Assimilation Framework (LAVENDAR) (Pinnington et al., 2020) to combine soil moisture estimates from the NASA SMAP mission with the Joint UK Land Environment Simulator (JULES) model run at a high-resolution (1 km) and update the parameters of the Tóth et al. (2015) pedotransfer functions for the van Genuchten (1980) soil model. In our experiments we assimilated 2016 SMAP data and then ran a hindcast for the year 2017. The experiments were conducted over a sub-domain of the UK due to considerations of computational expense. We selected the region of East Anglia due to it being equally susceptible to flooding and drought and therefore displaying a good dynamic range of soil moisture values. This region also had a good availability of high quality SMAP data (here we use Level-3 SMAP soil moisture observations) and a high distribution of COSMOS probes to allow for thorough validation of any results. While reducing the spatial domain in our experiments eased the computational load we were still modelling over 30,000 grid points due to the high-resolution of the domain.

We defined two objectives for this study. Firstly to examine the ability of 9 km SMAP data to update pedotransfer parameters in a 1 km land surface model. Second to asses the resulting prediction of modelled soil moisture against (a) SMAP data from a

different time period and (b) independent in-situ data from the COSMOS-UK network. We also asses the impact on modelled latent and sensible heat flux at 3 flux tower sites.

## 2 Method

### 2.1 JULES land surface model

The Joint UK Land Environment Simulator (JULES) is a community developed process based land surface model and forms the land surface component in the next generation UK Earth System Model (UKESM). A description of the energy and water fluxes is given in Best et al. (2011), with carbon fluxes and vegetation dynamics described in Clark et al. (2011). We drive the JULES model with the Climate Hydrology and Ecology research Support System meteorology (CHESS) dataset (Robinson et al., 2017) which is a 1 km daily dataset of meteorological variables, an example implementation of JULES with the CHESS-met dataset can be found in Martínez-de la Torre et al. (2019). In our experiments we have used JULES version 5.3, the code and model settings are available through the MetOffice JULES repository (https://code.metoffice.gov.uk/trac/jules), with Rose suite number u-bq357. This model setup is based on the Rose suite u-au394 used to create the CHESS-land dataset (Martinez-de la Torre et al., 2018). The JULES model utilises the Harmonized World Soil Database (HWSD) (Fischer et al., 2008) as the underlying soil texture map for the creation of its soil parameter ancillaries using a pedotransfer function, see Figure 1. The HWSD has been gap-filled in urban areas where no information is available as we ran JULES without urban tiles switched on. The soil scheme is made up of 4 separate layers with depths of 0.1 m, 0.25 m, 0.65 m and 2 m respectively. We have chosen to keep JULES in its default soil-layer setup so that our optimised parameters are relevant to the wider JULES modelling community. This is despite the fact that SMAP satellite observations are typically sensitive to the top $\sim 5$ cm of soil (Entekhabi et al., 2010), with some studies suggsting L-band radiometer measurements may only be sensitive to the top $\sim 2.5$cm (Zheng et al., 2019). This could introduce an additional source of error into our DA system. To ensure that the effect of this is not too great we show that there is only a small difference in soil moisture between depths of 10cm and 5 cm in the JULES model in Figure S1. We have also re-run the entire data assimilation experiment with a 5 cm top soil layer in JULES and show that the recovered parameter distributions are similar to those recovered with a 10 cm top soil layer in Figure S2 and S3. It is necessary to find an appropriate initial state before running a land surface model such as JULES and it has been shown that without a suitable spin-up period forecast skill can be impacted (Maurer and Lettenmaier, 2004). We include a 4 year spin up period at the start of each JULES run to allow the soil moisture state to reach a point of equilibrium after parameter values are changed. For the JULES spin-up the model is run from an initial value (defined by the saturated soil moisture model parameter) over the same year of forcing data, here 2015, to reach an equilibrium soil moisture state for any given set of soil hydraulic parameters. We show this model spin–up for 3 unique soil parameter sets at the same location in Figure S4.

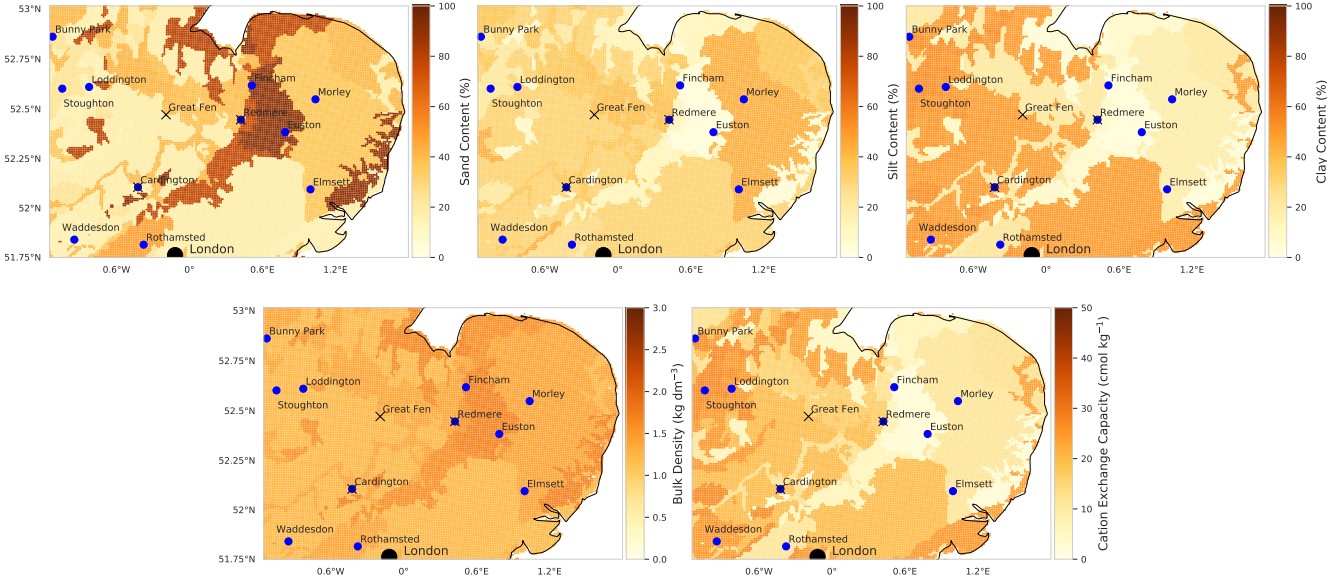

**Figure 1.** Maps of soil properties from the Harmonized World Soil Database (HWSD) (Fischer et al., 2008) used in the creation of the JULES soil parameter ancillaries with the Tóth et al. (2015) pedotransfer functions. Blue dots show locations of COSMOS-UK probes, crosses show flux tower locations and black dot shows location of London, UK.

## 2.2 Pedotransfer functions

120 The JULES model implements both the Brooks and Corey (1964) and the van Genuchten (1980) models of soil physics, with the model of choice being selected by a switch in the JULES namelist files. The JULES implementation of these models can be found in Clark et al. (2011). In this paper we have used the van Genuchten (1980) soil parameterisation scheme and have selected a set of pedotransfer functions from Tóth et al. (2015). The Tóth et al. (2015) pedotransfer functions have been calibrated across a large range of European soils and should be representative of the study area. The mathematical formulation

125 of these pedotransfer functions is:

$$
\begin{aligned}
\theta_{res} &= \begin{cases} 0.041 & f_{sand} \geqslant 2 \\ 0.179 & f_{sand} < 2 \end{cases} \\
\theta_{sat} &= \phi_a - \phi_b \rho^2 + \phi_c f_{clay} + \kappa_a \mathrm{pH}^2 \\
\log_{10}(\alpha) &= -\phi_d - \phi_e \rho^2 - \phi_f f_{clay} - \phi_g f_{silt} + \frac{\kappa_b}{(C_{organic} + 1)} + \kappa_c \mathrm{pH}^2 + \kappa_d \mathrm{topsoil} \\
\log_{10}(N - 1) &= -\phi_h - \phi_i \rho^2 - \phi_j f_{clay} - \phi_k f_{silt} + \frac{\kappa_e}{(C_{organic} + 1)} \\
\log_{10}(K_{sat}) &= \phi_l - \phi_m f_{clay} - \phi_n f_{silt} - \phi_o \mathrm{CEC} + \kappa_f \mathrm{pH}^2 + \kappa_g \mathrm{topsoil},
\end{aligned} \tag{1}
$$

where $\theta_{res}$ is the residual soil moisture (m$^3$ m$^{-3}$), $\theta_{sat}$ the saturated soil moisture (m$^3$ m$^{-3}$), $\alpha$ and $(N-1)$ parameters of the van Genuchten (1980) soil model (-), $K_{sat}$ the saturated hydraulic conductivity (kg m$^{-2}$ s$^{-1}$), $\phi_a, \ldots, \phi_o$ are model parameters to be optimised (values given in table 1) and $\kappa_a, \ldots, \kappa_g$ are static model parameters (values given in table 2). We optimise the parameters controlling the impact of the bulk density $\rho$ (g cm$^{-3}$), fraction of clay and silt ($f_{clay}$, $f_{silt}$) (%) and the cation exchange capacity (CEC) (meq 100g$^{-1}$) as these terms have a first order impact on the outputted van Genuchten (1980) soil parameters. The organic carbon content ($C_{organic}$) (%), soil pH value and topsoil flag have a less pronounced effect on the van Genuchten (1980) soil parameters. We treat the top two soil layers of JULES as topsoil (topsoil $= 1$) and the bottom two as subsoil (topsoil $= 0$). From equations (1) we can see that defining a soil as topsoil will act to increase the saturated hydraulic conductivity and the value of $\alpha$, which will both allow water to flow more freely through the soil. The prior values for the parameters ($\phi_a, \ldots, \phi_o$) are shown in Table 1. We used the values given by Tóth et al. (2015) for the prior except for $\phi_o$ for which we found better results (experiments not shown) when the magnitude of this parameter was increased. To create the JULES soil parameter ancillary files these pedotransfer functions are applied to soil texture information from the HWSD (Fischer et al., 2008) at a 1 km resolution. The DA system used here optimises values for the parameters in table 1 across the whole domain rather than on a grid-by-grid basis. In this way the varied soil properties across the domain give us a form of orthogonal constraint within the assimilation and allow us to recover a single set of pedotransfer functions that are valid in space and time.

| Parameter | Prior value |
|-----------|-------------|
| $\phi_a$ | 0.63052 |
| $\phi_b$ | 0.10262 |
| $\phi_c$ | 0.0003335 |
| $\phi_d$ | 1.16518 |
| $\phi_e$ | 0.16063 |
| $\phi_f$ | 0.008372 |
| $\phi_g$ | 0.01300 |
| $\phi_h$ | 0.25929 |
| $\phi_i$ | 0.10590 |
| $\phi_j$ | 0.009004 |
| $\phi_k$ | 0.001223 |
| $\phi_l$ | 0.40220 |
| $\phi_m$ | 0.02329 |
| $\phi_n$ | 0.01265 |
| $\phi_o$ | 0.10380 |

**Table 1.** Prior values for parameters of the Tóth et al. (2015) pedotransfer functions used in experiments.

| Parameter | Value |
|:---:|:---:|
| $\kappa_a$ | 0.0002904 |
| $\kappa_b$ | 0.40515 |
| $\kappa_c$ | 0.002166 |
| $\kappa_d$ | 0.08233 |
| $\kappa_e$ | 0.2568 |
| $\kappa_f$ | 0.26122 |
| $\kappa_g$ | 0.44565 |

**Table 2.** Static parameter values for the Tóth et al. (2015) pedotransfer functions used in experiments.

## 2.3 SMAP Observations

The NASA Soil Moisture Active Passive (SMAP) satellite mission provides estimates of soil moisture every 2-3 days (Entekhabi et al., 2010). The mission is an orbiting observatory with a passive radiometer and an active radar instrument. SMAP was designed to deliver a 36 km spatial resolution estimate of soil moisture from the passive instrument alongside a 9 km estimate from a retrieval using both the passive and active sensors. After its launch in January 2015 the radar instrument malfunctioned. Subsequently ESA's Sentinel 1 mission has been used as a replacement for the active sensor. For the work in this paper we use the 9 km Level-3 soil moisture product (version 3) this product has a relatively low bias (Colliander et al., 2017; Zhang et al., 2019). However, it has been shown there is a wet bias present in the Level-4 SMAP product (Reichle et al., 2017). As part of the retrieval procedure SMAP relies on some ancillary information, one example of this is soil texture where the Harmonized World Soil Database (HWSD) (Fischer et al., 2008) is used to calculate the soil dielectric constant for use within the retrieval algorithm. The use of such ancillary data in the retrieval could introduce additional biases into the SMAP soil moisture estimates that are not consistent with estimates from the land surface model we are comparing to. However, as the HWSD is also used to create the JULES soil parameter ancillary files this effect should be minimised. We prescribe an error of $0.05 \, \mathrm{m^3 \, m^{-3}}$ for SMAP observations in the assimilation algorithm. Although the SMAP baseline aim for error is $0.04 \, \mathrm{m^3 \, m^{-3}}$ other studies have found slightly higher values for the error in Level-3 SMAP observations ($0.043 \, \mathrm{m^3 \, m^{-3}}$ (Colliander et al., 2017), $0.057 \, \mathrm{m^3 \, m^{-3}}$ (Li et al., 2018) and $0.054 \, \mathrm{m^3 \, m^{-3}}$ (Zhang et al., 2019)), we therefore chose a value between these studies. We have only used SMAP observations corresponding to best retrieval quality flag and surface flag in experiments. The effect that removing poor quality observations has on the total number of observations assimilated can be seen in Figure 2. The experiment area of the East of England is predominantly flat arable land which should allow for good quality SMAP retrievals, there are also coastal and urban areas where SMAP retrievals will be unreliable. This area is also prone to cloud cover which could cause gaps in the SMAP observational record.

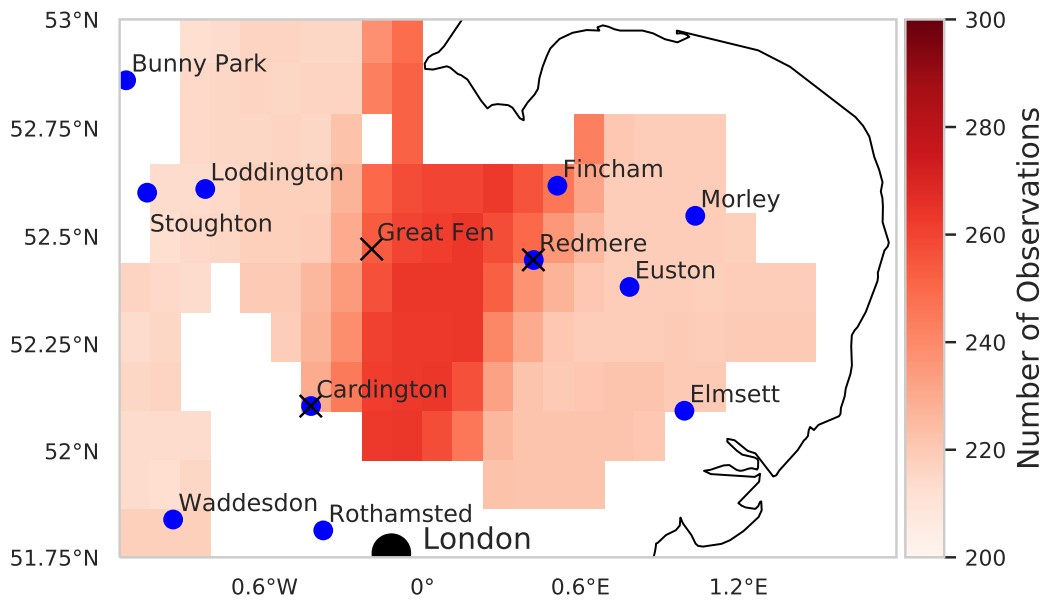

**Figure 2.** Location of COSMOS probes (blue circles) and flux towers (black crosses) used in validation. Red shading indicates number of SMAP observations assimilated in experiment period (2016). No colour corresponds to no observations being assimilated in that location due to low quality retrieval or surface flag. Black dot shows location of London, UK.

## 2.4 COSMOS–UK Observations

The COSMOS–UK network has been producing observations of soil moisture and other meteorological variables at an expanding number of stations (currently 52) since 2013 (Stanley et al., 2019). For the area of interest in this paper we have 11 stations available to us with data for the relevant time period, see Figure 2. Some of these stations may not be representative of JULES model estimates due to the current setup of JULES not considering some processes (ground water, organic soils, urban tiles). Cosmic–ray sensing soil moisture probes have a variable depth as well as horizontal sensitivity (Zreda et al., 2008). There

are many studies translating the cosmic–ray neutron intensity measured at COSMOS probe sites to soil moisture (Baatz et al., 2014; Bogena et al., 2015; Köhli et al., 2015). There have also been efforts to relate modelled soil moisture to cosmic–ray neutron intensity, such as the COsmic-ray Soil Moisture Interaction Code (COSMIC) (Shuttleworth et al., 2013; Rosolem et al., 2014). The COSMOS–UK network use the $N_0$–method described by Baatz et al. (2014) to diagnose values for the soil moisture and then the method of Köhli et al. (2015) to calculate the representative depth for each COSMOS probe measurement. The

COSMOS sites in our experiment domain have a representative depth of between 14 cm and 40 cm dependent on conditions when measurements are made. To make a fair comparison between the COSMOS–UK and JULES soil moisture estimates we have constructed a simple variable depth algorithm for JULES which takes a weighted average of the different soil layers of

the model given the relative depth of the COSMOS–UK observation. This is defined as

$$\theta_D = \begin{cases} \theta_{10}, & \text{if } D \leq 10 \text{ cm} \\ \frac{10}{D}\theta_{10} + \frac{(D-10)}{D}\theta_{25}, & \text{if } 10 \text{ cm} < D \leq 25 \text{ cm} \\ \frac{10}{D}\theta_{10} + \frac{25}{D}\theta_{25} + \frac{(D-35)}{D}\theta_{65}, & \text{if } 35 \text{ cm} < D \leq 65 \text{ cm} \end{cases} \tag{2}$$

where $\theta_D$ is the JULES modelled soil moisture at the COSMOS-UK representative depth ($D$) and $\theta_{10}, \theta_{25}$ and $\theta_{65}$ are the top, second and third layer soil moisture estimates from the JULES model.

## 2.5  Flux Tower Observations

In order to understand how updating the JULES soil parameters of the model might effect the model prediction of latent and sensible heat flux we compare prior and posterior estimates to observations at 3 flux towers. The location of these flux towers

are shown as black crosses on Figure 2, two of these flux towers are located near to COSMOS-UK sites (Cardington and Redmere) and so the black cross is displayed over the blue dot signifying the COSMOS-UK location.

The Met Office site at Cardington (29 m above sea level) is a 18-ha area laid mainly to manicured grass set within generally flat semi-rural surroundings (Osborne and Weedon, 2021). The site has been making continuous subsoil, surface and near-surface measurements since 1996. The turbulence fluxes we use here were calculated over 30-min intervals based on the

eddy-covariance technique using tower data at 10 m height. For latent heat fluxes, this was using the Licor LI-7500 high frequency open-path gas analyser for water vapour and the vertical wind component from a Gill HS-50 3-D sonic anemometer. The same anemometer was used to monitor the rapid response in both the virtual temperature and vertical wind required for the sensible heat flux.

The Redmere and Great Fen sites are located on lowland peat soils in the East Anglian Fens. Both sites are nodes of the UK

Land Flux Network (UKLFN) operated by the UK Centre for Ecology and Hydrology (UKCEH). The Redmere site is cropland, producing Maize and lettuce in 2016 and 2017, respectively. The Great Fen site is an area of extensively managed grassland. Instrumentation is identical at both locations, consisting of a Windmaster ultrasonic anemometer (Gill Instruments Ltd.) and a LI-7500A infrared gas analyser (LI-COR Biosciences, Ltd). Raw (20 Hz) EC data were reduced to thirty minute flux densities using the EddyPRO v7.0.6 Flux Calculation Software (Fratini and Mauder, 2014). Data quality control included outlier removal

and filtering using site specific friction velocity ($u^*$) thresholds (Papale et al., 2006; Reichstein et al., 2005). Gaps in the EC data were filled using the Marginal Distribution Sampling approach (Reichstein et al., 2005, 2014). The Redmere and Great Fen dataset and full details of the sites and flux methodology are available in Morrison et al. (2020).

## 2.6  Data Assimilation Framework

In order to estimate the identified pedotransfer function parameters we use the LAVENDAR data assimilation framework

(Pinnington et al., 2020). This framework utilises a hybrid DA technique similar to that of the Iterative Ensemble Kalman Smoother (IEnKS) (Bocquet and Sakov, 2013). A smoother is different than a filter (e.g. the Ensemble Kalman Filter (Evensen, 2003)) in that it uses batches of observations which are taken over a time window of given length and the whole spatial

domain, as opposed to just in a time instant. These observations are combined with the model evolution over this window and a minimization process is performed to obtain initial conditions for the state/parameter values. It is possible to run a sequence of smoother steps for successive windows, but our study only uses one year long assimilation window as the parameters we are optimising do not vary in time.

Using a smoother instead of a filter has advantages (Lorenc and Rawlins, 2005) in that (a) more observations can be used to constrain the problem solution, and (b) information from the model evolution is implicitly used in the search process. However, using a smoother requires computing the Jacobian of the model, the so–called tangent linear model (TLM) and the related adjoint model (AM). The TLM/AM (Courtier et al., 1994). Computing and maintaining the TLM/AM is not a trivial task, and in fact we do not have this for JULES. The IEnKS solves this problem by replacing the role of the TLM/AM by 4–dimensional covariances, i.e. covariances defined over time and space. These covariances are computed as sample estimators of a given ensemble. The iterative nature of the method means that it finds the solution to the minimization problems using inner iterations rather than a single step (hence the variational nature), and this helps when the distributions of the variables/parameters of interest are not Gaussian. We provide details of the method in Appendix A. Furthermore, to understand the variants of the ensemble Kalman Smoother and its position within the hybrid DA methods, the reader is referred to Evensen (2018).

We show a schematic of how this system works in Figure 3, this involves running an ensemble of JULES models, with each model in the ensemble utilising a distinct soil ancillary data–set. Each ensemble members ancillary file is created by sampling from the normal distribution defined by mean $\mathbf{x}_b$ and variance $(0.1 \times \mathbf{x}_b)^2$, where $\mathbf{x}_b = (\phi_a, \phi_b, \ldots, \phi_o)$ with $\phi_a, \ldots, \phi_o$ taking the values given in table 1, then using each unique set of sampled parameters within equations (1) applied to the HWSD maps of soil properties (see Figure 1) for the whole domain. Although van Genuchten and hydraulic conductivity parameters can be described by logarithmic distributions it is less clear what distribution is best for the PTF parameters optimized here. We therefore made the naive assumption of a normal distribution in the first instance as this gave us good results. In this type of experiment the number of ensemble members will control the quality of the results, with a larger ensemble more likely to identify the optimum parameters. However, running a land surface model at a 1 km spatial resolution over the specified domain is computationally expensive, we therefore use an ensemble size of 50 in our experiments. In order to compare the 1 km estimates of soil moisture from JULES to the 9 km SMAP estimates we create an observation operator which aggregates the JULES grid cells within each SMAP pixel by taking a spatial average of all JULES estimates which fall in the bounds of the SMAP grid cell. There is no need to project increments from the spatially averaged 9 km model estimates back to the 1 km model grid as the assimilation is only optimising the 15 PTF parameters $(\phi_a, \phi_b, \ldots, \phi_o)$ for the whole domain and the update to soil moisture will be implicit. The aggregated spatial observation operator will introduce an additional source of representativity error alongside the observational error of SMAP and the inherent model error within JULES. It has been shown that, for variational methods such as the one used in this paper, these additional sources of error (model error, representativity error, etc.) can be included in the observational term of the cost function by inflating the diagonal observation error variance (Howes et al., 2017). Although observation error inflation is rare in relation to filtering DA methods it is commonly used with variational methods and smoothers, especially in numerical weather prediction (Hilton et al., 2009; Bormann et al., 2015; Minamide and Zhang, 2017; Fowler et al., 2018; Wang et al., 2019). Observation error inflation is required due to the fact

that all observations are used at once in the assimilation whereby we minimise a cost function containing a prior term and an observational term. The greater the number of observations in the observational cost function term, the higher the weight they have in the optimization. This can lead to the prior term being completely negated and hence the retrieval of nonphysical parameters. Observation error inflation would not be required if the correct specification for the observation error correlations (in space and time), model error and representativity error were included. These, however, are hard to diagnose and it has been shown that in the absence of such information observation error inflation is required for an optimal DA system (Stewart et al., 2014). For this reason and due to the large number of observations assimilated in our one year assimilation window (28698) we inflate the specified observational error by a factor of four. If a filtering DA system were being used utilising a bias aware method such as that presented by Ridler et al. (2017) could help represent some of the additional sources of error discussed here.

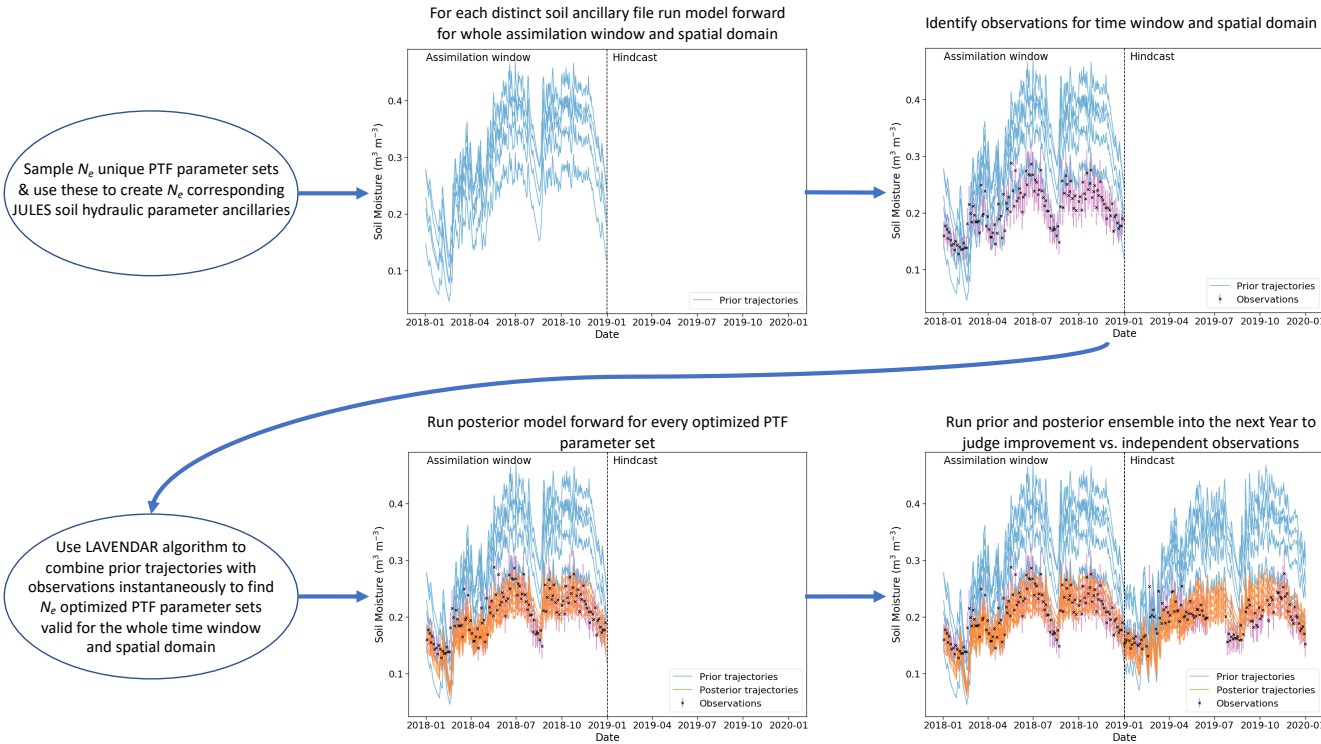

**Figure 3.** Schematic of the LAVENDAR data assimilation framework, showing the workflow for the experiment. Here $N_e$ is the chosen size of ensemble, in the schematic we show $N_e = 5$, but in practice for our experiments $N_e = 50$.

## 2.7 Experiment Formulation

We conducted our pedotransfer function parameter estimation for the year of 2016 using all SMAP observations in this period. We also ran the prior and posterior JULES ensembles into 2017 so that we could judge the results against independent SMAP

observations in a 'hindcast' experiment. Allowing us to judge if any skill added by the assimilation persisted into the future. For the 2016-2017 period we then used the available COSMOS probe observations for validation comparing both prior and posterior JULES soil moisture estimates to these observations. Using the COSMOS–UK observations in this way gives us a better understanding of whether information added by the assimilation of SMAP observations can help to improve model
estimates at in–situ scales.

## 3 Results

### 3.1 Assimilation Output

The input to the data assimilation routine is an ensemble of 50 unique Tóth et al. (2015) PTF parameter sets drawn from a prior distribution (representing our best a priori guess to the true PTF parameters), the corresponding JULES runs (2016-2017)
for each PTF parameter set and all the SMAP observations for the year 2016 over the experiment domain. The output of the data assimilation is an ensemble of 50 optimised (posterior) PTF parameter sets, valid for the whole experiment domain and time, this allows us to calculate the posterior JULES soil ancillary files for each optimized parameter set and the corresponding posterior JULES model runs for 2016-2017. Figure 4 shows the prior and posterior parameter distributions for the 15 optimized parameters of the Tóth et al. (2015) pedotransfer functions. Prior distributions for the 50 JULES ensemble members are shown
in light grey with posterior distributions shown as dark grey. We can see that while the DA procedure made large updates to some parameters compared to their prior values others have not changed, with their mean appearing to be in a very similar place. One of the parameters with a strong change is $\phi_a$ which is decreased compared to the prior, this parameter controls the absolute magnitude of the saturated soil moisture ($\theta_{sat}$). Decreasing it will reduce the absolute saturated soil moisture and allow the soil texture information to have more impact on the diagnosed van Genuchten (1980) model parameter. This can be
seen in Figure 5 where we show the updated PTF parameters effect on the mean estimate to the JULES model soil parameters when applied to the spatial maps of soil properties from the HWSD. We can see how different areas of distinct soil texture (see Figure 1) behave differently based on the PTF parameter updates after DA. For some parameters we see the majority of grid cell parameter values increase or decrease, $\theta_{sat}$ and $\frac{1}{N-1}$ respectively. Whereas for $\frac{1}{\alpha}$ and $\theta_{crit}$ we see an increase or decrease in grid cell parameter values dependent on the underlying soil properties (sandier soils lead to an increase, less sand more clay
correspond to a decrease).

In Figure 6 we show the difference between mean water budget variable estimates (soil moisture, evapotranspiration and run off) in 2016 for the prior and posterior JULES model ensemble. The grid cells that are darker blue correspond to the posterior ensemble estimate being higher after assimilation and grid cells that are darker red correspond to the posterior estimate being lower. We can see that after calibration of the pedotransfer function parameters the domain has not had a uniform increment
to the value of mean soil moisture, evapotranspiration or run off. This is due to the fact that soil texture specific parameters have been optimised allowing the different distinct areas of soil type defined by the HWSD (see Figure 1) to behave differently rather than having a uniform correction across the modelled area. Across the whole domain we find an average increase of 0.03 m$^3$ m$^{-3}$ in mean soil moisture estimates after data assimilation. We can see that in order to update PTF parameter values

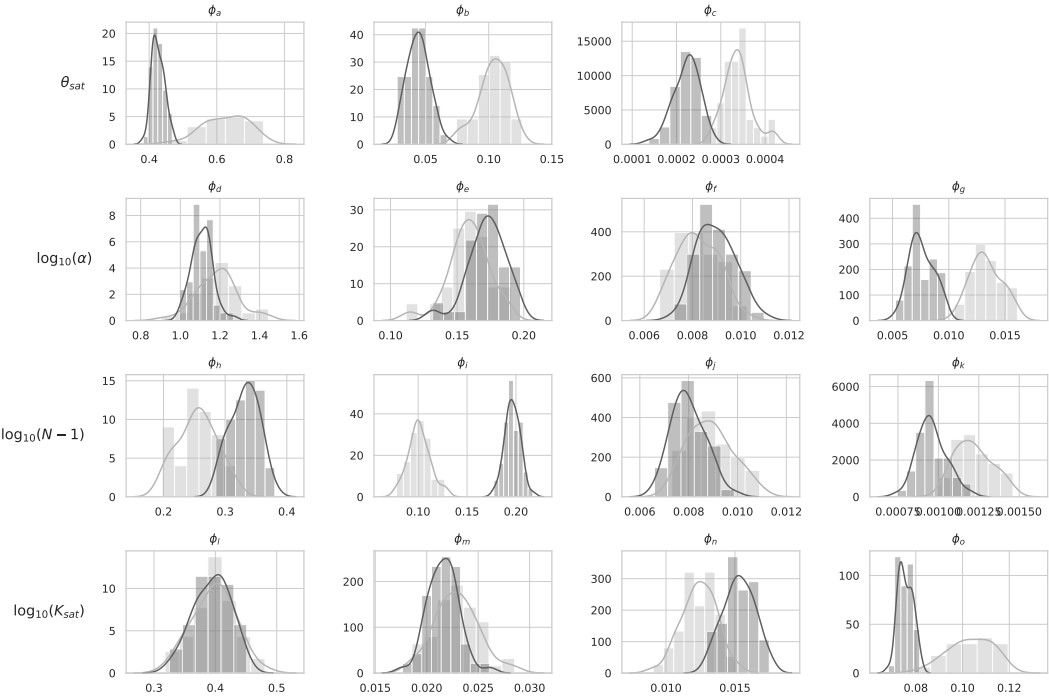

**Figure 4.** Distributions of prior and posterior pedotransfer function parameters grouped by the term in the equations (1) that they relate to (see row labels). Light grey: parameter distribution for the prior ensemble, dark grey: parameter distribution for the posterior ensemble.

to find soil moisture estimates that more closely match the SMAP observations both evapotranspiration and run off model
estimates have also been modified. In areas of sandy soils wetter soil moisture values have been achieved by a decrease in
evapotranspiration offsetting a slight increase in runoff. In areas of high clay content wetter soil moisture values have been
achieved by a larger decrease in run off compared to an increase in evapotranspiration. For silty soils we find a drier value of
soil moisture for the posterior compared to the prior with a less prominent impact on evapotranspiration and run off. Figure 6
also allows us to see the high–resolution of the JULES model when run with the CHESS data, for this domain we have over
30,000 individual model grid cells.

    Figure 7 shows the error reduction after performing data assimilation when comparing JULES spatially aggregated estimates
to SMAP observations. This is computed as $100 \times \frac{(RMSE_{prior} - RMSE_{post})}{RMSE_{prior}}$, where $RMSE_{prior}$ is the JULES prior ensemble
mean RMSE when compared to 2016 SMAP observations and $RMSE_{post}$ is the JULES posterior ensemble mean RMSE when
compared to 2016 SMAP observations. As we are minimising a cost function to find optimised values of PTF parameters valid
for the whole spatial and temporal domain it is possible the optimisation may have to degrade the fit of the model estimates
to the SMAP observations at certain locations in order to improve the picture as a whole. This could be due to errors at these
locations in driving data, the underlying soil property map (*e.g.* presence of organic soils) or indeed in the model structure.
For the majority of the domain we find a reduction in error after assimilation, with a mean error reduction of 20% in 2016 and

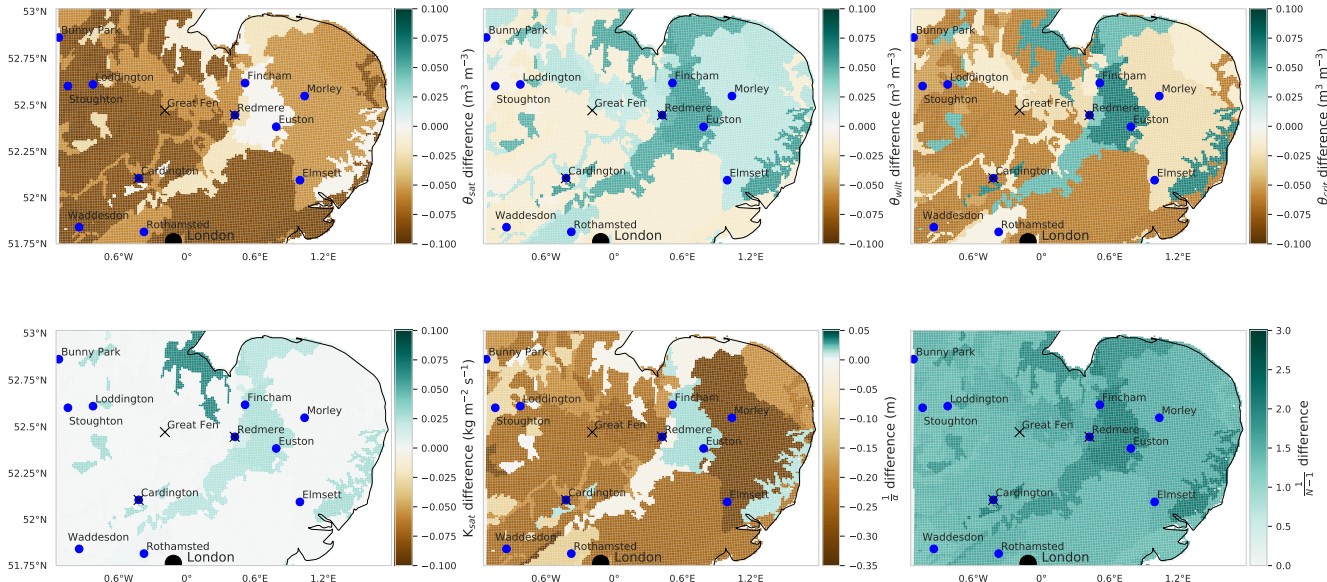

**Figure 5.** Maps showing the difference between the prior and posterior mean JULES model soil parameters, created by applying the prior/posterior PTF's to the HWSD maps of soil properties. Brown corresponds to a decrease in the soil parameter after data assimilation, green to an increase.

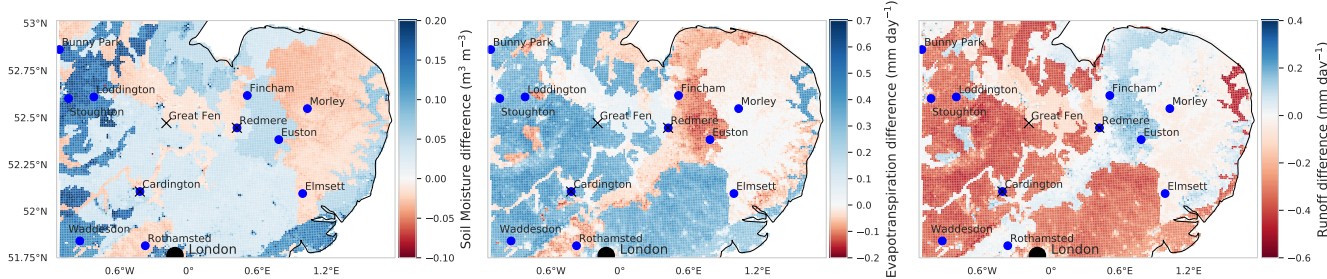

**Figure 6.** Map showing the difference between yearly mean soil moisture for the prior and posterior ensemble of JULES model runs in 2016. Blue corresponds to the posterior ensemble estimate being wetter, red corresponds to the posterior being drier.

21% in 2017. The exception to this being the area corresponding to the city of London. There are two reasons for this, firstly we have not assimilated SMAP soil moisture estimates over this area due to the surface flag corresponding to poor quality observations (poor quality SMAP grid cells are shown in Figure 7 with stippling). Secondly the setup of JULES we have used in our experiments does not have the urban tile turned on, instead we have had to gap-fill the HWSD over London with the surrounding grid cells soil type. This means that soil moisture estimates for this location will not be realistic. To visualise what the time-series of results looks like we plot SMAP observations and JULES model predictions for different pixels in Figure 8 & 9. From these figures we can see the distinct seasonal dynamics of soil moisture in this region, with the highest

moisture being in the winter months and a distinct dry down from April into the summer months. This seasonal cycle is seen for both the JULES model and SMAP observed estimates. For Figure 8 we can clearly see the improvement in the posterior JULES ensemble estimate when compared to the prior. This improvement continues into the 2017 hindcast period when judged against observations that have not been used in the cost function of the data assimilation framework. We can see that although the dynamics in 2017 are distinct from those used for calibration in 2016 we still match the SMAP estimates for dry-down and re-wetting of the soil in this period. From Figure 8 we can also see the spread in our model estimates with the JULES ensemble standard deviation displayed as shading. This spread is decreased from the prior to posterior estimates. In Figure 9 we plot the results for a SMAP pixel over London where the posterior error increases compared to the prior. However, we can see that the SMAP observations do not appear reliable here with many observations hitting the lower bound of soil moisture in the SMAP retrieval. In Figure 10 we show the RMSE averaged in space for the JULES model prior and posterior mean estimate, when compared to SMAP, alongside the JULES model prior and posterior ensemble spread. At all times the posterior JULES RMSE is lower than that of the prior, showing that the DA system has found a set of PTF parameters that improve the fit to the SMAP observations through time, this continues into the hindcast period (2017) when judged against observations that were not included in the DA cost function. We find slight peaks in the RMSE values throughout the time period corresponding to wetter conditions, this could be due to slight errors in the precipitation driving data used to force the model. It is optimal to have an ensemble spread that matches the magnitude of the ensemble mean RMSE and this relationship should hold given a large enough ensemble size (Houtekamer and Mitchell, 1998). We can see that this relationship holds for our prior estimates. However, after DA the posterior ensemble spread is slightly lower than that of the ensemble mean RMSE. This is perhaps unsurprising as we are conducting just a single assimilation step using all observations (over 28000) at once in space and time with a relatively small ensemble size (50). This can lead to some of the posterior parameter distributions becoming narrow, as with increasing observations we increase the confidence in our posterior, thus tightening the retrieved distributions and reducing the model ensemble spread. This result suggests that ensemble inflation (Anderson and Anderson, 1999) may be necessary if this ensemble was to be used in subsequent assimilation experiments.

## 3.2 Comparison to COSMOS–UK

After performing the data assimilation procedure we use the observation operator described in section 2.4 to compare the prior and posterior JULES 4–layer soil moisture estimates to the 11 COSMOS probes located in our experiment domain. For each COSMOS site we select the nearest JULES grid–cell to the given site longitude and latitude. In Figure 11 we show results at the Cardington COSMOS site, here we can see the posterior JULES estimate is a large improvement from the prior, although some of the driest values are still not captured. From Figure 11 we can also see there is an increase in evapotranspiration and a decrease in runoff, this effect can also be seen from Figure 6. Figure 12 shows results for Morley COSMOS site where both prior and posterior JULES estimates perform similarly, we also have less of an update to evapotranspiration but a decrease in modelled runoff. There are also some sites where even after calibration we still do not capture the COSMOS estimates, Stoughton in Figure 13 is such an example where both prior and posterior estimates are too dry. However, here the posterior estimate is still much improved from the prior. We also find large increases in evapotranspiration and reductions in runoff

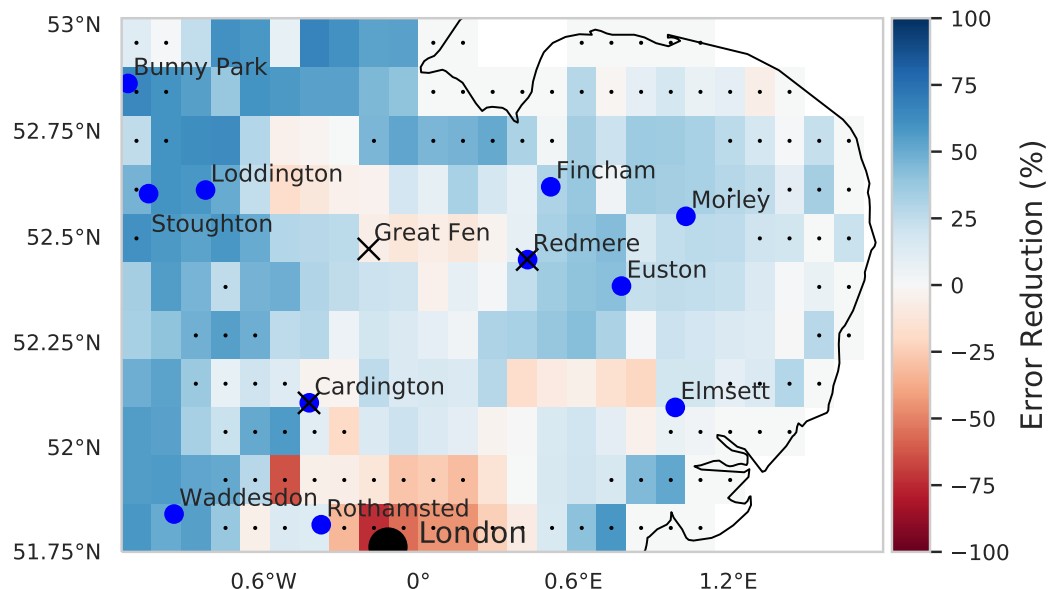

**Figure 7.** Map showing the difference between Root-Mean Squared Error (RMSE) when JULES spatially aggregated estimates are compared to SMAP observations for the prior and posterior ensemble. Blue corresponds to reductions in RMSE for the posterior ensemble, red to an increase. Grid-cells displaying stippling signify low quality SMAP pixels which have not been used in the assimilation procedure. Over the whole domain we find an average reduction in RMSE of 20% after data assimilation for 2016 and 21% for 2017.

for Stoughton. Figure 14 is an example where both prior and posterior perform equally poorly. The fact that the estimates and updates after DA are so different for Figures 11 - 14 despite all using the same PTF parameters highlights the effect that the underlying soil properties are having on soil hydraulic conductivity. At all sites the JULES model predicts top layer soil temperature well when both prior and posterior estimates are compared to in-situ observations. In table 3 we show summary statistics for soil moisture at the 11 COSMOS sites, we see that when looking over all sites the posterior estimate yields a 16%

increase in correlation, 16% reduction in unbiased Root-Mean-Squared Error (ubRMSE) and a 22% reduction in Root-Mean-Squared Error (RMSE) when compared to the prior.

The COSMOS-UK observations we have used for independent validation of the results are respresentative of depths from 14 cm up to around 40 cm. The SMAP satellite observations, used within the assimilation algorithm to find a new set of pedotransfer functions for the experiment domain, are representative of soil moisture for the top 2.5 - 5 cm of soil. Therefore

the fact that after assimilation we find such a distinct improvement at in-situ COSMOS probe locations indicates that although the SMAP observations are only sensitive to shallow depths, by combining these with the JULES model we are also improving estimates at deeper levels. The large errors in our prior JULES estimates for the COSMOS sites in Figure 13 and 14 could point towards some systematic bias within the model. However, it is important to note that the COSMOS-UK observations are independent of the data assimilation. For the assimilated SMAP observations it may be optimal to have errors centred around

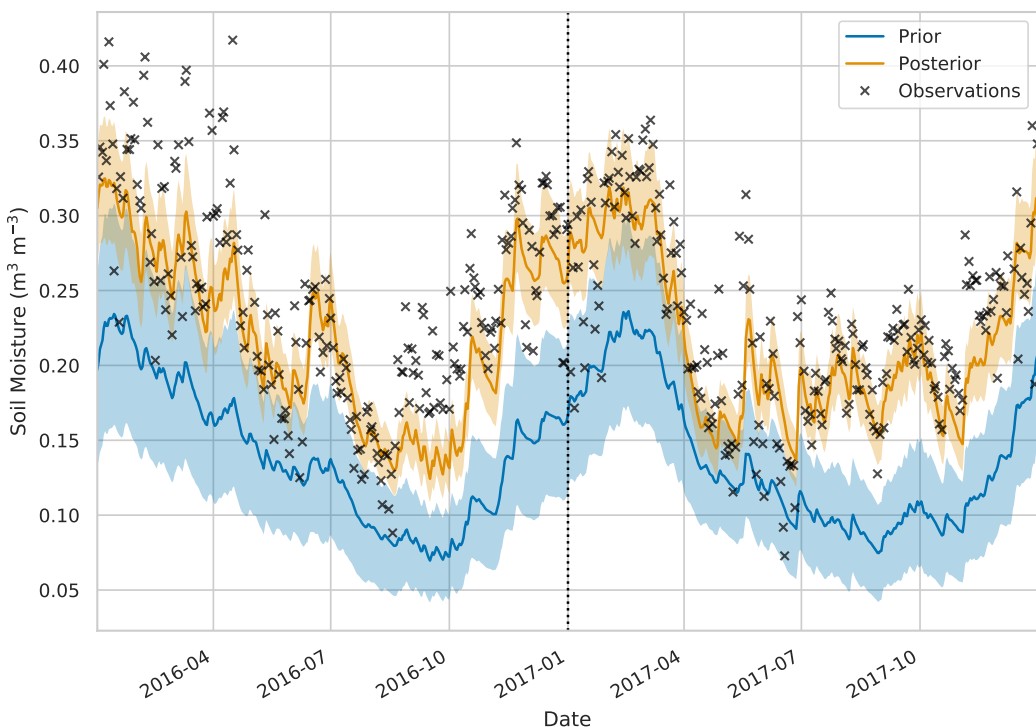

**Figure 8.** Time–series of soil moisture for 52.96° N 0.40° W. Black crosses: SMAP observations, blue line and shading: prior JULES mean and ensemble spread, orange line and shading: posterior ensemble mean and spread. Black dotted line represents the end of the assimilation window and start of the hindcast period.

zero but for the independent in-situ validation data there will be many competing errors that may make this impossible. There will be errors in the forcing meteorology (here we are using CHESS 1km forcing data and not observed in-situ meteorology), errors in the model grid and its representativity to the in-situ location, structural model errors (we currently have no ground water model in JULES and some in-situ sites may be more ground water dominated), errors in the vegetation fractions, and many more. At the larger SMAP scale many of these effects will be minimised when looking at the 9 km spatial scale that is

more representative of modelled estimates.

### 3.3   Comparison to Flux Tower observations

In this section we compare our results to heat flux observations made at 3 flux tower sites during the experiment period. Although updating the soil parameters and soil moisture in our experiments will have an impact on the modelled heat fluxes there are multiple model components that will effect the heat flux estimates (vegetation schemes, roughness length parameterisa-

tions, *etc.*). So that improving modelled soil moisture does not necessarily lead to improved modelled heat fluxes. However, if these other model components perform adequately we should see some improvement in heat flux estimates from improved soil

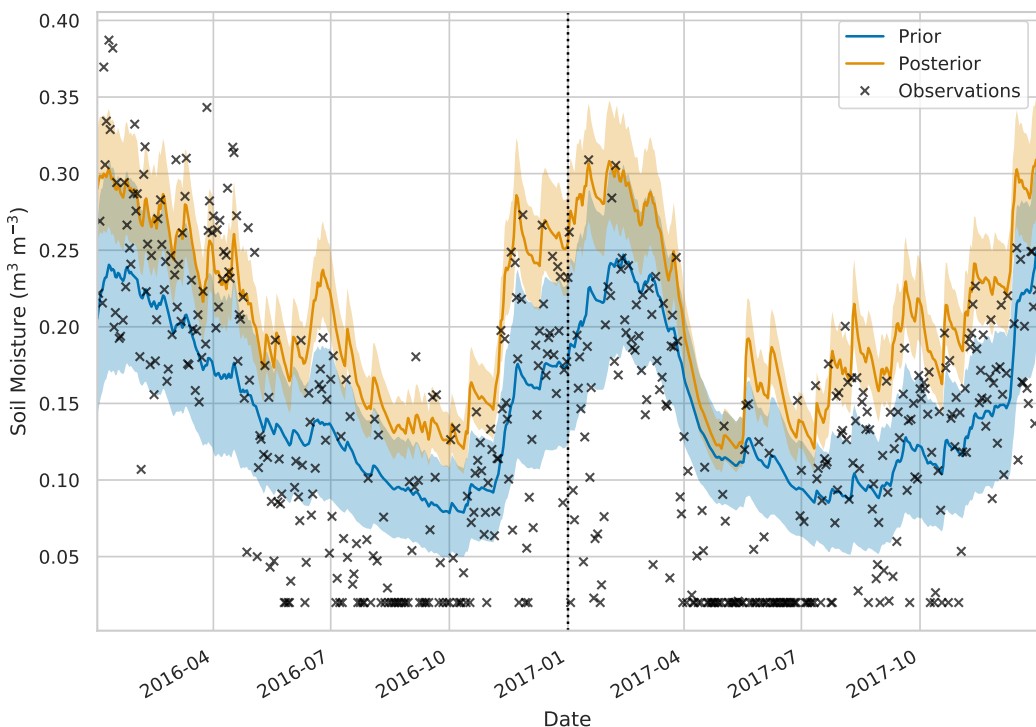

**Figure 9.** Time–series of soil moisture for 51.81° N 0.17° W. Black crosses: SMAP observations, blue line and shading: prior JULES mean and ensemble spread, orange line and shading: posterior ensemble mean and spread. Black dotted line represents the end of the assimilation window and start of the hindcast period.

moisture predictions. In Figure 15 to 17 we show the prior and posterior JULES estimates compared to flux tower observations at each site alongside the prior and posterior soil moisture for the model grid-cell nearest the flux tower. For Figure 15 we can see that at Cardington for latent heat the posterior JULES estimates move toward the flux tower observations, this is also the case to a lesser degree for sensible heat flux, with these changes corresponding to a large update to the soil moisture trajectory after assimilation. For the Great Fen flux tower site in Figure 16 we can see we have less available observations, at this site we have a smaller update to the soil moisture trajectory after data assimilation with the prior and posterior both matching the SMAP observations well. Even with this slight update to soil moisture at the Great Fen site we see a moderate improvement in latent and sensible heat flux compared to the observations. We have a similar situation for Redmere in Figure 17, where a small increment to the soil moisture trajectory corresponds to moderate improvements in the model estimated heat fluxes. In Table 4 and 5 we show summary statistics for the model performance of latent and sensible heat at the 3 flux tower sites. From these tables we find the largest improvement in modelled heat fluxes after data assimilation at the Cardington flux tower site. This also corresponds to the site with the largest improvement in modelled soil moisture. However, even at the Great Fen and Redmere sites where we see less of an impact on modelled soil moisture after data assimilation we still see some

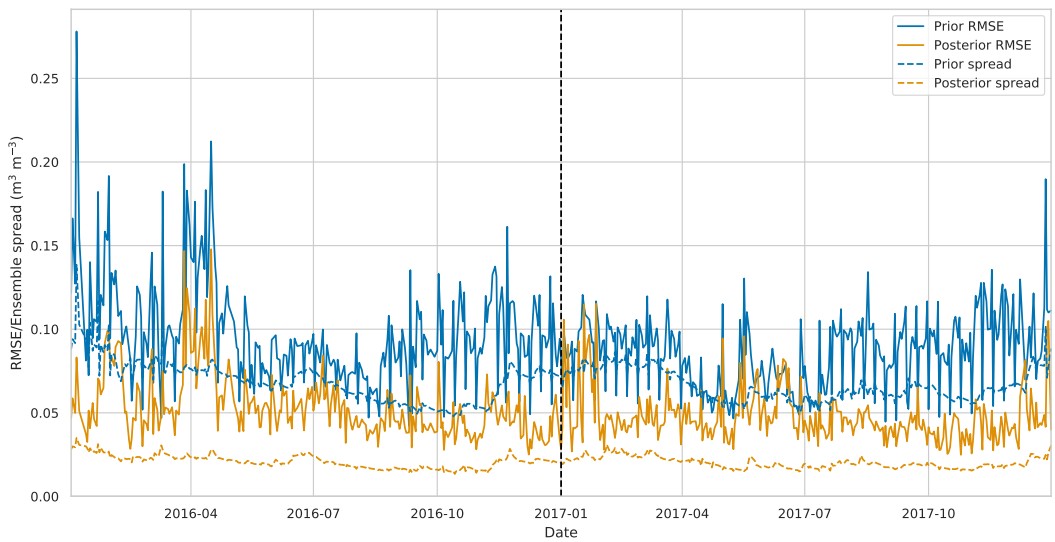

**Figure 10.** Spatially averaged RMSE and ensemble spread for JULES prior and posterior model estimate. Blue solid line: prior JULES RMSE, blue dashed line: prior JULES ensemble spread, orange solid line: posterior JULES rmse, orange dashed line: posterior JULES ensemble spread. Black dotted line represents the end of the assimilation window and start of the hindcast period.

improvement in the modelled heat fluxes. In all cases its seems that JULES slightly under predicts latent heat and slightly over predicts sensible heat compared to the observations. As previously noted this under/over prediction is likely dues to other model components, such as vegetation, where the models representation may be different to the truth. This is especially true for the Redmere flux site that is positioned in a cropland with a rotation of maize and lettuce, both of which are not represented in the current configuration of JULES.

## 4 Discussion

This study aimed to determine the suitability of satellite observations to optimise pedotransfer functions and improve soil moisture estimates for a land surface model. Currently pedotranfer functions are calibrated through analyses of point soil samples and it is unclear how these calibrations and their resultant soil model parameters relate to the varying spatial resolutions of modern land surface models. Adding additional information from satellite estimates into the callibration of pedotransfer functions should address a key uncertainty with respect to the larger scales of land surface model estimates.

We used the LAVENDAR hybrid data assimilation framework (Pinnington et al., 2020) to optimise the parameters of the Tóth et al. (2015) pedotransfer functions by combining them with SMAP Level-3 9 km satellite observations and the JULES land surface model run at a 1 km resolution. This framework outputs a single set of PTF parameters valid in space and time by utilising all data at once through the minimisation of a cost function. The optimized pedotransfer functions found after DA were shown to improve model estimates of soil moisture when compared to SMAP data from a different time period (21% reduction

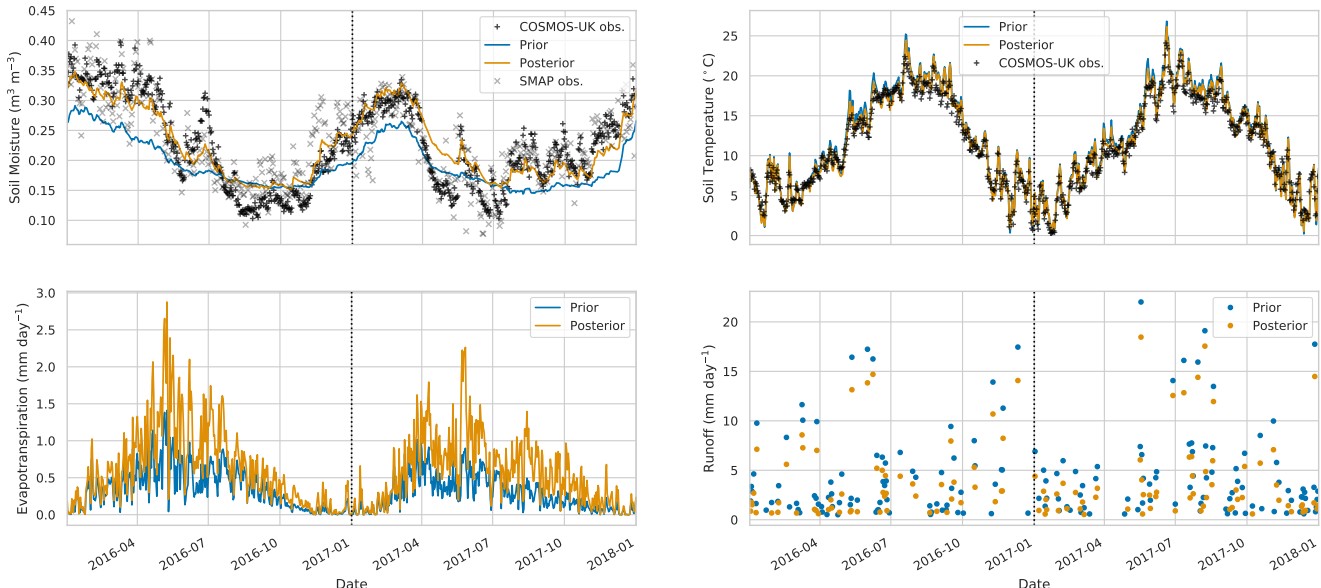

**Figure 11.** Times–series of water budget variables and soil temperature at Cardington COSMOS site. Black plus signs: COSMOS-UK observations, grey crosses: SMAP observations for closest 9 km pixel, blue line: prior JULES estimate for closest 1 km grid cell, orange line: posterior JULES estimate for closest 1 km grid cell.

in RMSE) and independent in-situ observations from the COSMOS-UK network (16% increase in correlation, 16% reduction in ubRMSE and 22% reduction in RMSE over 11 sites), while also seeing some improvement in modelled sensible and latent heat flux at 3 independent flux tower sites. This demonstrates that satellite observations can be used to update pedotransfer functions and improve estimates of soil moisture for land surface models. Previous studies have shown that satellite observations can be

used to improve model estimates of soil moisture by directly updating soil model parameters on a grid by grid basis. Han et al. (2014) used observations from the Soil Moisture Ocean Salinity (SMOS) mission (Kerr et al., 2001) to update parameters of the Community Land Model (CLM) in a Local Ensemble Transform Kalman Filter (LETKF) and improved model estimates. Yang et al. (2016) used a variational method to combine observations from the Advanced Microwave Scanning Radiometer for Earth Observing System (AMSR-E) (Kawanishi et al., 2003) with a land surface model to improve estimates over the Tibetan and

Mongolian Plateau. (Nearing et al., 2010) used calibration techniques to update NOAH land surface model parameters using synthetic aperture radar imagery at a site in Arizona, USA. Our results show similar improvements are achieved by updating PTF parameters with SMAP satellite data. We also demonstrate that information from such satellite observations which are representative of a larger spatial area (9 km) and shallow soil depth (2.5 - 5 cm) allow us to improve 1 km model estimates at independent COSMOS probe sites. The COSMOS probes are representative of a much smaller spatial scale (∼300 m) and

a deeper soil layer (14 - 40 cm), meaning that by combining SMAP observations with the JULES model we are able to find PTF's that better represent finer spatial scales and deeper soil moisture's.

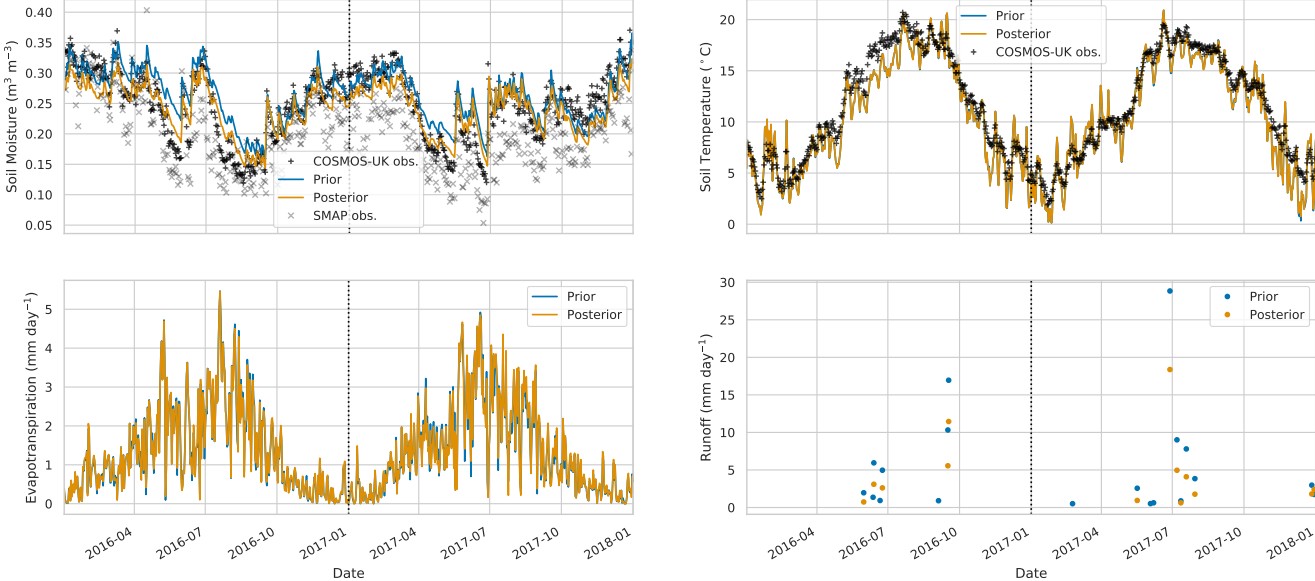

**Figure 12.** Times–series of water budget variables and soil temperature at Morley COSMOS site. Black plus signs: COSMOS-UK observations, grey crosses: SMAP observations for closest 9 km pixel, blue line: prior JULES estimate for closest 1 km grid cell, orange line: posterior JULES estimate for closest 1 km grid cell.

The correlated nature of the PTF parameters in equation (1) presents a potential source of equifinality (e.g. both $\phi_a$ and $\phi_c$ both act to increase the magnitude of $\theta_{sat}$ in the presence of clay soils), this means that we could achieve the same soil hydraulic conductivity with multiple realisations of PTF parameters at any individual grid cell. The effect of this is greatly
reduced as we are performing the optimization over the whole domain and not on a grid cell by grid cell basis. In effect this means the unique soil properties at each of the 30614 model grid cells act as orthogonal constraints within the DA algorithm and reduce the issue of equifinailty for the optimized PTF parameters as the DA algorithm is having to fit the assimilated soil moisture observations for many different soil textures at once. It may also be possible to improve results further by including information on such correlations within our prior. Such estimates have been included in a variational DA framework for the
carbon cycle and shown to improve posterior estimates (Pinnington et al., 2016). Previous studies have noted the issue of equifinality when optimising soil model parameters on a grid by grid basis (Beven, 2001). Samaniego et al. (2010) proposed the multiscale parameter regionalization method to alleviate this issue by performing a spatial uniforming function and linking parameters at coarser scales to those at finer resolutions. Our technique also allows for a vastly reduced parameter space by moving from updating gridded soil model parameters to instead optimizing a single set of pedotransfer function parameters
valid in space and time. This could also lead to issues as we are not considering uncertainty in the underlying soil property database (Fischer et al., 2008), which could contain errors (Tifafi et al., 2018). It may be appropriate when performing such a

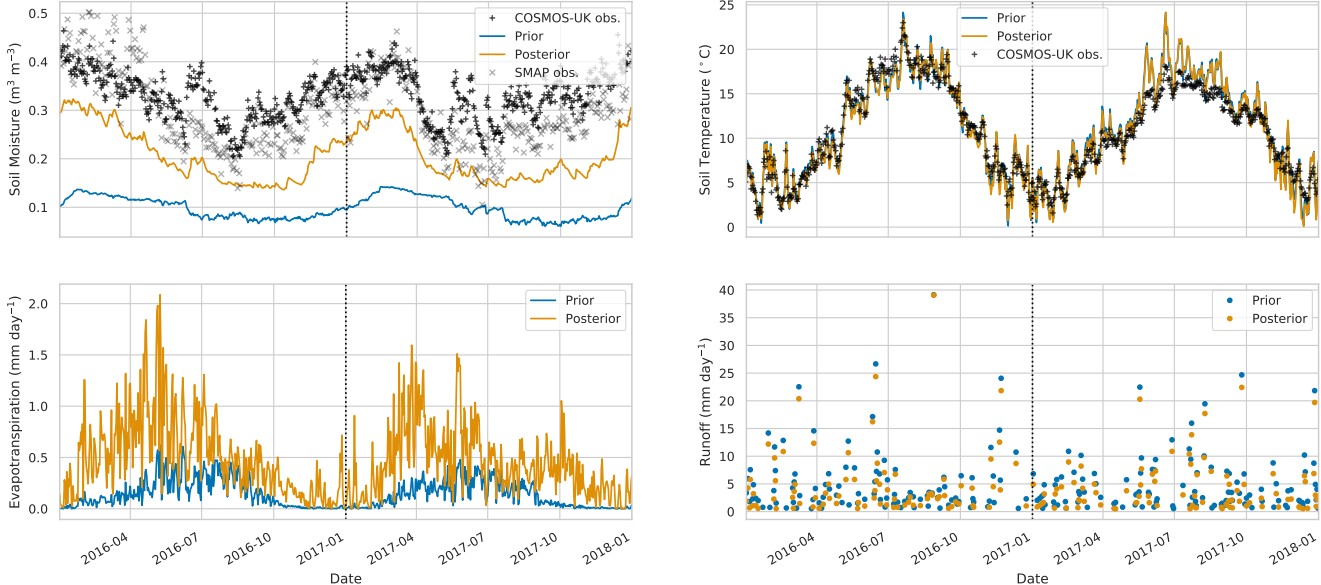

**Figure 13.** Times–series of water budget variables and soil temperature at Stoughton COSMOS site. Black plus signs: COSMOS-UK observations, grey crosses: SMAP observations for closest 9 km pixel, blue line: prior JULES estimate for closest 1 km grid cell, orange line: posterior JULES estimate for closest 1 km grid cell.

technique at a larger scale that the optimization is split up into different calibration zones as it has been shown that pedotransfer functions in certain regions can have a different form (e.g. tropical soils (Marthews et al., 2014)).

Within the DA procedure used to optimise the PTF parameters there are uncertainties that have not been explicitly prescribed.
There will be inherent bias and errors in both the observations and model. For SMAP any bias contained in the observations could cause us to retrieve PTF parameters that result in erroneous soil hydraulic conductivity's and ultimately degrade the performance of other model components. It has been shown that the Level-3 9 km SMAP observations used here do not have a significant bias (Colliander et al., 2017) especially in temperate regions (Zhang et al., 2019). The fact that after assimilation of the SMAP data we not only reduce the RMSE of JULES compared to SMAP but also reduce the RMSE of JULES compared to
independent COSMOS estimates also gives us confidence that the bias in the assimilated SMAP data is relatively low. We have dealt with the many errors contained within our DA cost function by inflating the observation uncertainty within the observation error covariance matrix, as described in section 2.6. However, specifying the errors arising from structural uncertainties and missing processes within the JULES model is difficult. We can see these errors manifesting themselves in our comparisons to COSMOS–UK observations in Figures 11 to 14. Figure 11 displays results for the Cardington cosmic–ray probe, this site is
a level well–managed grassland with a typical mineral soil and is therefore well modelled by JULES which has the ability to represent the processes of such a site. Both the Morley and Stoughton sensors (Figure 12 and 13 respectively) are positioned on arable land with typical mineral soils and while we model Morley well we struggle to match the magnitude of the Stoughton

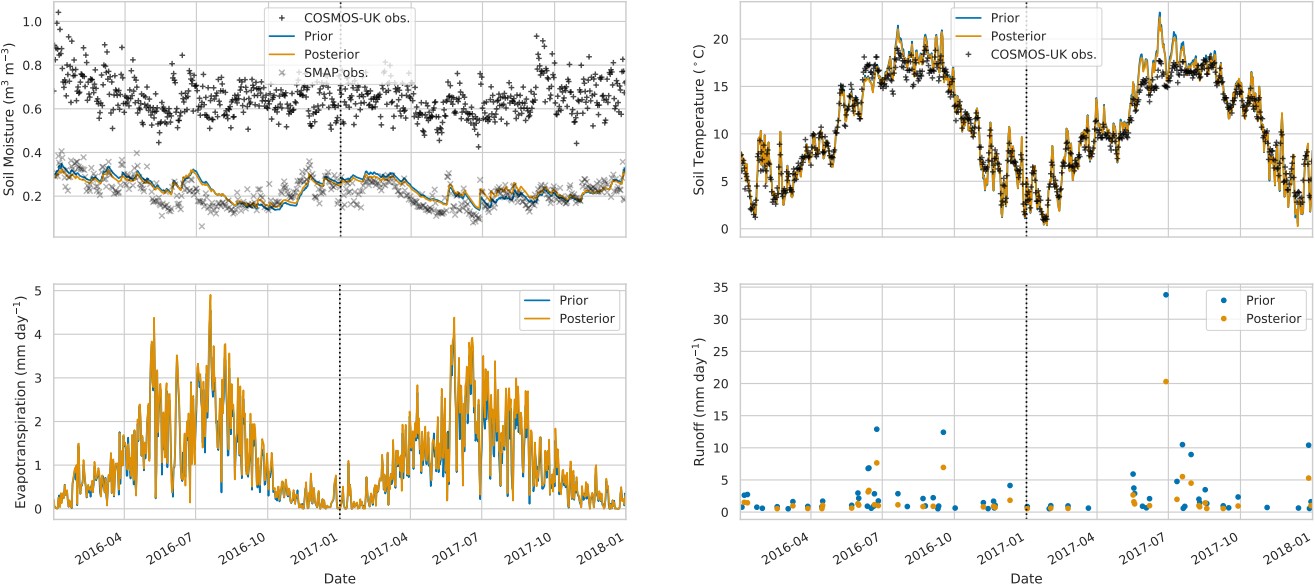

**Figure 14.** Times–series of water budget variables and soil temperature at Redmere COSMOS site. Black plus signs: COSMOS-UK observations, grey crosses: SMAP observations for closest 9 km pixel, blue line: prior JULES estimate for closest 1 km grid cell, orange line: posterior JULES estimate for closest 1 km grid cell.

observations. It is possible that different management practices at the respective sites are impacting on the ability of JULES to predict the observations. In this paper we have not run JULES with its in–built crop model turned on, so that the model will struggle to represent heavily managed crops that behave distinctly from a grassland. The site at which both prior and posterior perform worst is Redmere (Figure 14), this cosmic-ray probe is again on arable land but with a soil type of peat. In its current configuration JULES does not model organic soils and estimates of soil moisture from microwave satellite sensors over peatland are problematic (Zhang et al., 2019), so it is understandable that we are unable to match the much wetter conditions observed at this site. The accuracy of JULES posterior estimates is also contingent on the assimilated SMAP observations, so if SMAP estimates have large errors compared to cosmic-ray probe observations JULES will be unable to improve from its prior predictions.

In the initial application of this technique we have focused on a specific region at a high resolution. Here we have utilised 256 processors to run the JULES model ensemble, with each JULES run utilising message parsing interfaces to disaggregate the spatial domain of the model and split the computational load across multiple processors. In this set up it has taken approximately 1.5 days to complete 100 JULES model runs, with each model being for 30614 grid cells and over 6 years (2016 to 2017, with a 4 year spin-up). In order to find a set of pedotransfer function parameters valid at the global scale, using the technique presented here, we would need to decrease the spatial resolution. Working at the scale of 0.5 degrees we would have approximately 67000 land grid cells globally. Using our fairly modest experimental setup and assuming a linear scaling repeating at the global scale

|  | Correlation | | ubRMSE | | RMSE | |
|---|---|---|---|---|---|---|
| Site | Prior | Posterior | Prior | Posterior | Prior | Posterior |
| Bunny Park | 0.86 | 0.89 | 0.02 | 0.02 | 0.07 | 0.04 |
| Cardington | 0.85 | 0.91 | 0.05 | 0.03 | 0.06 | 0.03 |
| Elmsett | 0.81 | 0.82 | 0.04 | 0.04 | 0.16 | 0.17 |
| Euston | 0.90 | 0.92 | 0.04 | 0.04 | 0.05 | 0.04 |
| Fincham | 0.83 | 0.85 | 0.02 | 0.02 | 0.19 | 0.13 |
| Loddington | 0.45 | 0.79 | 0.06 | 0.04 | 0.39 | 0.31 |
| Morley | 0.86 | 0.89 | 0.03 | 0.03 | 0.03 | 0.03 |
| Redmere | 0.33 | 0.35 | 0.08 | 0.08 | 0.43 | 0.43 |
| Rothamsted | 0.85 | 0.89 | 0.03 | 0.03 | 0.05 | 0.07 |
| Stoughton | 0.30 | 0.76 | 0.05 | 0.04 | 0.24 | 0.13 |
| Waddesdon | 0.63 | 0.87 | 0.07 | 0.05 | 0.27 | 0.19 |
| All Sites | 0.70 | 0.81 | 0.045 | 0.038 | 0.18 | 0.14 |

**Table 3.** Summary statistics for comparison of JULES-CHESS soil moisture estimates to COSMOS probe observations over the experiment period. Over all sites we find a 16% increase in correlation, 16% reduction in ubRMSE and 22% reduction in RMSE after performing the calibration using LAVENDAR.

|  | Correlation | | ubRMSE | | RMSE (W m$^{-2}$) | |
|---|---|---|---|---|---|---|
| Site | Prior | Posterior | Prior | Posterior | Prior | Posterior |
| Cardington | 0.51 | 0.72 | 20.00 | 16.08 | 23.35 | 16.29 |
| Great Fen | 0.43 | 0.57 | 22.82 | 20.74 | 32.66 | 26.04 |
| Redmere | 0.56 | 0.71 | 20.40 | 17.38 | 32.97 | 23.48 |

**Table 4.** Summary statistics for comparison of JULES-CHESS latent heat estimates to flux tower observations over the experiment period. Over all sites we find a 34% increase in correlation, 15% reduction in ubRMSE and 26% reduction in RMSE after performing the calibration using LAVENDAR.

would still only take a little over 3 days. However, it may be beneficial to focus on regional efforts to ensure the optimised
pedotransfer functions best reflect the behaviour of local soils. The global domain could then be decomposed into sub-regions with specific parameters being found for each distinct region.

    Both SMAP and COSMOS–UK observations represent a valuable resource for validation and improvement of land surface models and could be further utilised still. It is possible that our formation of a spatially aggregated observation operator to compare SMAP 9 km estimates to JULES 1 km estimates could be improved upon and that more signal may be coming from
the centre of the satellite pixel, so that we could weight these JULES model pixels more highly within the observation operator.

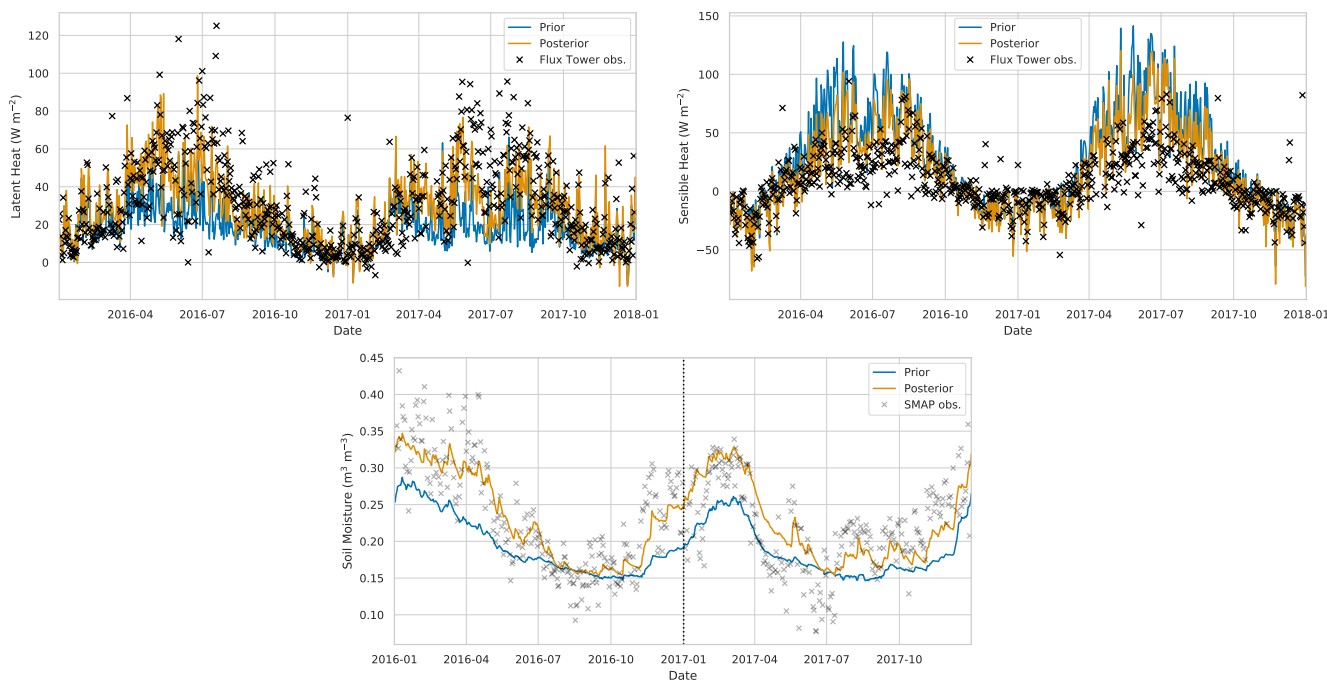

**Figure 15.** Times–series of heat flux variables and soil moisture at Cardington flux tower site. Black crosses signs: flux tower observations, grey crosses: SMAP observations for closest 9 km pixel, blue line: prior JULES estimate for closest 1 km grid cell, orange line: posterior JULES estimate for closest 1 km grid cell.

|  | Correlation | | ubRMSE | | RMSE (W m$^{-2}$) | |
| --- | --- | --- | --- | --- | --- | --- |
| Site | Prior | Posterior | Prior | Posterior | Prior | Posterior |
| Cardington | 0.80 | 0.82 | 27.10 | 22.27 | 32.18 | 23.99 |
| Great Fen | 0.72 | 0.72 | 38.40 | 34.30 | 52.50 | 44.99 |
| Redmere | 0.77 | 0.77 | 26.51 | 21.10 | 39.69 | 29.57 |

**Table 5.** Summary statistics for comparison of JULES-CHESS sensible heat estimates to flux tower observations over the experiment period. Over all sites we find a 1% increase in correlation, 16% reduction in ubRMSE and 22% reduction in RMSE after performing the calibration using LAVENDAR.

In future work it may also be beneficial to build towards a full radiative transfer scheme on top of JULES to assimilate the raw brightness temperature observations from the SMAP satellite to increase the representativty between the observations and the model and reduce sources of bias that may be introduced by the use of ancillary data in the soil moisture retrieval. Other studies utilising different land surface models have shown this works well (Han et al., 2014; Yang et al., 2016; Lievens et al., 475 2017) The COSMOS–UK observations could also be used within the data assimilation algorithm, rather than just acting as

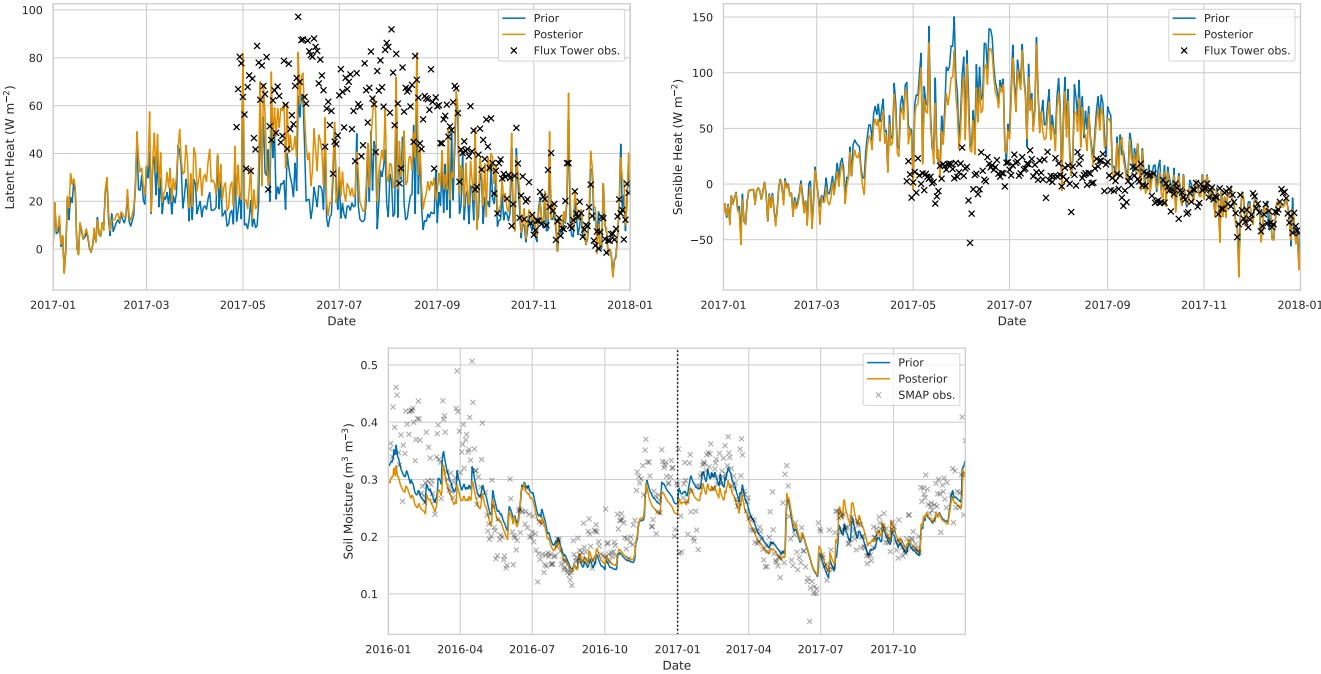

**Figure 16.** Times–series of heat flux variables and soil moisture at Great Fen flux tower site. Black crosses signs: flux tower observations, grey crosses: SMAP observations for closest 9 km pixel, blue line: prior JULES estimate for closest 1 km grid cell, orange line: posterior JULES estimate for closest 1 km grid cell.

validation, to capture information on another spatial scale. Much work would be needed here to process and organise site level driving data and understand the different characteristics of each site before combining these observations with the JULES land surface model.

In this paper we have focused on the optimisation of pedotransfer function parameters to improve estimates of water balance from land surface models. In other regions across the globe where underlying soil texture maps are highly uncertain it may be necessary to also consider optimising estimates of soil properties per-grid cell, given satellite and in-situ observations (Pinnington et al., 2018). This could further increase the skill of estimates in problematic areas. There is also the opportunity to incorporate other streams of observations into the data assimilation procedure. For example the use of stream flow data could give us a powerful integrated constraint on land surface model estimates of water balance and run-off (Abbaszadeh et al., 2020). Flux tower observations of latent and sensible heat could also provide useful constraints on assimilation outputs. Within the Hydro–JULES project work is being undertaken to improve the representation of hydrological processes at different scales, especially lateral soil water flow and groundwater. The development of the new JULES groundwater component will allow for the use of observations from the Gravity Recovery and Climate Experiment (GRACE) satellites (Tapley et al., 2004) which have the ability to monitor changes in the Earth's underground water storage. It will be informative to re–run this parameter estimation experiment again as new processes are added to the model to understand the effect on the retrieved pedotransfer

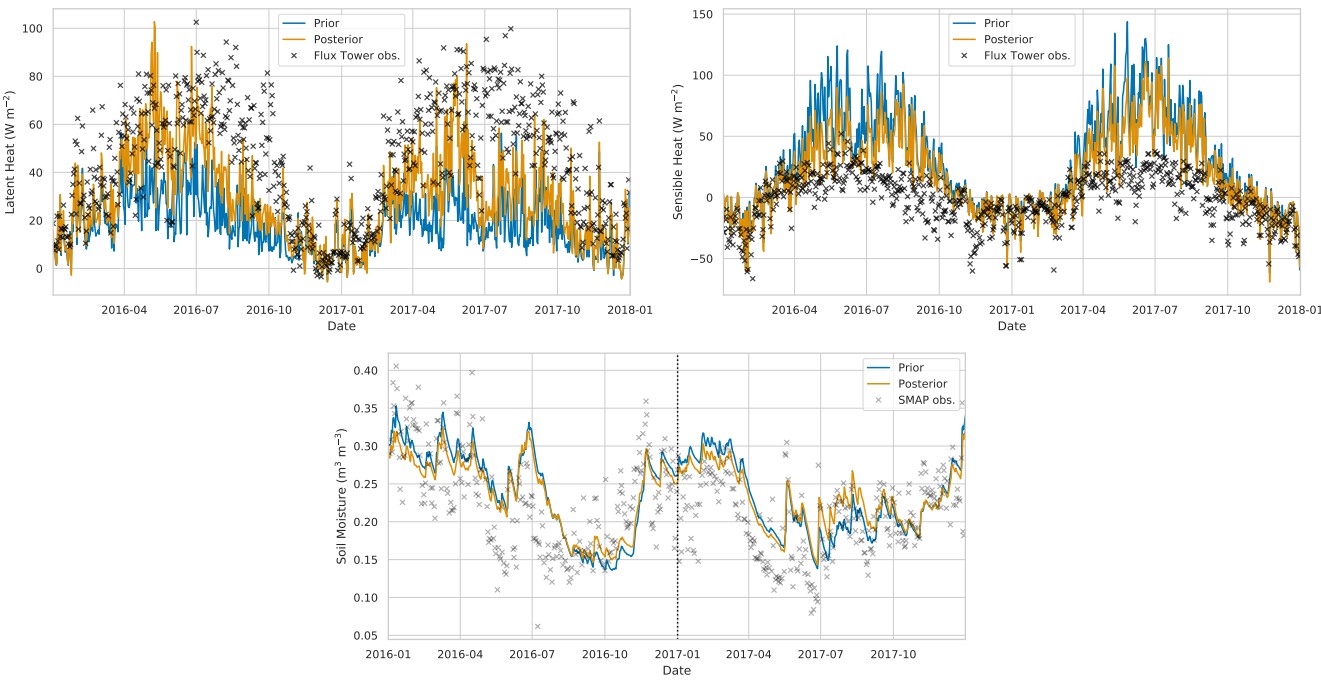

**Figure 17.** Times–series of heat flux variables and soil moisture at Redmere flux tower site. Black crosses signs: flux tower observations, grey crosses: SMAP observations for closest 9 km pixel, blue line: prior JULES estimate for closest 1 km grid cell, orange line: posterior JULES estimate for closest 1 km grid cell.

function parameters. We will then be able to see where we might be over–fitting these parameters to account for current structural deficiencies within the model (such as the current lack of a groundwater model).

## 5   Conclusions

We have presented novel methods for calibrating pedotranfer functions used to create the soil parameter ancillaries of a land surface model by using satellite data from the NASA SMAP mission. After the retrieval of an optimized parameter set, using new hybrid data assimilation techniques, we find an average 20% reduction in error for JULES model estimates of soil moisture when compared to SMAP satellite estimates. There are still areas which remain problematic such as working over urban locations and peatlands. These will require additional modelling efforts and new model components. The resultant posterior pedotransfer functions also improve the prediction of soil moisture and heat fluxes for the JULES land surface model when compared to independent in-situ estimates from the COSMOS–UK network and 3 flux tower sites. At 11 COSMOS–UK research sites distributed across the experiment domain we find an average 16% increase in correlation, 16% reduction in ubRMSE and a 22% reduction in RMSE for the posterior pedotransfer functions compared to the prior.

*Code availability.* The code used in experiments is available from the MetOffice JULES repository (https://code.metoffice.gov.uk/trac/jules) under Rose suite number u-bq357. The LAVENDAR data assimilation first release is available here: https://github.com/pyearthsci/lavendar (last access: 20 February 2019).

## Appendix A: Computing the posterior ensemble

In this appendix we summarise the process to get the analysis (or posterior) ensemble of extended state variables (variables and parameters) In the case of this paper the variables and parameters correspond to the 15 PTF parameters in Table 1. The following steps are a recapitulation and continuation of the equations in Pinnington et al. (2018).

Let us start with a background ensemble of $N_e$ joint state-parameter vectors:

$$\mathbf{X}_b = \left[ \mathbf{x}_b^1, \mathbf{x}_b^2, \ldots, \mathbf{x}_b^{N_e} \right]. \tag{A1}$$

In our experiments each $\mathbf{x}_b^i$ corresponds to a unique set of 15 PTF parameters ($\mathbf{x}_b^i = (\phi_a^i, \phi_b^i, \ldots, \phi_o^i)$) and $N_e = 50$. We can define the sample background (or prior) mean as:

$$\bar{\mathbf{x}}_b = \frac{1}{N_e} \sum_{n=1}^{N_e} \mathbf{x}_b^n \tag{A2}$$

and sample background perturbation matrix as

$$\mathbf{X}_b' = \frac{1}{\sqrt{N_e - 1}} \left[ \mathbf{x}_b^1 - \bar{\mathbf{x}}_b, \mathbf{x}_b^2 - \bar{\mathbf{x}}_b, \ldots, \mathbf{x}_b^{N_e} - \bar{\mathbf{x}}_b \right]. \tag{A3}$$

The ensemble background error covariance matrix defined by

$$\mathbf{P}_b = \mathbf{X}_b' \mathbf{X}_b'^T. \tag{A4}$$

To reduce the difficulty in finding the ensemble analysis mean, we use an incremental and pre-conditioned algorithm. Incremental means that we express the analysis mean which is a perturbation from the background mean, i.e.:

$$\bar{\mathbf{x}}_a = \bar{\mathbf{x}}_b + \delta\mathbf{x}. \tag{A5}$$

The pre-conditioned part means that the departure $\delta\mathbf{x}$ can be written by a control variable pre-multiplied by a conditioning matrix. In particular we choose the departure vector to be written as a linear combination of the background ensemble of perturbations, i.e.

$$\bar{\mathbf{x}}_a = \bar{\mathbf{x}}_b + \mathbf{X}_b' \mathbf{w}_a, \tag{A6}$$

where $\mathbf{w}_a$ is a vector of weights, which becomes the object we are solving for in the estimation process. This formulation has been used in several formulations, starting with Bishop et al (2001) and Wang et al (2004). We do not use localisation in this

work, but in the presence of localisation it would be applied in the manner of Hunt et al (2007). This vector of weights is the minimiser of a cost function which can be written in ensemble space as:

$$J(\mathbf{w}) = \frac{1}{2}\mathbf{w}^T\mathbf{w} + \frac{1}{2}(\hat{\mathbf{H}}\mathbf{X}'_b\mathbf{w} + \hat{\mathbf{h}}(\bar{\mathbf{x}}_b) - \hat{\mathbf{y}})^T\hat{\mathbf{R}}^{-1}(\hat{\mathbf{H}}\mathbf{X}'_b\mathbf{w} + \hat{\mathbf{h}}(\bar{\mathbf{x}}_b) - \hat{\mathbf{y}}) \tag{A7}$$

with gradient

$$\nabla J(\mathbf{w}) = \mathbf{w} + (\hat{\mathbf{H}}\mathbf{X}'_b)^T\hat{\mathbf{R}}^{-1}(\hat{\mathbf{H}}\mathbf{X}'_b\mathbf{w} + \hat{\mathbf{h}}(\mathbf{x}^b) - \hat{\mathbf{y}}), \tag{A8}$$

where $\hat{\mathbf{y}}$ are the observations for the whole time-window and spatial domain (here 2016 SMAP observations over the East of England, with units $\text{m}^3\ \text{m}^{-3}$), $\hat{\mathbf{H}}$ and $\hat{\mathbf{h}}$ are the linearised and non-linear observation operator respectively (here the JULES model, which includes both a time integration and conversion into observation space to match the SMAP observations) and $\hat{\mathbf{R}}$ is the observation error covariance matrix (here containing the error estimates for the assimilated SMAP observations).

In practice we do not compute the linearised version of JULES. Instead one can define statistics in the observation space in the following manner. The background ensemble of $N_e$ joint state-parameter vectors in observation space is obtained by applying the observation operator to each ensemble member:

$$\mathbf{Y}_b = \left[\mathbf{y}_b^1 = \hat{\mathbf{h}}\left(\mathbf{x}_b^1\right), \mathbf{y}_b^2 = \hat{\mathbf{h}}\left(\mathbf{x}_b^2\right), \ldots, \mathbf{y}_b^{N_e} = \hat{\mathbf{h}}\left(\mathbf{x}_b^{N_e}\right)\right] \tag{A9}$$

The sample background mean in observation space is:

$$\bar{\mathbf{y}}_b = \frac{1}{N_e}\sum_{n=1}^{N_e}\mathbf{y}_b^n \tag{A10}$$

and the sample background perturbation matrix in observation space is:

$$\mathbf{Y}'_b = \frac{1}{\sqrt{N_e - 1}}\left[\mathbf{y}_b^1 - \bar{\mathbf{y}}_b, \mathbf{y}_b^2 - \bar{\mathbf{y}}_b, \ldots, \mathbf{y}_b^{N_e} - \bar{\mathbf{y}}_b\right] \tag{A11}$$

Using these considerations, (A7) and (A8) become (approximately):

$$J(\mathbf{w}) = \frac{1}{2}\mathbf{w}^T\mathbf{w} + \frac{1}{2}(\mathbf{Y}'_b\mathbf{w} + \bar{\mathbf{y}}_b - \hat{\mathbf{y}})^T\hat{\mathbf{R}}^{-1}(\mathbf{Y}'_b\mathbf{w} + \bar{\mathbf{y}}_b - \hat{\mathbf{y}}) \tag{A12}$$

and

$$\nabla J(\mathbf{w}) = \mathbf{w} + (\mathbf{Y}'_b)^T\hat{\mathbf{R}}^{-1}(\mathbf{Y}'_b\mathbf{w} + \bar{\mathbf{y}}^b - \hat{\mathbf{y}}). \tag{A13}$$

Computing the minimum of the cost function (A12) using gradient (A13) yields the maximum-a-posteriori estimate $\mathbf{w}_a$ which inserting into equation (A6) gives us the maximum-a-posteriori estimate to the parameter and/or state variables $\mathbf{x}_a$. The analysis error covariance matrix ($\mathbf{P}_a$) is given by (Evensen, 2003):

$$\mathbf{A} = (\mathbf{I} - \mathbf{K}\hat{\mathbf{H}})\mathbf{P}_b \implies \mathbf{X}'_a\mathbf{X}'^T_a = (\mathbf{I} - \mathbf{K}\hat{\mathbf{H}})\mathbf{X}'_b\mathbf{X}'^T_b \tag{A14}$$

where $\mathbf{K}$ is the Kalman gain matrix and

$$(\mathbf{I} - \mathbf{K}\hat{\mathbf{H}}) = (\mathbf{I} + \hat{\mathbf{H}}\mathbf{X}'^T_b\hat{\mathbf{R}}^{-1}\hat{\mathbf{H}}\mathbf{X}_b)'^{-1} \approx (\mathbf{I} + \mathbf{Y}'^T_b\hat{\mathbf{R}}^{-1}\mathbf{Y}'_b)^{-1}. \tag{A15}$$

Then

$$\mathbf{X}'_a \mathbf{X}'^T_a = \mathbf{X}'_b (\mathbf{I} - \mathbf{K}\hat{\mathbf{H}})\mathbf{X}'^T_b \implies \mathbf{X}'_a = \mathbf{X}'_b (\mathbf{I} + \mathbf{Y}'^T_b \hat{\mathbf{R}}^{-1} \mathbf{Y}'_b)^{-\frac{1}{2}} \tag{A16}$$

i.e. the analysis ensemble of perturbations can be obtained by a right multiplication of the background ensemble of perturbations times a matrix of weights defined as:

$$\mathbf{W}_a = (\mathbf{I} + \mathbf{Y}'^T_b \hat{\mathbf{R}}^{-1} \mathbf{Y}'_b)^{-\frac{1}{2}}. \tag{A17}$$

In our case the matrix square root is computed via Cholesky decomposition. Finally the posterior ensemble of $N_e$ parameter/state vectors $(\mathbf{X}_a)$ is constructed as

$$\mathbf{X}_a = \left[\mathbf{x}^a + \mathbf{X}'_{a,1}, \mathbf{x}^a + \mathbf{X}'_{a,2}, \ldots, \mathbf{x}^a + \mathbf{X}'_{a,N_e}\right]. \tag{A18}$$

This posterior parameter ensemble and corresponding set of JULES runs can then be used to provide uncertainty estimates on our posterior model predictions and can also be used in future calibration studies or as an ensemble forecast for state estimation.

*Author contributions.* EP designed the data assimilation system and conducted all experiments with input from all co-authors. JA contributed to the underlying mathematical framework. EC wrote the algorithm to compare JULES soil moisture to cosmic-ray probe measurements. ER processed the HWSD and built the initial system for relating soil textural information to the parameters of JULES used in these experiments. EP prepared the manuscript with input from all co-authors.

*Competing interests.* No competing interests present.

*Acknowledgements.* This work was funded by the UK Natural Environment Research Council's Hydro–JULES project (NE/S017380/1). TQ and JA contribution was funded via the UK National Centre for Earth Observation (NCEO) at University of Reading (NCEO grant number: nceo020004). The authors gratefully acknowledge the provision by UKCEH of hydrometeorological and soil data collected by the COSMOS-UK project.

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
