# Peer review of "Improving Soil Moisture Prediction of a High–Resolution Land Surface Model by Parameterising Pedotransfer Functions through Assimilation of SMAP Satellite Data"

_Hydrology and Earth System Sciences, 2020_

## Referee Comment (RC1) · Anonymous Referee #1 · 25 Aug 2020

The study describes results of a data assimilation experiment, assimilating soil moisture data of the Soil Moisture Active Passive mission into the UK land surface model JULES. The assimilation updates states and parameters. Resulting soil moisture is compared to SMAP data and data of an independent network of cosmic ray neutron probes.

The title and general content of the manuscript are promising, while the manuscript itself exhibits lack of detail which would be required for following the study and reproducing the results. Below, my concerns, starting with the general ones, and followed

by detailed comments.

1. Well known bias in the SMAP satellite product and impact on pedotransfer functions is not discussed (e.g. Reichle et al. https://doi.org/10.1029/2019MS001729 or Colliander et al 2017 https://doi.org/10.1016/j.rse.2017.01.021 ). This would be a key asset of the paper.

2. Which SMAP level data was used. It will help the reader in understanding the results. Please point this out in the introduction and methods sections. What are the implications?

3. Discussion is not based on literature but merely on own postulations. A good guide is located here: https://www.biosciencewriters.com/How-to-Write-a-Strong-Discussion-in-Scientific-Manuscripts.aspx

4. Please add conceptual details on how the 4DEnVar (an optimization method) is combined with EnKF (optimization) (see page 7 lines 159-164). I imagine this can be done by text or together with a figure. Also address why are both optimization methods combined at all?

5. Please add how is the state vector in Appendix A is composed in the present case (variables, parameters, IenKS posterior?) and which units do the variables in Appendix A have.

6. Please clarify, what are prior and posterior with respect to two data assimilation methods? How can posteriors be worse then priors considering that the results are optimized using the evaluation data? Please plot as well the data assimilation performance over time with regard to RMSE and parameter convergence as for example in Poterjoy et al. 2017 https://doi.org/10.1175/MWR-D-16-0298.1 , Botto et al. 2018 https://doi.org/10.5194/hess-22-4251-2018 and Baatz et al. https://doi.org/10.5194/hess-21-2509-2017 .

7. Please add results after the 4DEnVar assimilation in order to demonstrate what an
additional assimilation yields in terms of skill.

8. Please expand on why to add another 1% SWC error to SMAP (from 0.04 to 0.05 cm3/cm3, page 6 line 123) and multiply by four (20cm3/cm3 error?) for observation inflation, a rather seldomly used method. Inflation is rather used for covariance inflation during the run time of the data assimilation experiment (e.g. Jamal and Linker 2020, https://doi.org/10.1002/vzj2.20000 or Whitaker et al. 2011 DOI: 10.1175/MWR-D-11-00276.1). Please cite more studies were observation inflation is directly used and discuss why a bias aware data assimilation method was not used (e.g. Ridler et al. 2018 https://doi.org/10.2166/nh.2017.117)

9. Please add legend to the graphs (Figure 6, 7 etc.).

10. Please discuss cross-correlation among the parameters of pedotransfer functions. From Equation 1 in the author's paper, it is clear that many parameters cross-correlate. Take for example Phi a and Phi c crosscorrelate strongly. What is the impact on saturated soil hydraulic conductivity?

11. Please expand on the JULES hydrologic water components (ET, ground water, surface water flow, overland flow, infiltration, snow). How exactly was the 4 year spin up done? Was it done in ensemble mode? How were parameters perturbed? Please provide groundwater and soil moisture development over time at four cosmic ray neutron probe locations during the spinup period to elucidate the reader about the spinup performance.

12. In this realm, a discussion of main characteristics, limitations and specifics of the study area with regard to SMAP data is essential to understand the manuscript. This would include addressing topography, land cover, other factors.

13. Equation 1 – please list the units of the parameters in these physical equations.

14. Page 7 line 145 – why did the authors chose 10% standard deviation when it is well known that many van Genuchten parameters and soil hydraulic conductivity is

logarithmic scale. What does 10% standard devation mean? Does it mean 0.63+/-0.063 for phi a and 0.0003 +/-0.00003 for phi c for example?

15. Why did the authors not use a known weighting function for JULES soil moisture to compare with cosmic ray neutron sensors. Köhli et al. 2014 https://doi.org/10.1002/2015WR017169 Baatz et al. https://doi.org/10.5194/hess-21-2509-2017 or Shuttleworth et al. 2014 doi:10.5194/hess-17-3205-2013 provide already well tested methods. How does the author's method compare with these results?

16. Aside, Desilets and Zreda, 2013 doi:10.1002/wrcr.20187, 2013 consider the diameter being 600 meter, not the radius.

17. Figure 2: please add a map of soil textures. Please discuss the sharp light blue – dark red gradient at 0.9°E. Is this an artifact from data assimilation?

18. Page 9 line 196 – adding London in all maps for the non UK citizens would be a great asset.

19. Page 10 line 206 – please define observation operator and outline the details on how this operator was developed, calibrated and validated. There are existing operators already (see point 13).

20. Page 13: Please separate discussion and outlook clearly. The authors use repeatedly phrases on future work e.g. 'work is being undertaken' (line 238), 'we will' (line 241), 'is possible' (line 244) 'could be' (line 245) 'it may' (line 247) and so on... Also references to e.g. GRACE are missing.

21. Also, a discussion on literature with previous published assimilation experiments on soil hydraulic parameters will be useful. Here, the paper can give a valuable contribution to exhisting literature. Especially considering the authors going the extra step to assimilate often cross correlated parameters of pedotransfer functions.

22. Figure 11: Symbols with a center point are more precise and clearer than circles. Please use smaller dots, or even better symbols with a center point such as +,*, x

and use different symbols for Cosmic Ray Calibration data and SMAP data points. Also please add SMAP soil moisture to the plots with cosmic ray neutron probe data, although these are not the equivalent depths as cosmic ray neutron probe soil moisture.

---

## Referee Comment (RC2) · Anonymous Referee #2 · 3 Sep 2020

Accurate soil moisture simulation has always been a tough issue due to various sources of errors, including biased forcing, unrealistic model parameters, defect model structure and/or parameterizations. Focusing on uncertainties in pedotransfer functions, this study calibrates some of the key pedotransfer parameters through the assimilation of SMAP soil moisture product, and have obtained lower RMSD and higher correlation coefficients in posteriors. Independent evaluation against COSMOS observations also suggests promising results.

In general, this work presents a good example of utilizing satellite data to improve land

surface models. The current layout and interpretation within the manuscript are mostly valid to me, except some remained concerns on the detailed DA implementations and soil moisture evaluations, as depicted below.

1. My biggest concern is on the comparison of modeled soil moisture from a relatively 'thick' layer of 0-0.1 m with SMAP retrievals, which in most conditions corresponds to only a few centimeters of the topsoil (∼2.5 cm, according to Zheng et al. 2019). Under some circumstances, soil moisture may vary a lot with depth. Is soil moisture mostly consistent and exhibits less vertical gradient within the 0-0.1m layer across the study domain? Otherwise the evaluation and the subsequent conclusions presented in this study maybe questioned. Please elaborate. Reference: Zheng, D., Li, X., Wang, X., Wang, Z., Wen, J., van der Velde, R., Schwank, M., & Su, Z. (2019). Sampling depth of L-band radiometer measurements of soil moisture and freeze-thaw dynamics on the Tibetan Plateau. Remote Sensing of Environment, 226, 16-25

2. Looks typo in the third equation of Eq(1): should âĹĚ_e f_clay be âĹĚ_f f_clay ?

3. For the pedotransfer parameters shown in Table 1, are they independently calibrated grid by grid, or they share the same values across the whole domain?

4. L138-140: it is interesting to know to which depth each COSMOS monitors soil wetness. Together with results shown in section 3, it can help understand to what extend the innovation introduced into the surface layer can propagate into deep soils. That being said, I also expect the authors to spend a short paragraph to discuss this issue.

5. L149-150: how is the observation operator like? Do you simply spatially average estimates from all the 1 km grids, and how do you project increments from the 9 km grid back to the 1 km grids? Please clarify. In addition, which variables are exactly included in the joint state-parameters?

6. L153: "…by a factor a four…"–not sure how this is done, may need to provide more

details on the implementation of inflation.

7. Fig. 3: if possible, better to show prior and posterior distributions of some of the soil hydraulic parameters (e.g. $\theta$sat,Ksat) in Eq(1) as well, as they directly regulate soil water within the land model.

8. L192: urban areas are known to have problems in both remote sensing and land surface modeled soil moisture. I would suggest excluding urban areas in all the plots in Figs.(2, 4-5). Meanwhile, the authors may want to show some of the COSMOS sites in these plots to help better interpret results in Figs. 8-11.

---

## Referee Comment (RC3) · Anonymous Referee #3 · 11 Sep 2020

The paper explores the use of SMAP soil moisture products with the JULES land surface model with a data assimilation framework. The framework is applied in a region of the UK where soil properties from pedotransfer functions are constrained with data assimilation. The topic has potential and the paper started very well with its Introduction and Methods sections. However, I found the Results and Discussion very weakly presented, without in-depth analyses and implications. It is not clear what is the lessons learned and how it can benefit the wider community. In addition, these two sections read much more like a technical report. There are many additional tests that can be

made to improve this study (I've made some suggestions). For that reason, I believe this paper manuscript requires considerable changes, hence I recommend major revisions before making my decision on its acceptance.

List of comments:

L63-64: Notice there are several approaches that constrain model parameters that do account for uncertainties, please refer to works by Keith Beven, Jim Freer, Jasper Vrugt, Grey Nearing, Hamid Moradkhani, Martyn Clark; to name a few.

L64-65: First, can the authors please point out the references for the 'Previous studies' mentioned in the sentence?

L65-66: Note that usually, the term data assimilation has been used in different ways by the atmospheric sciences and land surface modeling community in relation to the hydrological modeling community. 'Data assimilation' in general refers to using/fusing observed quantities to better constrain model components (i.e., parameter, states, etc...). Typically, the use of 'parameter estimation', 'state estimation', or 'dual parameter-state estimation' would be more clear. The reason I am mentioning this is because, although not technically a classic data assimilation application, the group by Luis Samaniego in UFZ Germany has explored similar approaches to this one using their mHM with their MPR framework. Additional work 'assimilating' both state and parameters include groups from Harrie-Jan Hendricks-Franssen, for example.

L79-81: This seems to be related to Results, not sure why it is included at the end of the Introducion section.

L94-95: The direct information obtained from SMAP is typically for the first few centimeters of soil; yet your JULES model is configured with a relatively thick initial soil layer and only 4 layers in general. Have the authors considered revising their soil layers in JULES? Have they done any simple sensitivity study to check how influential the choice of soil layer discretization is when assimilating SMAP data. If I recall correctly,
CLM (which is similar to JULES) is run with a much finer soil layer discretization.

L113-115 and Table 1: It is unclear to me how the prior is used. Don't you need an emsemble (i.e., range) for each prior factor shown in this Table? How is a single prior applied in this case?

L138-140: Can the authors be more specific about this? There are many studies that have used the COSMIC operator which is available (refer to works by Jim Shuttleworth, Rafael Rosolem, Harrie-Jan Hendricks-Franssen, as examples). Have the authors consider implementing this operator?

Section 2.6: Needs to be expanded as it is very vague and general.

Figure 3: Typically, DA are justified as an operational tool for models (in the case of state estimation). This figure here shows the Bayesian optimization approach (prior –> likelihood –> posterior) which is fine. However, I'd be interested to see the time-series of the final soil parameters (produced with the updated pedotransfer function) to check for any inconsistencies in the way a particular parameter change from time to time. I'd expect soil properties to be fairly constant (relatively to the fluxes and states in the JULES model). Also, the authors should consider checking which of the PDFs shown in the figure are expected to be significantly different. One way to do this is for example by checking whether two samples come (or not) from the same probability distribution. This can be easily done with a two-sample Kolmogorov-Smirnov test.

Figure 6: It is important to show how the prior and posterior spread compare with the actual RMSE calculated against the actual observation to check for consistencies with the DA setup. Without this analysis shown (for some points and maybe regionally), it is hard to diagnose the DA results. The goal is for the spread to have the same magnitude of the RMSE (not too large, nor too small)

Figure 4: It is not clear to me how RMSE is calculated in percentage. Maybe I missed something. Can the authors made this clear in the captions.

[Figure]

Results section: I found the results section to be presented in a very weak way. It seems to be rushed with the same regional map shown only for different metrics. The section is written almost like a technical report just going from figure to figure with very little in-depth analysis. How does the soil moisture in the region change from time to time (the metrics are only aggregated for the period)? Are the soil properties and consequently soil moisture profiles realistic? What are the impacts on other components of the model? Does 'improving' soil moisture improves other fluxes in JULES? My understanding is that COSMOS-UK also has flux data that can be used (H, LE, G???). The simple exercise of assimilating soil moisture to constrain parameters and/or states and evaluate the impact on soil moisture only does not seem to be particularly novel in my opinion (the DA frameowrk and the use of COSMOS-UK do, but should be explored further). This item is a major issue I have with the current manuscript.

Figure 10: There seems to be some systematic biases in the model that suggests non-optimal DA setup (DA requires errors to be around a zero mean). How much that impacts the results? Are there other sites with similar issues (can you expand the discussion)? Have you tried some initial pre-calibration prior to runninng the DA to reduce/remove the biases?

Discussion section: I also found the discussion a bit weak. Very little is further discussed and explored. Sometimes the discussion is mainly focused on aspects that can be done in the future. I'd suggest the authors to define 2-5 clear objectives –> questions –> hypotheses that can be presented in more detail in the Results section, and discussed more in-depth in this current section.

---

## Author Comment (AC1) · 7 Oct 2020

Reviewer 1 (R#1)

The study describes results of a data assimilation experiment, assimilating soil moisture data of the Soil Moisture Active Passive mission into the UK land surface model JULES. The assimilation updates states and parameters. Resulting soil moisture is compared to SMAP data and data of an independent network of cosmic ray neutron probes.

The title and general content of the manuscript are promising, while the manuscript itself exhibits lack of detail which would be required for following the study and reproducing the results. Below, my concerns, starting with the general ones, and followed by detailed comments.

We thank the reviewer for their comments which will undoubtedly help to strengthen this manuscript. We outline below our responses and proposed changes.

1. Well known bias in the SMAP satellite product and impact on pedotransfer functions is not discussed (e.g. Reichle et al. https://doi.org/10.1029/2019MS001729 or Colliander et al 2017 https://doi.org/10.1016/j.rse.2017.01.021 ). This would be a key asset of the paper.

This is a very good point. As per the papers mentioned, if the SMAP product is biased high there could possibly be an impact on the retrieved pedotransfer function (PTF) parameters. This would likely exhibit itself in PTF parameters that would artificially increase the values of the saturated soil moisture and possibly decrease saturated conductivity given the underlying soil textural information. The comparison to COSMOS probe data should also allow us to comment further on this and whether the bias in SMAP has had a significant impact on the posterior model skill. We will add this discussion and further analysis into the paper alongside the stated references.

2. Which SMAP level data was used. It will help the reader in understanding the results. Please point this out in the introduction and methods sections. What are the implications?

We used the L3 SMAP v3 9-km radiometer-radar combined product. We will include this and possible implications as requested.

3. Discussion is not based on literature but merely on own postulations. A good guide is located here: https://www.biosciencewriters.com/How-to-Write-a-Strong-Discussionin-Scientific-Manuscripts.aspx
We agree the discussion could be strengthened and will endeavour to do so (see later related points).

4. Please add conceptual details on how the 4DEnVar (an optimization method) is combined with EnKF (optimization) (see page 7 lines 159-164). I imagine this can be done by text or together with a figure. Also address why are both optimization methods combined at all?
5. Please add how is the state vector in Appendix A is composed in the present case (variables, parameters, IenKS posterior?) and which units do the variables in Appendix A have.
6. Please clarify, what are prior and posterior with respect to two data assimilation methods? How can posteriors be worse then priors considering that the results are optimized using the evaluation data? Please plot as well the data assimilation performance over time with regard to RMSE and parameter convergence as for example in Poterjoy et al. 2017 https://doi.org/10.1175/MWR-D-16-0298.1, Botto et al. 2018 https://doi.org/10.5194/hess-22-4251-2018 and Baatz et al. https://doi.org/10.5194/hess-21-2509-2017 .
7. Please add results after the 4DEnVar assimilation in order to demonstrate what an additional assimilation yields in terms of skill.

We have grouped together points 4-7 here as we believe these all stem from us not adequately describing the data assimilation technique used in the current manuscript. We have referenced a previous paper centred around the development of the technique and have not supplied enough information here for readers to properly understand what we have done.

4. 4DEnVar is not combined with the EnKF, 4DEnVar is a hybrid technique combining elements of both ensemble and variational data assimilation methods. This is done in practice because we want to use

a variational technique here, combining all observations over a time window with a prior prediction, to retrieve a set of optimised parameters that do not vary in time (which you would retrieve from a technique such as the EnKF or other sequential methods). However, the majority of variational techniques require the adjoint and derivative of the model code. We do not have this for JULES and it is very costly to compute. 4DEnVar, the IEnKS and other related hybrid methods allow us to approximate this adjoint from an ensemble of model runs. On reflection the way we have described this in the manuscript is not clear and we agree that the use of a diagram could be beneficial to illustrate the technique (see example diagram below). The description shall be improved and a diagram added.

[Figure]

5. In Appendix A the state vector is just the vector of 15 PTF parameters as defined in section 2.2 Table1 We will include this in the Appendix and make it clearer as to what the different variables relate too.

6. There is only a single assimilation step being used, we will aim to make this clearer. In Figures 4 to 7 the prior is just the mean and standard deviation of the 50 prior JULES ensemble members before DA and the posterior is the mean and standard deviation of the 50 posterior JULES ensemble members after DA. We are optimizing 15 PTF parameters for the whole time window (<28000) and the whole spatial domain (<30000 gridcells) in a single assimilation step by minimising a cost function. This is unlike sequential methods such as the EnKF or ETKF which step through time updating estimates at each step with available observations. We retrieve a single set of 15 PTF parameters valid over the whole domain and for the whole time period. This means that the optimisation may have to degrade the fit at certain locations to allow the 15 PTF parameters to improve the picture as a whole. This could be due to errors at these locations in driving data, the underlaying soil property map or indeed in the model structure (as is the case over urban areas in our results). As the DA method here is fundamentally different from the techniques in the papers mentioned, we are not able to reproduce the stated plots for parameter convergence as we retrieve just one set of parameters valid for the whole time window. However, we can plot the RMSE over time for both the prior and posterior ensemble members (see plot below). We will also aim to increase the distinction between previous

sequential DA methods and the variational method we have used for this paper.

[Figure]

7. We believe this is already shown and hopefully once we have strengthened the description of the DA algorithm will become more clear.

8. Please expand on why to add another 1% SWC error to SMAP (from 0.04 to 0.05 cm3/cm3, page 6 line 123) and multiply by four (20cm3/cm3 error?) for observation inflation, a rather seldomly used method. Inflation is rather used for covariance inflation during the run time of the data assimilation experiment (e.g. Jamal and Linker 2020, https://doi.org/10.1002/vzj2.20000 or Whitaker et al. 2011 DOI: 10.1175/MWR-D-11-00276.1). Please cite more studies were observation inflation is directly used and discuss why a bias aware data assimilation method was not used (e.g. Ridler et al. 2018 https://doi.org/10.2166/nh.2017.117)

Although the baseline aim for SMAP is 0.04 cm3/cm3 other studies have found higher values; 0.043 (Colliander et al 2017 https://doi.org/10.1016/j.rse.2017.01.021), 0.054 (Zhang et al. https://doi.org/10.1016/j.rse.2019.01.015), 0.054 (Li et al. https://doi.org/10.3390/rs10040535). We therefore chose a value between these studies of 0.05 as a form of expert elicitation. Although observation error inflation is seldom used in sequential data assimilation it is quite common place in variational methods (such as the one in this paper) and especially in numerical weather prediction (Wang et al. http://dx.doi.org/10.1029/2019JD031029, Bormann et al. http://dx.doi.org/10.21957/gq8j2gjp7, Fowler et al. https://doi.org/10.1002/qj.3183, Hilton et al. https://www.ecmwf.int/node/15331). The observation error inflation is required due to the fact that all observations are used at once in the assimilation whereby we minimise a cost function containing a prior term and an observational term. The greater the number of observations in the observational cost function term, the higher the weight they have in the optimization. This can lead to the prior term being completely negated and hence the retrieval of unphysical parameters. Observation error inflation would not be required if the correct specification for the observation error correlations (in space and time) and model error was included. These, however, are hard to diagnose and it has been shown that in the absence of such information inflation is required for an optimal DA system (Stewart et al. https://doi.org/10.1002/qj.2211). It has also been shown that for variational DA model errors can be included in the observational cost function term by inflating the diagonal variances, Howes et al. https://doi.org/10.1002/qj.2996. We will include further references to this in the text and strengthen the discussion around the inflation. Hopefully the improved description of the DA technique will also help here and the distinction between sequential and variational DA. Although we

agree a bias aware data assimilation could be more optimal, the one proposed is in relation to a sequential technique (the ETKF) and we are using a variational method.

9. Please add legend to the graphs (Figure 6, 7 etc.).
We will add legends as requested.

10. Please discuss cross-correlation among the parameters of pedotransfer functions. From Equation 1 in the author's paper, it is clear that many parameters cross-correlate. Take for example Phi a and Phi c crosscorrelate strongly. What is the impact on saturated soil hydraulic conductivity?
We agree added discussion on this would be beneficial. It is also possible that the inclusion of such correlations could improve data assimilation results and we have shown this in previous examples Pinnington et al. https://doi.org/10.1016/j.agrformet.2016.07.006 . As this is a first attempt at DA with pedotransfer functions this has not been investigated yet but comment on this will be added.

11. Please expand on the JULES hydrologic water components (ET, ground water, surface water flow, overland flow, infiltration, snow). How exactly was the 4 year spin up done? Was it done in ensemble mode? How were parameters perturbed? Please provide groundwater and soil moisture development over time at four cosmic ray neutron probe locations during the spinup period to elucidate the reader about the spinup performance.
We will add plots of other water budget components and details on the spin up to the text. The spin up is done for each prior and posterior ensemble member, with the parameters either being sampled from the defined prior distribution or as outputs from the DA system in the case of the posterior. The model is run from an initial value (defined by the saturated soil moisture model parameter) over the same year of forcing data to reach an equilibrium soil moisture state for any given set of parameters. The plot below shows this for three distinct ensemble members which are all defined by unique sets of PTF parameters. We can see how these unique realisations of PTF parameters define unique soil moisture trajectories. The JULES model does not contain a groundwater component in the current configuration but we will add spinup plots for soil moisture and other relevant variables to the text. We will also add plots of the other model water components (see below).

[Figure]

[Figure]

12. In this realm, a discussion of main characteristics, limitations and specifics of the study area with regard to SMAP data is essential to understand the manuscript. This would include addressing topography, land cover, other factors.
We will included a broader description as requested.

13. Equation 1 – please list the units of the parameters in these physical equations.
To be included.

14. Page 7 line 145 – why did the authors chose 10% standard deviation when it is well known that many van Genuchten parameters and soil hydraulic conductivity is logarithmic scale. What does 10% standard devation mean? Does it mean 0.63+/-0.063 for phi a and 0.0003 +/-0.00003 for phi c for example?
The reviewer is correct in their example of a 10% standard deviation, this is used to define a Gaussian distribution that 50 unique parameter sets are sampled from. It is true that van Genuchten and soil hydraulic conductivity parameters can be described by logarithmic distributions, but it is less clear what the best distributions are for the PTF parameters that are used to calculate the van Genuchten and soil hydraulic conductivity parameters. We therefore made a naïve assumption of a 10% standard deviation for our prior distribution and did not look further at this as we achieve good results when compared to in-situ COSMOS probe data. It is an important point that this is an area that could be investigated further in future studies and we will make sure this is communicated within the manuscript.

15. Why did the authors not use a known weighting function for JULES soil moisture to compare with cosmic ray neutron sensors. Köhli et al. 2014 https://doi.org/10.1002/2015WR017169 Baatz et al. https://doi.org/10.5194/hess-21-2509-2017 or Shuttleworth et al. 2014 doi:10.5194/hess-17-3205-2013 provide already well tested methods. How does the author's method compare with these results?
Apologies we did not make this clear; the COSMOS-UK network does use the method of Köhli et al. 2015 https://doi.org/10.1002/2015WR017169 to diagnose SM and relative depth of the COSMOS probe measurements. We then use a simple operator on this information and the JULES model output to compare to the COSMOS SM estimates. This operator was developed as part of the Hydro-JULES project by colleagues at UKCEH by Cooper et al. https://doi.org/10.5194/hess-2020-359. We will highlight this in the text.

16. Aside, Desilets and Zreda, 2013 doi:10.1002/wrcr.20187, 2013 consider the diameter being 600 meter, not the radius.

Noted will update.

17. Figure 2: please add a map of soil textures. Please discuss the sharp light blue – dark red gradient at 0.9E. Is this an artifact from data assimilation?
The adding of soil texture maps is a great idea and will help with interpretation of the results (we include these below also). We can see that the dark red gradient at 0.9E in Figure 2 is a result of a distinct area of soil texture in the HWSD and how this is responding to the pedotransfer functions.

[Figure]

18. Page 9 line 196 – adding London in all maps for the non UK citizens would be a great asset.
Noted, see above.

19. Page 10 line 206 – please define observation operator and outline the details on how this operator was developed, calibrated and validated. There are existing operators already (see point 15).
We have addressed this in point 15 and will ensure the operator is further defined.

20. Page 13: Please separate discussion and outlook clearly. The authors use repeatedly phrases on future work e.g. 'work is being undertaken' (line 238), 'we will' (line 241), 'is possible' (line 244) 'could be' (line 245) 'it may' (line 247) and so on. . . Also references to e.g. GRACE are missing.
Noted will split into subsections.

21. Also, a discussion on literature with previous published assimilation experiments on soil hydraulic parameters will be useful. Here, the paper can give a valuable contribution to exhisting literature. Especially considering the authors going the extra step to assimilate often cross correlated parameters of pedotransfer functions.
We agree including additional literature is important here. Also, given responses to the previous comments on how this DA method differs from previous examples, we will make sure to include the cited literature and add comment.

22. Figure 11: Symbols with a center point are more precise and clearer than circles. Please use smaller dots, or even better symbols with a center point such as +,*, x and use different symbols for Cosmic Ray Calibration data and SMAP data points. Also please add SMAP soil moisture to the plots with cosmic ray neutron probe data, although these are not the equivalent depths as cosmic ray neutron probe soil moisture.
We will update plots accordingly (see subplot included for the Cardington site above).

---

## Author Comment (AC2) · 7 Oct 2020

Reviewer 2 (R#2)

Accurate soil moisture simulation has always been a tough issue due to various sources of errors, including biased forcing, unrealistic model parameters, defect model structure and/or parameterizations. Focusing on uncertainties in pedotransfer functions, this study calibrates some of the key pedotransfer parameters through the assimilation of SMAP soil moisture product, and have obtained lower RMSD and higher correlation coefficients in posteriors. Independent evaluation against COSMOS observations also suggests promising results.
In general, this work presents a good example of utilizing satellite data to improve land surface models. The current layout and interpretation within the manuscript are mostly valid to me, except some remained concerns on the detailed DA implementations and soil moisture evaluations, as depicted below.
We thank the reviewer for their comments which will help us to improve this manuscript. We outline our responses and proposed changes below.

1. My biggest concern is on the comparison of modelled soil moisture from a relatively 'thick' layer of 0-0.1 m with SMAP retrievals, which in most conditions corresponds to only a few centimeters of the topsoil (_2.5 cm, according to Zheng et al. 2019). Under some circumstances, soil moisture may vary a lot with depth. Is soil moisture mostly consistent and exhibits less vertical gradient within the 0-0.1m layer across the study domain? Otherwise the evaluation and the subsequent conclusions presented in this study maybe questioned. Please elaborate. Reference: Zheng, D., Li, X., Wang, X.,Wang, Z., Wen, J., van der Velde, R., Schwank, M., & Su, Z. (2019). Sampling depth of L-band radiometer measurements of soil moisture and freeze-thaw dynamics on the Tibetan Plateau. Remote Sensing of Environment, 226, 16-25
We agree the comparison between SMAP and the model top 10cm could present issues due to the representative depths. However, as stated in your comment the model soil moisture does not exhibit a great deal of variability in the top 10cm as shown in the below plot where we have run JULES with a 5cm soil depth. We made that choice to use 10cm as this is the default JULES top layer soil depth and we wanted the optimized soil parameter ancillary files to be useful to the wider JULES community. To ensure the effects of this choice were minimal on the results we have re-run the experiments using a 5cm top layer in JULES. We attach plots for the retrieved parameters in both cases and can see that the optimised distributions are very similar whether a 10cm or 5cm top layer is used. We will ensure we highlight this choice as a potential source of error and discuss the referenced paper.

[Figure]

| 5cm experiment | 10cm experiment |

2. Looks typo in the third equation of Eq(1): should âˊLEˇ _e f_clay be âˊLEˇ _f f_clay ?
Noted, will correct.

3. For the pedotransfer parameters shown in Table 1, are they independently calibrated grid by grid, or they share the same values across the whole domain?
These share the same values across the whole domain. We will clarify this within the text and also strengthen description around the data assimilation technique.

4. L138-140: it is interesting to know to which depth each COSMOS monitors soil wetness. Together with results shown in section 3, it can help understand to what extend the innovation introduced into the surface layer can propagate into deep soils. That being said, I also expect the authors to spend a short paragraph to discuss this issue.
Noted, we will discuss this and include information of the COSMOS observation depth at the sites.

5. L149-150: how is the observation operator like? Do you simply spatially average estimates from all the 1 km grids, and how do you project increments from the 9 km grid back to the 1 km grids? Please clarify. In addition, which variables are exactly included in the joint state-parameters?
Yes, we spatially average the 1km model estimates to the 9km SMAP grid. The variables included in the joint state-parameter vector are just the 15 pedotransfer function (PTF) parameters, with 50 realisations of these making up the ensemble. Each realisation will also uniquely define a model trajectory of soil moisture. Unlike sequential DA techniques we solve the problem for all observations over the whole domain at once by minimising a cost function. For this method it is not necessary to project any increments back to the 1km grid as the increments we find correspond to which parameter sets allow us to best fit the data given all relevant uncertainties.

6. L153: ": : :by a factor a four: : :"–not sure how this is done, may need to provide more details on the implementation of inflation.
This is also noted by Reviewer 1 (comment number 8). We will make sure to provide more details on this and why it is necessary for the implemented DA technique.

7. Fig. 3: if possible, better to show prior and posterior distributions of some of the soil hydraulic parameters (e.g. _sat,Ksat) in Eq(1) as well, as they directly regulate soil water within the land model.
It will be difficult to show distributions of the hydraulic parameters as they vary across the domain dependent on the underlaying soil texture map. Instead we propose to show maps of the resultant soil hydraulic parameters and how these have changed after DA (see below).

[Figure]

8. L192: urban areas are known to have problems in both remote sensing and land surface modeled soil moisture. I would suggest excluding urban areas in all the plots in Figs.(2, 4-5). Meanwhile, the authors may want to show some of the COSMOS sites in these plots to help better interpret results in Figs. 8-11.

We agree including the COSMOS stations on the plots may help interpretation and will do so (see above). We may leave urban areas just as a point of discussion on current limitations.

---

## Author Comment (AC3) · 7 Oct 2020

Reviewer 3 (R#3)

The paper explores the use of SMAP soil moisture products with the JULES land surface model with a data assimilation framework. The framework is applied in a region of the UK where soil properties from pedotransfer functions are constrained with data assimilation. The topic has potential and the paper started very well with its Introduction and Methods sections. However, I found the Results and Discussion very weakly presented, without in-depth analyses and implications. It is not clear what is the lessons learned and how it can benefit the wider community. In addition, these two sections read much more like a technical report. There are many additional tests that can be made to improve this study (I've made some suggestions). For that reason, I believe this paper manuscript requires considerable changes, hence I recommend major revisions before making my decision on its acceptance.

We thank the reviewer for their comments and hope to be able to strengthen the paper in line with their specifications. Below we present our responses to comments and proposed changes.

List of comments:
L63-64: Notice there are several approaches that constrain model parameters that do account for uncertainties, please refer to works by Keith Beven, Jim Freer, Jasper Vrugt, Grey Nearing, Hamid Moradkhani, Martyn Clark; to name a few.

We agree that it is beneficial to mention such studies with relation to the catchment scale.

L64-65: First, can the authors please point out the references for the 'Previous studies' mentioned in the sentence?

Apologies, these were included in the previous sentence but missed here.

L65-66: Note that usually, the term data assimilation has been used in different ways by the atmospheric sciences and land surface modeling community in relation to the hydrological modeling community. 'Data assimilation' in general refers to using/fusing observed quantities to better constrain model components (i.e., parameter, states, etc...). Typically, the use of 'parameter estimation', 'state estimation', or 'dual parameter-state estimation' would be more clear. The reason I am mentioning this is because, although not technically a classic data assimilation application, the group by Luis Samaniego in UFZ Germany has explored similar approaches to this one using their mHM with their MPR framework. Additional work 'assimilating' both state and parameters include groups from Harrie-Jan Hendricks-Franssen, for example.

We agree. We are coming at this problem from a different background and so accept it may be helpful to update the wording here to make things more clear for the reader. We will also discuss the work and approaches you mention more in the manuscript discussion.

L79-81: This seems to be related to Results, not sure why it is included at the end of the Introducion section.

We will move this to relevant section.

L94-95: The direct information obtained from SMAP is typically for the first few centimeters of soil; yet your JULES model is configured with a relatively thick initial soil layer and only 4 layers in general. Have the authors considered revising their soil layers in JULES? Have they done any simple sensitivity study to check how influential the choice of soil layer discretization is when assimilating SMAP data. If I recall correctly, CLM (which is similar to JULES) is run with a much finer soil layer discretization.

Reviewer #2 had a similar concern (comment number 1). We chose to keep JULES in its default soil layer configuration so that the optimized soil ancillaries would be useful to the wider JULES community. We have run JULES with a 5cm top layer to confirm that the vertical variability at this depth in the model is not too great (see below). We have also re-run the entire experiment using this

soil layer to confirm that we retrieve very similar parameter distributions from the DA procedure for both the 5cm and 10cm soil depth JULES models.

[Figure]

L113-115 and Table 1: It is unclear to me how the prior is used. Don't you need an emsemble (i.e., range) for each prior factor shown in this Table? How is a single prior applied in this case?

For our prior we have 50 realisations of the parameters in Table 1. Each realisation is drawn from a normal distribution defined by the mean (shown in Table 1) and standard deviation (taken as 10% of the mean). This gives us the light grey distributions shown in the plots above. We will expand the text here to improve this description.

L138-140: Can the authors be more specific about this? There are many studies that have used the COSMIC operator which is available (refer to works by Jim Shuttleworth, Rafael Rosolem, Harrie-Jan Hendricks-Franssen, as examples). Have the authors consider implementing this operator?
Section 2.6: Needs to be expanded as it is very vague and general.

Apologies, we should have added more here and also referenced the works noted above. The COSMOS-UK network does use the method of Köhli et al. 2015 https://doi.org/10.1002/2015WR017169 to diagnose SM and relative depth of the COSMOS probe measurements. We then use a simple operator on this information and the JULES model output to compare to the COSMOS SM estimates. This operator was developed as part of the Hydro-JULES project (Our paper also forms part of this project) by colleagues at UKCEH (Cooper et al. https://doi.org/10.5194/hess-2020-359). We will highlight this in the text.

Figure 3: Typically, DA are justified as an operational tool for models (in the case of state estimation). This figure here shows the Bayesian optimization approach (prior –> likelihood –> posterior) which is fine. However, I'd be interested to see the time-series of the final soil parameters (produced with the updated pedotransfer function) to check for any inconsistencies in the way a particular parameter change from time to time. I'd expect soil properties to be fairly constant (relatively to the fluxes and states in the JULES model). Also, the authors should consider checking which of the PDFs shown in the

figure are expected to be significantly different. One way to do this is for example by checking whether two samples come (or not) from the same probability distribution. This can be easily done with a two-sample Kolmogorov-Smirnov test.

There is no optimization through time here. All the data is used at once over the whole time window and spatial domain to find optimized values for the 15 pedotransfer function (PTF) parameters valid in time and space. In this way we are avoiding the physically unrealistic artefact of time-varying parameters for the PTF's but also for the final soil hydraulic parameters. We do not believe we have made the DA approach clear enough in the paper as R#1 (comments number 4-7) also had issues in understanding what had been done, also assuming we were using a sequential assimilation method. This is a variational method and we will try to bring this out in the updated manuscript by also including a diagram (see below). Our technique is analogous to carrying out a single step of a sequential method such as the EnKF but using all information from observations and model dynamics for the whole time period and spatial domain to find a single set of PTF parameters that are valid in time and space.

[Figure]

Figure 6: It is important to show how the prior and posterior spread compare with the actual RMSE calculated against the actual observation to check for consistencies with the DA setup. Without this analysis shown (for some points and maybe regionally), it is hard to diagnose the DA results. The goal is for the spread to have the same magnitude of the RMSE (not too large, nor too small)

Figure 4: It is not clear to me how RMSE is calculated in percentage. Maybe I missed something. Can the authors made this clear in the captions.

We agree this is a useful check to make. We will include plots of the prior and posterior spread in the model predictions of soil moisture averaged in space to compare to the model RMSE averaged in space (see below). We can see from this plot that for the prior we have the desired relationship with the ensemble spread being around the same magnitude as the prior RMSE. However, for the posterior we do find an ensemble spread with a slightly lower magnitude than the posterior RMSE. This is perhaps unsurprising as we are conducting just a single assimilation step but using all <30000 observations at once in space and time, so we may find that some of the posterior parameter distributions become too narrow, as with increasing observations we increase the confidence in our posterior, thus tightening the retrieved distributions. If we were to use our posterior optimized parameters in onward experiments we would require some form of ensemble inflation.

For the spatial plots of error reduction we have just calculated the percentage change between the prior JULES prediction RMSE (compared to SMAP) and the posterior JULES prediction RMSE, both averaged in time. We will define this in the manuscript.

[Figure]

Results section: I found the results section to be presented in a very weak way. It seems to be rushed with the same regional map shown only for different metrics. The section is written almost like a technical report just going from figure to figure with very little in-depth analysis. How does the soil moisture in the region change from time to time (the metrics are only aggregated for the period)? Are the soil properties and consequently soil moisture profiles realistic? What are the impacts on other components of the model? Does 'improving' soil moisture improves other fluxes in JULES? My understanding is that COSMOS-UK also has flux data that can be used (H, LE, G???). The simple exercise of assimilating soil moisture to constrain parameters and/or states and evaluate the impact on soil moisture only does not seem to be particularly novel in my opinion (the DA frameowrk and the use of COSMOS-UK do, but should be explored further). This item is a major issue I have with the current manuscript.

We agree that the results section could be expanded to add more detail on the impact of the DA to other model components. We have had similar comments from other reviewers. We suggest adding plots of soil texture and how the resultant soil parameters change after DA. Also we agree looking at the performance of the models through time would be useful (see plot above). Unfortunately flux observations are not available from COSMOS-UK, we do have soil temperature which would allow us to judge the performance of another model component and we will include this. We will also include other water budget variables (ET, Run off) at each of the current 4 flux sites shown so that we can judge the impact of the DA on these variables (please see below for proposed plots). We believe that the ability to calibrate pedotransfer functions at a large scale using a considerable amount of satellite data (<30000 observations) in an instantaneous innovative data assimilation system does present novelty. Especially when it is shown that from this very large scale we are able to improve independent in-situ estimates from the model. However, we do still agree that including extra variables will strengthen the paper and we will include those stated above.

Soil properties:

[Figure]

Soil hydraulic parameter updates:

Impact on water budget variables:

COSMOS-UK site example (Cardington):

[Figure]

Figure 10: There seems to be some systematic biases in the model that suggests non-optimal DA setup (DA requires errors to be around a zero mean). How much that impacts the results? Are there other sites with similar issues (can you expand the discussion)? Have you tried some initial pre-calibration prior to runninng the DA to reduce/remove the biases?

The observations shown here are independent of the data assimilation. We agree it would be optimal to have errors centred around a zero mean for the assimilated SMAP observations. However, for independent in-situ validation data there will many competing errors that may make this impossible. There will be errors in the forcing meteorology (here we are using CHESS 1km forcing data and not observed in-situ meteorology), errors in the model grid and its representativity to the in-situ location, structural model errors (we currently have no ground water model in JULES and some in-situ sites may be more ground water dominated), errors in the vegetation fractions, and many more. Although at the larger scale for SMAP some pre-calibration could help we do not believe this is necessarily the case for the in-situ data. We will expand the discussion here to outline these issues.

Discussion section: I also found the discussion a bit weak. Very little is further discussed and explored. Sometimes the discussion is mainly focused on aspects that can be done in the future. I'd suggest the authors to define 2-5 clear objectives –> questions –> hypotheses that can be presented in more detail in the Results section, and discussed more in-depth in this current section.

We agree the discussion section could be improved and had similar comments from R#1. We suggested to include more literature in the discussion and will also make more comment on the include points above and new figures proposed for the results section.

Our suggested objectives are:

1) To examine the ability of 9km SMAP data to update PTF parameters in a 1km model.

2) To assess the resulting predictions of modelled soil moisture against (a) SMAP data from a different time period and (b) COSMOS probe data.

---

## Author Response (AR1)

Editor

Your manuscript "Improving Soil Moisture Prediction of a High-Resolution Land Surface Model by Parameterising Pedotransfer Functions through Assimilation of SMAP Satellite Data" has been subjected now to review by three reviewers. All of them recommend major revision. Given the many and different comments by the reviewers, the manuscript is borderline to rejection. Please consider whether it is possible to handle all reviewer comments in due time. The main points to be handled are: (i) the introduction should be strengthened with references; (ii) the manuscript lacks details in many places, for example regarding the data assimilation setup, measurement operator, handling of the vertical scale mismatch (SMAP vs model); (iii) time series of estimated soil parameters should be presented, and an evaluation for filter inbreeding (underestimation of variance); (iv) the handling of bias should at least be discussed, and additional simulations may be necessary; (v) the results and discussion sections should be improved with more scientific interpretation and discussion.

In summary, I suggest major revision and additional review of the paper. Please consider whether such a revision is feasible in the available time framework.
In your answer to the main points and detailed comments, please indicate how comments have been handled exactly, indicating also whether text has been deleted and what the position of newly included text blocks is. I am looking forward to the new version of the paper.

Dear Professor Hendricks-Franssen,

We thank you for providing a decision on our manuscript and the clarification of points which need addressing. We have made substantial changes to the manuscript in line with the reviewers comments and have conducted additional model and experiment runs to provide additional output variables and address issues of the SMAP vs model depth mismatch. We have increased the level of detail throughout the manuscript. In relation to point (iii) we are unable to provide time-series of soil parameter values since we use a smoother with a single assimilation window. This means that we consider all the available observations over the spatial domain in a time window of a given length. In our case, we chose this length to be the entire experiment time frame. This can be done since we are searching for parameters which are static in time. In retrospect we think we had not adequately described the DA technique so have strengthened this as requested and included a diagram. For point (iii) we have instead included maps of how the soil parameters change and also a plot of time-series RMSE and ensemble spread. We have discussed the handling of bias and strengthened both the results and discussions section, including additional figures and reference to other scientific literature. Thank you again for considering our manuscript and please find our detailed responses and updates below along with a marked-up version of the new manuscript.

Kind Regards,
Ewan Pinnington

Reviewer 1 (R#1)

The study describes results of a data assimilation experiment, assimilating soil moisture data of the Soil Moisture Active Passive mission into the UK land surface model JULES. The assimilation updates states and parameters. Resulting soil moisture is compared to SMAP data and data of an independent network of cosmic ray neutron probes.

The title and general content of the manuscript are promising, while the manuscript itself exhibits lack of detail which would be required for following the study and reproducing the results. Below, my concerns, starting with the general ones, and followed by detailed comments.

We thank the reviewer for their comments which will undoubtedly help to strengthen this manuscript. We outline below our responses and updates to the paper.

1. Well known bias in the SMAP satellite product and impact on pedotransfer functions is not discussed (e.g. Reichle et al. https://doi.org/10.1029/2019MS001729 or Colliander et al 2017 https://doi.org/10.1016/j.rse.2017.01.021 ). This would be a key asset of the paper.

2. Which SMAP level data was used. It will help the reader in understanding the results. Please point this out in the introduction and methods sections. What are the implications?

This is a good point. As per the paper mentioned, if the Level-4 SMAP product was used that is biased high there could possibly be an impact on the retrieved pedotransfer function (PTF) parameters. This would likely exhibit itself in PTF parameters that would artificially increase the values of the saturated soil moisture and possibly decrease saturated conductivity given the underlying soil textural information. We have included discussion of this at line 387:

"For SMAP any bias contained in the observations could cause us to retrieve PTF parameters that result in erroneous soil hydraulic conductivity's and ultimately degrade the performance of other model components. It has been shown that the Level-3 9 km SMAP observations used here do not have a significant bias (Colliander et al., 2017) especially in temperate regions (Zhang et al., 2019). The fact that after assimilation of the SMAP data we not only reduce the RMSE of JULES compared to SMAP but also reduce the RMSE of JULES compared to independent COSMOS estimates also gives us confidence that the bias in the assimilated SMAP data is relatively low."

We used the L3 SMAP v3 9-km radiometer-radar combined product. We have included this in the introduction at line 84:

"[...]high quality SMAP data (here we use Level-3 SMAP soil moisture observations) and a high distribution of COSMOS probes [..]"

We have also included more information in the methods section on the Level-3 data and bias as requested at line 146:

"For the work in this paper we use the 9 km Level-3 soil moisture product (version 3) this product has a relatively low bias (Colliander et al., 2017, Zhang et al., 2019). However, it has been shown there is a wet bias present in the Level-4 SMAP product (Reichle et al., 2017). As part of the retrieval procedure SMAP relies on some ancillary information, one example of this is soil texture where the Harmonized World Soil Database (HWSD) (Fischer et al., 2008) is used to calculate the soil dielectric constant for use within the retrieval algorithm. The use of such ancillary data in the retrieval could introduce additional biases into the SMAP soil moisture estimates that are not consistent with estimates from the land surface model we are comparing to. However, as the HWSD is also used to create the JULES soil parameter ancillary files this effect should be minimised."

3. Discussion is not based on literature but merely on own postulations. A good guide is located here: https://www.biosciencewriters.com/How-to-Write-a-Strong-Discussionin-Scientific-Manuscripts.aspx
We agree the discussion could be strengthened and have endeavoured to do so by restructuring and including more literature. See also later related points.

4. Please add conceptual details on how the 4DEnVar (an optimization method) is combined with EnKF (optimization) (see page 7 lines 159-164). I imagine this can be done by text or together with a figure. Also address why are both optimization methods combined at all?
5. Please add how is the state vector in Appendix A is composed in the present case (variables, parameters, IenKS posterior?) and which units do the variables in Appendix A have.
6. Please clarify, what are prior and posterior with respect to two data assimilation methods? How can posteriors be worse then priors considering that the results are optimized using the evaluation data? Please plot as well the data assimilation performance over time with regard to RMSE and parameter convergence as for example in Poterjoy et al. 2017 https://doi.org/10.1175/MWR-D-16-0298.1, Botto et al. 2018 https://doi.org/10.5194/hess-22-4251-2018 and Baatz et al. https://doi.org/10.5194/hess-21-2509-2017 .
7. Please add results after the 4DEnVar assimilation in order to demonstrate what an additional assimilation yields in terms of skill.
We have grouped together points 4-7 here as we believe these all stem from us not adequately describing the data assimilation technique used in the current manuscript. We have referenced a previous paper centred around the development of the technique and have not supplied enough information here for readers to properly understand what we have done.

4. 4DEnVar is not combined with the EnKF, 4DEnVar is a hybrid technique combining elements of both ensemble and variational data assimilation methods. On reflection the way we have described this in the manuscript is not clear and we believe this has caused a misunderstanding of the results. The method we have used is closer to that of the Iterative Ensemble Kalman Smoother (IEnKS). We have removed references to 4DEnVar to avoid confusion and have strengthened the description of the DA method at line 181. We agree that the use of a diagram will be beneficial to illustrate the technique (see diagram below).
"       In order to estimate the identified pedotransfer function parameters we use the LAVENDAR data assimilation framework (Pinnington et al., 2020). This framework utilises a hybrid DA technique similar to that of the Iterative Ensemble KalmanSmoother (IEnKS) (Bocquet and Sakov, 2013). A smoother is different than a filter (e.g. the Ensemble Kalman Filter (Evensen,2003)) in that it uses batches of observations which are taken over a time window of given length and the whole spatial domain, as opposed to just in a time instant. These observations are combined with the model evolution over this window and a minimization process is performed to obtain initial conditions for the state/parameter values. It is possible to run a sequence of smoother steps for successive windows, but our study only uses one year long assimilation window as the parameters we are optimising do not vary in time.
       Using a smoother instead of a filter has advantages (Lorenc and Rawlins, 2005) in that (a) more observations can be used to constrain the problem solution, and (b) information from the model evolution is implicitly used in the search process. However, using a smoother requires computing the Jacobian of the model, the so–called tangent linear model (TLM) and the related adjoint model (AM). The TLM/AM (Courtier et al., 1994). Computing and maintaining the TLM/AM is not a trivial task, and in fact we do not have this for JULES. The IEnKS solves this problem by replacing the role of the TLM/AM by 4–dimensional covariances, i.e. covariances defined over time and space. These covariances are computed as sample estimators of a given ensemble. The iterative nature of the method means that it finds the solution to the minimization problems using inner iterations rather than a single step (hence the variational nature), and this helps when the distributions of the variables/parameters of interest are not Gaussian. We provide details of the method in Appendix A.

Furthermore, to understand the variants of the ensemble Kalman Smoother and its position within the hybrid DA methods, the reader is referred to Evensen (2018).
        We show a schematic of how this system works in Figure 3, [...]"

We have also updated text in the introduction to make this more clear at line 71:
"Many previous studies optimising model soil parameters have taken a filtering DA approach (Moradkhani et al., 2005; Montzka et al., 2011;Han et al., 2014; Baatz et al., 2017; Botto et al., 2018) leading to the recovery of a time-series of parameter values as additional data is assimilated through time. In this study we use a smoother method, i.e. one that uses all observations in the spatial do-main within a time window of a given length. Then, the static parameters are obtained by a single minimization process (which can contain iterative steps). Smoothers can be used in a sequence of 'analysis windows' (as it is done in operational numerical weather prediction), but in this study we only use one of these windows since the parameters we search for do not vary in time."

[Figure]

5. In Appendix A the state vector is the vector of 15 PTF parameters as defined in section 2.2 Table1
We have included this in the Appendix at line 460:
"In the case of this paper the variables and parameters correspond to the 15 PTF parameters in Table 1. [...] In our experiments each $x^i_b$ corresponds to a unique set of 15 PTF parameters and Ne = 50. [...] where y are the observations for the whole time-window and spatial domain (here 2016 SMAP observations over the East of England, with units m3 m−3), H and h are the linearised and non-linear observation operator respectively (here the JULES model, which includes both a time integration and conversion into observation space to match the SMAP observations) and R is the observation error covariance matrix (here containing the error estimates for the assimilated SMAP observations)."

6. There is only a single assimilation step being used, we have aimed to make this clearer (see point 4 response). Using only one assimilation window is feasible because the parameters we are looking for are static in time. In Figures 4 to 7 the prior is just the mean and standard deviation of the 50 prior JULES ensemble members before DA and the posterior is the mean and standard deviation of the 50 posterior JULES ensemble members after DA. We have expanded the description of prior and posterior in the results section at line 240:
"The input to the data assimilation routine is an ensemble of 50 unique Tóth et al. (2015) PTF parameter sets drawn from a prior distribution (representing our best a priori guess to the true PTF parameters), the corresponding JULES runs (2016-2017) for each PTF parameter set and all the SMAP

observations for the year 2016 over the experiment domain. The output of the data assimilation is an ensemble of 50 optimised (posterior) PTF parameter sets, valid for the whole experiment domain and time, this allows us to calculate the posterior JULES soil ancillary files for each optimized parameter set and the corresponding posterior JULES model runs for 2016-2017."

We are optimizing 15 PTF parameters for the whole time window (<28000 observations) and the whole spatial domain (<30000 gridcells) in a single assimilation step by minimising a cost function. This is unlike sequential methods such as the EnKF or ETKF which step through time updating estimates at each step with available observations. We retrieve a single set of 15 PTF parameters valid over the whole domain and for the whole time period. This means that the optimisation may have to degrade the fit at certain locations to allow the 15 PTF parameters to improve the picture as a whole. This could be due to errors at these locations in driving data, the underlaying soil property map or indeed in the model structure (as is the case over urban areas in our results). We have included text to this effect at line 276:
"As we are minimising a cost function to find optimised values of PTF parameters valid for the whole spatial and temporal domain it is possible the optimisation may have to degrade the fit of the model estimates to the SMAP observations at certain locations in order to improve the picture as a whole. This could be due to errors at these locations in driving data, the underlying soil property map or indeed in the model structure."

As the DA method here is fundamentally different from the techniques in the papers mentioned, we are not able to reproduce the requested plots for parameter convergence as we retrieve just one set of parameters valid for the whole time window. However, we can plot the RMSE over time for both the prior and posterior ensemble members (see plot below). We have also aimed to increase the distinction between previous filtering DA methods and the variational smoother method we have used for this paper (see also response to point 4). We have included relevant text at line 296:
"In Figure 10 we show the RMSE averaged in space for the JULES model prior and posterior mean estimate, when compared to SMAP, alongside the JULES model prior and posterior ensemble spread. At all times the posterior JULES RMSE is lower than that of the prior, showing that the DA system has found a set of PTF parameters that improve the fit to the SMAP observations through time, this continues into the hindcast period (2017) when judged against observations that were not included in the DA cost function. We find slight peaks in the RMSE values throughout the time period corresponding to wetter conditions, this could be due to slight errors in the precipitation driving data used to force the model. It is optimal to have an ensemble spread that matches the magnitude of the ensemble mean RMSE and this relationship should hold given a large enough ensemble size (Houtekamer and Mitchell, 1998). We can see that this relationship holds for our prior estimates. However, after DA the posterior ensemble spread is slightly lower than that of the ensemble mean RMSE. This is perhaps unsurprising as we are conducting just a single assimilation step using all observations (over 28000) at once in space and time with a relatively small ensemble size (50). This can lead to some of the posterior parameter distributions becoming narrow, as with increasing observations we increase the confidence in our posterior, thus tightening the retrieved distributions and reducing the model ensemble spread. This result suggests that ensemble inflation (Anderson and Anderson, 1999) may be necessary if this ensemble was to be used in subsequent assimilation experiments."

[Figure]

7. We believe this is already shown and hopefully with the strengthened description of the DA algorithm this has become more clear (see response to point 4).

8. Please expand on why to add another 1% SWC error to SMAP (from 0.04 to 0.05 cm3/cm3, page 6 line 123) and multiply by four (20cm3/cm3 error?) for observation inflation, a rather seldomly used method. Inflation is rather used for covariance inflation during the run time of the data assimilation experiment (e.g. Jamal and Linker 2020, https://doi.org/10.1002/vzj2.20000 or Whitaker et al. 2011 DOI: 10.1175/MWR-D-11-00276.1). Please cite more studies were observation inflation is directly used and discuss why a bias aware data assimilation method was not used (e.g. Ridler et al. 2018 https://doi.org/10.2166/nh.2017.117)

Although the baseline aim for SMAP is 0.04 cm3/cm3 other studies have found higher values; 0.043 (Colliander et al 2017 https://doi.org/10.1016/j.rse.2017.01.021), 0.054 (Zhang et al. https://doi.org/10.1016/j.rse.2019.01.015), 0.057 (Li et al. https://doi.org/10.3390/rs10040535). We have included extra text at line 153:
"We prescribe an error of 0.05 m3 m−3 for SMAP observations in the assimilation algorithm. Although the SMAP baseline aim for error is 0.04 m3 m−3 other studies have found slightly higher values for the error in Level-3 SMAP observations (0.043 m3 m−3 (Colliander et al., 2017), 0.057 m3 m−3 (Li et al., 2018) and 0.054 m3 m−3 (Zhang et al., 2019)), we therefore chose a value between these studies."

Although observation error inflation is seldom used in sequential filtering data assimilation it is quite common place in variational methods and smoothers (such as the one in this paper), especially in numerical weather prediction (for example, Wang et al. http://dx.doi.org/10.1029/2019JD031029, Bormann et al. http://dx.doi.org/10.21957/gq8j2gjp7, Fowler et al. https://doi.org/10.1002/qj.3183, Hilton et al. https://www.ecmwf.int/node/15331). Observation error inflation is required due to the fact that all observations are used at once in the assimilation whereby we minimise a cost function containing a prior term and an observational term. The greater the number of observations in the observational cost function term, the higher the weight they have in the optimization. This can lead to the prior term being completely negated and hence the retrieval of unphysical parameters. Observation error inflation would not be required if the correct specification for the observation error correlations (in space and time) and model error was included. These, however, are hard to diagnose and it has been shown that in the absence of such information inflation is required for an optimal DA

system (Stewart et al. https://doi.org/10.1002/qj.2211). It has also been shown that for variational DA model errors can be included in the observational cost function term by inflating the diagonal variances, (Howes et al. https://doi.org/10.1002/qj.2996). We hope the improved description of the DA technique will also help here and the distinction between sequential and variational DA. Although we agree a bias aware data assimilation could be more optimal, the one proposed is in relation to a sequential technique (the ETKF) and we are using a variational method. We have added text around this at line 207:

"It has been shown that, for variational methods such as the one used in this paper, these additional sources of error (model error, representativity error, etc.) can be included in the observational term of the cost function by inflating the diagonal observation error variance (Howes et al., 2017). Although observation error inflation is rare in relation to sequential DA methods it is commonly used with variational methods and especially in numerical weather prediction (Hilton et al., 2009; Bormann et al., 2015; Minamide and Zhang, 2017; Fowler et al., 2018; Wang et al., 2019). Observation error inflation is required due to the fact that all observations are used at once in the assimilation whereby we minimise a cost function containing a prior term and an observational term. The greater the number of observations in the observational cost function term, the higher the weight they have in the optimization. This can lead to the prior term being completely negated and hence the retrieval of nonphysical parameters. Observation error inflation would not be required if the correct specification for the observation error correlations (in space and time), model error and representativity error were included. These, however, are hard to diagnose and it has been shown that in the absence of such information observation error inflation is required for an optimal DA system (Stewart et al., 2014). For this reason and due to the large number of observations assimilated in our one year assimilation window (28698) we inflate the specified observational error by a factor of four. If a filtering DA system were being used utilising a bias aware DA system such as that presented by Ridler et al. (2017) could help represent some of the additional sources of error discussed here."

9. Please add legend to the graphs (Figure 6, 7 etc.).
Legends have been added to plots as requested.

10. Please discuss cross-correlation among the parameters of pedotransfer functions. From Equation 1 in the author's paper, it is clear that many parameters cross-correlate. Take for example Phi a and Phi c crosscorrelate strongly. What is the impact on saturated soil hydraulic conductivity?
We agree added discussion on this would be beneficial. We have added discussion at line 369:
"The correlated nature of the PTF parameters in equation (1) presents a potential source of equifinality (e.g. both $\phi a$ and $\phi c$ both act to increase the magnitude of $\theta sat$ in the presence of clay soils), this means that we could achieve the same soil hydraulic conductivity with multiple realisations of PTF parameters at any individual grid cell. The effect of this is greatly reduced as we are performing the optimization over the whole domain and not on a grid cell by grid cell basis. In effect this means the unique soil properties at each of the 30614 model grid cells act as orthogonal constraints within the DA algorithm and reduce the issue of equifinailty for the optimized PTF parameters as the DA algorithm is having to fit the assimilated soil moisture observations for many different soil textures at once. It may also be possible to improve results further by including information on such correlations within our prior. Such estimates have been included in a variational DA framework for the carbon cycle and shown to improve posterior estimates (Pinnington et al., 2016)."

11. Please expand on the JULES hydrologic water components (ET, ground water, surface water flow, overland flow, infiltration, snow). How exactly was the 4 year spin up done? Was it done in ensemble mode? How were parameters perturbed? Please provide groundwater and soil moisture development over time at four cosmic ray neutron probe locations during the spinup period to elucidate the reader about the spinup performance.

The spin up is done for each prior and posterior ensemble member, with the parameters either being sampled from the defined prior distribution or as outputs from the DA system in the case of the posterior. The model is run from an initial value (defined by the saturated soil moisture model parameter) over the same year of forcing data to reach an equilibrium soil moisture state for any given set of parameters. The plot below shows this for three distinct ensemble members which are all defined by unique sets of PTF parameters. We can see how these unique realisations of PTF parameters define unique soil moisture trajectories. The JULES model does not contain a groundwater component in the current configuration but we have added a plot of the spin-up for soil moisture to the supplementary material along with increased description of the spinup technique at line 111:

"It is necessary to find an appropriate initial state before running a land surface model such as JULES and it has been shown that without a suitable spin-up period forecast skill can be impacted (Maurer and Lettenmaier, 2004). We include a 4 year spin up period at the start of each JULES run to allow the soil moisture state to reach a point of equilibrium after parameter values are changed. For the JULES spin-up the model is run from an initial value (defined by the saturated soil moisture model parameter) over the same year of forcing data, here 2015, to reach an equilibrium soil moisture state for any given set of soil hydraulic parameters. We show this model spin-up for 3 unique soil parameter sets at the same location in Figure S4."

[Figure]

We have rerun the JULES experiments outputting additional water budget variables and have included maps of how these variables change before/after data assimilation along with relevant text at line 258:

"In Figure 6 we show the difference between mean water budget variable estimates (soil moisture, evapotranspiration and runoff) in 2016 for the prior and posterior JULES model ensemble. The grid cells that are darker blue correspond to the posterior ensemble estimate being higher after assimilation and grid cells that are darker red correspond to the posterior estimate being lower. We can see that after calibration of the pedotransfer function parameters the domain has not had a uniform increment to the value of mean soil moisture, evapotranspiration or run off. This is due to the fact that soil texture specific parameters have been optimised allowing the different distinct areas of soil type defined by the HWSD (see Figure 1) to behave differently rather than having a uniform correction across the modelled area. Across the whole domain we find an average increase of 0.03 m³ m⁻³ in mean soil moisture estimates after data assimilation. We can see that in order to update PTF parameter values to find soil moisture estimates that more closely match the SMAP observations both evapotranspiration and run off model estimates have also been modified. In areas of sandy soils wetter soil moisture values have been achieved by a decrease in evapotranspiration offsetting a slight increase in runoff. In areas of high clay content wetter soil moisture values have been achieved by a larger decrease in run off compared to an increase in evapotranspiration. For silty soils we find a drier value of soil moisture for the posterior compared to the prior with a less prominent impact on evapotranspiration and run off. Figure 6 also allows us to see the high–resolution of the JULES model when run with the CHESS data, for this domain we have over 30,000 individual model grid cells."

[Figure]

For each of the COSMOS probe figures we have also included these additional water budget variable for the specific location, also including in-situ observations and model estimates to soil temperature in the top layer. We have added relevant text around this at line 314:

"In Figure 11 we show results at the Cardington COSMOS site, here we can see the posterior JULES estimate is a large improvement from the prior, although some of the driest values are still not captured. From Figure 11 we can also see that there is an increase in evapotranspiration and a decrease in runoff, this effect can also be seen from Figure 6. Figure 12 shows results for Morley COSMOS site where both prior and posterior JULES estimates perform similarly, we also have less of an update to evapotranspiration but a decrease in modelled runoff. There are also some sites where even after calibration we still do not capture the COSMOS estimates, Stoughton in Figure 13 is such an example where both prior and posterior estimates are too dry. However, here the posterior estimate is still much improved from the prior. We also find large increases in evapotranspiration and reductions in runoff for Stoughton. Figure 14 is an example where both prior and posterior perform equally poorly. The fact that the estimates and updates after DA are so different for Figures 11 - 14 despite all using the same PTF parameters highlights the effect that the underlying soil properties are having on soil hydraulic conductivity."

[Figure]

12. In this realm, a discussion of main characteristics, limitations and specifics of the study area with regard to SMAP data is essential to understand the manuscript. This would include addressing topography, land cover, other factors.
We have added this at line 159:
"The experiment area of the East of England is predominantly flat arable land which should allow for good quality SMAP retrievals, there are also coastal and urban areas where SMAP retrievals will be unreliable. This area is also prone to cloud cover which could cause gaps in the SMAP observational record."

13. Equation 1 – please list the units of the parameters in these physical equations.
These have been added.

14. Page 7 line 145 – why did the authors chose 10% standard deviation when it is well known that many van Genuchten parameters and soil hydraulic conductivity is logarithmic scale. What does 10% standard devation mean? Does it mean 0.63+/-0.063 for phi a and 0.0003 +/-0.00003 for phi c for example?

The reviewer is correct in their example of a 10% standard deviation, this is used to define a Gaussian distribution that 50 unique parameter sets are sampled from. It is true that van Genuchten and soil hydraulic conductivity parameters can be described by logarithmic distributions, but it is less clear what the best distributions are for the PTF parameters that are used to calculate the van Genuchten and soil hydraulic conductivity parameters. We therefore made a naïve assumption of a 10% standard deviation for our prior distribution and did not look further at this as we achieve good results when compared to in-situ COSMOS probe data. It is an important point that this is an area that could be investigated further in future studies. We have added text to this effect at line 200:

"Each ensemble members ancillary file is created by sampling from the normal distribution defined by mean xb and variance $(0.1 \times xb)^2$, where xb = $(\phi a, \phi b, ..., \phi o)$ with $\phi a, ..., \phi o$ taking the values given in table 1, then using each unique set of sampled parameters within equations (1) applied to the HWSD maps of soil properties (see Figure 1) for the whole domain. Although van Genuchten and hydraulic conductivity parameters can be described by logarithmic distributions it is less clear what distribution is best for the PTF parameters optimized here. We therefore made the naive assumption of a normal distribution in the first instance as this gave us good results."

15. Why did the authors not use a known weighting function for JULES soil moisture to compare with cosmic ray neutron sensors. Köhli et al. 2014 https://doi.org/10.1002/2015WR017169 Baatz et al. https://doi.org/10.5194/hess-21-2509-2017 or Shuttleworth et al. 2014 doi:10.5194/hess-17-3205-2013 provide already well tested methods. How does the author's method compare with these results?

Apologies we did not make this clear. The method of Baatz et al., 2014 is used by COSMOS-UK. The JULES operator was developed as part of the Hydro-JULES project (this paper also falls under this project) by colleagues at UKCEH (Cooper et al. https://doi.org/10.5194/hess-2020-359).  We have strengthened the description of this at line 167:

"There are many studies translating the cosmic–ray neutron intensity measured at COSMOS probe sites to soil moisture (Baatz et al., 2014; Bogena et al., 2015; Köhli et al., 2015). There have also been efforts to relate modelled soil moisture to cosmic–ray neutron intensity, such as the COsmic-ray Soil Moisture Interaction Code (COSMIC) (Shuttleworth et al., 2013;Rosolem et al., 2014). The COSMOS–UK network use the $N_0$–method described by Baatz et al. (2014) to diagnose values for the soil moisture and then the method of Köhli et al. (2015) to calculate the representative depth for each COSMOS probe measurement. To make a fair comparison between the COSMOS–UK and JULES soil moisture estimates we have constructed a simple variable depth algorithm for JULES which takes a weighted average of the different soil layers of the model given the relative depth of the COSMOS–UK observation. This is defined as […] where $\theta_D$ is the JULES modelled soil moisture at the COSMOS-UK representative depth (D) and $\theta_{10}$, $\theta_{25}$ and $\theta_{65}$ are the top, second and third layer soil moisture estimates from the JULES model."

16. Aside, Desilets and Zreda, 2013 doi:10.1002/wrcr.20187, 2013 consider the diameter being 600 meter, not the radius.

Updated.

17. Figure 2: please add a map of soil textures. Please discuss the sharp light blue – dark red gradient at 0.9E. Is this an artifact from data assimilation?

The adding of soil texture maps is a useful suggestion and will help with interpretation of the results (we include these below also). We can see that the dark red gradient at 0.9E in Figure 2 is a result of a

distinct area of soil texture in the HWSD and how this is responding to the pedotransfer functions. Extra text has been added to the manuscript (see response to point 11).

[Figure]

18. Page 9 line 196 – adding London in all maps for the non UK citizens would be a great asset.
Noted, see above.

19. Page 10 line 206 – please define observation operator and outline the details on how this operator was developed, calibrated and validated. There are existing operators already (see point 15).
We agree we did not provide sufficient detail in the initial paper here, please see response to point 15 where we have now included these details.

20. Page 13: Please separate discussion and outlook clearly. The authors use repeatedly phrases on future work e.g. 'work is being undertaken' (line 238), 'we will' (line 241), 'is possible' (line 244) 'could be' (line 245) 'it may' (line 247) and so on. . . Also references to e.g. GRACE are missing.
We have restructured the discussion and moved such statements into the final 2 paragraphs.

21. Also, a discussion on literature with previous published assimilation experiments on soil hydraulic parameters will be useful. Here, the paper can give a valuable contribution to existing literature. Especially considering the authors going the extra step to assimilate often cross correlated parameters of pedotransfer functions.
We have included more literature within the discussion and expanded comment on how optimising the PTF parameters differs from previous studies focusing on soil model parameters. Please see line 343 onwards.

22. Figure 11: Symbols with a center point are more precise and clearer than circles. Please use smaller dots, or even better symbols with a center point such as +,*, x and use different symbols for Cosmic Ray Calibration data and SMAP data points. Also please add SMAP soil moisture to the plots with cosmic ray neutron probe data, although these are not the equivalent depths as cosmic ray neutron probe soil moisture.
Plots have been updated in accordance with the Reviewers comments.

Accurate soil moisture simulation has always been a tough issue due to various sources of errors, including biased forcing, unrealistic model parameters, defect model structure and/or parameterizations. Focusing on uncertainties in pedotransfer functions, this study calibrates some of the key pedotransfer parameters through the assimilation of SMAP soil moisture product, and have obtained lower RMSD and higher correlation coefficients in posteriors. Independent evaluation against COSMOS observations also suggests promising results.

In general, this work presents a good example of utilizing satellite data to improve land surface models. The current layout and interpretation within the manuscript are mostly valid to me, except some remained concerns on the detailed DA implementations and soil moisture evaluations, as depicted below.

We thank the reviewer for their comments which will help us to improve this manuscript. We outline our responses and the changes we have made below.

1. My biggest concern is on the comparison of modelled soil moisture from a relatively 'thick' layer of 0-0.1 m with SMAP retrievals, which in most conditions corresponds to only a few centimeters of the topsoil (_2.5 cm, according to Zheng et al. 2019). Under some circumstances, soil moisture may vary a lot with depth. Is soil moisture mostly consistent and exhibits less vertical gradient within the 0-0.1m layer across the study domain? Otherwise the evaluation and the subsequent conclusions presented in this study maybe questioned. Please elaborate. Reference: Zheng, D., Li, X., Wang, X.,Wang, Z., Wen, J., van der Velde, R., Schwank, M., & Su, Z. (2019). Sampling depth of L-band radiometer measurements of soil moisture and freeze-thaw dynamics on the Tibetan Plateau. Remote Sensing of Environment, 226, 16-25

We agree the comparison between SMAP and the model top 10cm could present issues due to the representative depths. However, as stated in your comment the model soil moisture variability does not change a great deal across the top 10cm as shown in the below plot where we have run JULES with a 5cm soil depth. We made the choice to use 10cm as this is the default JULES top layer soil depth and we wanted the optimized soil parameter ancillary files to be useful to the wider JULES community. To ensure the effects of this choice were minimal on the results we have re-run the experiments using a 5cm top layer in JULES. We attach plots for the retrieved parameters in both cases and can see that the optimised distributions are very similar whether a 10cm or 5cm top layer is used. We have added text on this at line 104:

"The soil scheme is made up of 4 separate layers with depths of 0.1 m, 0.25 m, 0.65 m and 2 m respectively. We have chosen to keep JULES in its default soil-layer setup so that our optimised parameters are relevant to the wider JULES modelling community. This is despite the fact that SMAP satellite observations are typically sensitive to the top ~5cm of soil (Entekhabiet al., 2010), with some studies suggesting L-band radiometer measurements may only be sensitive to the top ~2.5cm (Zhenget al., 2019). This could introduce an additional source of error into our DA system. To ensure that the effect of this is not too great we show that there is only small difference in soil moisture between depths of 10cm and 5 cm in the JULES model in Figure S1. We have also re-run the entire data assimilation experiment with a 5 cm top soil layer in JULES and show that the recovered parameter distributions are similar to those recovered with a 10 cm top soil layer in Figure S2."

[Figure]

[Figure]

| 5cm experiment | 10cm experiment |

2. Looks typo in the third equation of Eq(1): should âˊLEˇ _e f_clay be âˊLEˇ _f f_clay ?
Corrected.

3. For the pedotransfer parameters shown in Table 1, are they independently calibrated grid by grid, or they share the same values across the whole domain?
These share the same values across the whole domain. We have clarifed this within the text at line 137, we have also strengthen description around the data assimilation technique (see Reviewer #1 point 4).
"The DA system used here optimises values for the parameters in table 1 across the whole domain rather than on a grid-by-grid basis. In this way the varied soil properties across the domain give us a form of orthogonal constraint within the assimilation and allow us to recover a single set of pedotransfer functions that are valid in space and time."

4. L138-140: it is interesting to know to which depth each COSMOS monitors soil wetness. Together with results shown in section 3, it can help understand to what extend the innovation introduced into the surface layer can propagate into deep soils. That being said, I also expect the authors to spend a short paragraph to discuss this issue.
We have added this information at line 172:
"The COSMOS sites in our experiment domain have a representative depth of between 14 cm and 40 cm dependent on conditions when measurements are made."
We have also added discussion on this in the results section at line 329:
"The COSMOS-UK observations we have used for independent validation of the results are representative of depths from 14 cm up to around 40 cm. The SMAP satellite observations, used within the assimilation algorithm to find a new set of pedotransfer functions for the experiment domain, are representative of soil moisture for the top 2.5 - 5 cm of soil. Therefore the fact that after assimilation we find such a distinct improvement at in-situ COSMOS probe locations indicates that

although the SMAP observations are only sensitive to shallow depths, by combining these with the JULES model we are also improving estimates at deeper levels."

5. L149-150: how is the observation operator like? Do you simply spatially average estimates from all the 1 km grids, and how do you project increments from the 9 km grid back to the 1 km grids? Please clarify. In addition, which variables are exactly included in the joint state-parameters?
Yes, we spatially average the 1km model estimates to the 9km SMAP grid. The variables included in the joint state-parameter vector are just the 15 pedotransfer function (PTF) parameters, with 50 realisations of these making up the ensemble. Each realisation will also uniquely define a model trajectory of soil moisture. Unlike sequential DA techniques we solve the problem for all observations over the whole domain at once by minimising a cost function. For this method it is not necessary to project any increments back to the 1km grid as the increments we find correspond to which parameter sets allow us to best fit the data given all relevant uncertainties. We have clarified this in the text at line 208:
"In order to compare the 1 km estimates of soil moisture from JULES to the 9 km SMAP estimates we create an observation operator which aggregates the JULES grid cells within each SMAP pixel by taking a spatial average of all JULES estimates which fall in the bounds of the SMAP grid cell. There is no need to project increments from the spatially averaged 9 km model estimates back to the 1 km model grid as the assimilation is only optimising the 15 PTF parameters ($\phi_a$, $\phi_b$, ..., $\phi_o$) for the whole domain and the update to soil moisture will be implicit."

6. L153: ": : :by a factor a four: : :"—not sure how this is done, may need to provide more details on the implementation of inflation.
This is also noted by Reviewer 1 (comment number 8). We have included additional text on this at line 214:
"It has been shown that, for variational methods such as the one used in this paper, these additional sources of error (model error, representativity error, etc.) can be included in the observational term of the cost function by inflating the diagonal observation error variance (Howes et al., 2017). Although observation error inflation is rare in relation to sequential DA methods it is commonly used with variational methods and especially in numerical weather prediction (Hilton et al., 2009; Bormann et al., 2015; Minamide and Zhang, 2017; Fowler et al., 2018; Wang et al., 2019). Observation error inflation is required due to the fact that all observations are used at once in the assimilation whereby we minimise a cost function containing a prior term and an observational term. The greater the number of observations in the observational cost function term, the higher the weight they have in the optimization. This can lead to the prior term being completely negated and hence the retrieval of nonphysical parameters. Observation error inflation would not be required if the correct specification for the observation error correlations (in space and time), model error and representativity error were included. These, however, are hard to diagnose and it has been shown that in the absence of such information observation error inflation is required for an optimal DA system (Stewart et al., 2014). For this reason and due to the large number of observations assimilated in our one year assimilation window (28698) we inflate the specified observational error by a factor of four. If a filtering DA system were being used utilising a bias aware DA system such as that presented by Ridler et al. (2017) could help represent some of the additional sources of error discussed here."

7. Fig. 3: if possible, better to show prior and posterior distributions of some of the soil hydraulic parameters (e.g. _sat,Ksat) in Eq(1) as well, as they directly regulate soil water within the land model.
It will be difficult to show probability distributions of the hydraulic parameters as they vary across the domain dependent on the underlaying soil texture map. Instead we have included maps of the resultant soil hydraulic parameters and how these have changed after DA (see below). Text also added at line 251:

"This can be seen in Figure 5 where we show the updated PTF parameters effect on the mean estimate to the JULES model soil parameters when applied to the spatial maps of soil properties from the HWSD. We can see how different areas of distinct soil texture (see Figure 1) behave differently based on the PTF parameter updates after DA. For some parameters we see the majority of gridcell parameter values increase or decrease, θsat and 1/(N−1) respectively. Whereas for 1/α and θcrit we see an increase or decrease in grid cell parameter values dependent on the underlying soil properties (sandier soils lead to an increase, less sand more clay correspond to a decrease)."

[Figure]

8. L192: urban areas are known to have problems in both remote sensing and land surface modeled soil moisture. I would suggest excluding urban areas in all the plots in Figs.(2, 4-5). Meanwhile, the authors may want to show some of the COSMOS sites in these plots to help better interpret results in Figs. 8-11.
We agree including the COSMOS stations on the plots may help interpretation and will do so (see above). We have left urban areas just as a point of discussion on current limitations.

Reviewer 3 (R#3)

The paper explores the use of SMAP soil moisture products with the JULES land surface model with a data assimilation framework. The framework is applied in a region of the UK where soil properties from pedotransfer functions are constrained with data assimilation. The topic has potential and the paper started very well with its Introduction and Methods sections. However, I found the Results and Discussion very weakly presented, without in-depth analyses and implications. It is not clear what is the lessons learned and how it can benefit the wider community. In addition, these two sections read much more like a technical report. There are many additional tests that can be made to improve this study (I've made some suggestions). For that reason, I believe this paper manuscript requires considerable changes, hence I recommend major revisions before making my decision on its acceptance.

We thank the reviewer for their comments and have aimed to strengthen the paper in line with their specifications. Below we present our responses to comments and outline the changes we have made.

List of comments:
L63-64: Notice there are several approaches that constrain model parameters that do account for uncertainties, please refer to works by Keith Beven, Jim Freer, Jasper Vrugt, Grey Nearing, Hamid Moradkhani, Martyn Clark; to name a few.

We agree that it is beneficial to mention such studies and have included the following at line 63: "Unlike traditional calibration procedures data assimilation and other associated Bayesian optimisation methods always take into account the relative uncertainties given to both model and observed estimates to find a maximum-a-posteriori estimate (Beven and Binley, 1992; Thiemann et al., 2001; Vrugt et al., Moradkhani et al., 2005; Nearing et al., 2010; Mizukami et al., 2017)."

L64-65: First, can the authors please point out the references for the 'Previous studies' mentioned in the sentence?

Apologies, these were missed here. We have added these on line 66: "Previous studies have used data assimilation to update the soil parameters of land surface models (Rasmy et al., 2011; Sawada and Koike, 2014; Yang et al., 2016; Han et al.,2014)"

L65-66: Note that usually, the term data assimilation has been used in different ways by the atmospheric sciences and land surface modeling community in relation to the hydrological modeling community. 'Data assimilation' in general refers to using/fusing observed quantities to better constrain model components (i.e., parameter, states, etc...). Typically, the use of 'parameter estimation', 'state estimation', or 'dual parameter-state estimation' would be more clear. The reason I am mentioning this is because, although not technically a classic data assimilation application, the group by Luis Samaniego in UFZ Germany has explored similar approaches to this one using their mHM with their MPR framework. Additional work 'assimilating' both state and parameters include groups from Harrie-Jan Hendricks-Franssen, for example.

We agree. We are coming at this problem from a different background and so accept it may be helpful to update the wording here to make things more clear for the reader. We have updated the wording to include "state estimation" and "parameter estimation" line 60: "These techniques can either be used for state estimation to update soil moisture values of the model in real-time as new observations are available [...] or for model parameter estimation to find improved calibrations which better represent the observations [...]."
We have also discussed work by Luis Samaniego in the discussion section at line 370.

L79-81: This seems to be related to Results, not sure why it is included at the end of the Introduction section.
We have removed the relevant text.

L94-95: The direct information obtained from SMAP is typically for the first few centimeters of soil; yet your JULES model is configured with a relatively thick initial soil layer and only 4 layers in general. Have the authors considered revising their soil layers in JULES? Have they done any simple sensitivity study to check how influential the choice of soil layer discretization is when assimilating SMAP data. If I recall correctly, CLM (which is similar to JULES) is run with a much finer soil layer discretization.

Reviewer #2 had a similar concern (see R#2 comment number 1). We chose to keep JULES in its default soil layer configuration so that the optimized soil ancillaries would be useful to the wider JULES community. We have run JULES with a 5cm top layer to confirm that the vertical variability at this depth in the model is not too great (see below). We have also re-run the entire experiment using this soil layer to confirm that we retrieve very similar parameter distributions from the DA procedure for both the 5cm and 10cm soil depth JULES models. Relevant text has been added at line 104: "The soil scheme is made up of 4 separate layers with depths of 0.1 m, 0.25 m, 0.65 m and 2 m respectively. We have chosen to keep JULES in its default soil-layer setup so that our optimised parameters are relevant to the wider JULES modelling community. This is despite the fact that SMAP satellite observations are typically sensitive to the top ~5cm of soil (Entekhabiet al., 2010), with some studies suggesting L-band radiometer measurements may only be sensitive to the top ~2.5cm (Zhenget al., 2019). This could introduce an additional source of error into our DA system. To ensure that the effect of this is not too great we show that there is only small difference in soil moisture between depths of 10 cm and 5 cm in the JULES model in Figure S1. We have also re-run the entire data assimilation experiment with a 5 cm top soil layer in JULES and show that the recovered parameter distributions are similar to those recovered with a 10 cm top soil layer in Figure S2."

[Figure]

| 5cm experiment | 10cm experiment |

[Figure]

L113-115 and Table 1: It is unclear to me how the prior is used. Don't you need an emsemble (i.e., range) for each prior factor shown in this Table? How is a single prior applied in this case?

For our prior we have 50 realisations of the parameters in Table 1. Each realisation is drawn from a normal distribution defined by the mean (shown in Table 1) and standard deviation (taken as 10% of

the mean). This gives us the light grey distributions shown in the plots above. We have expanded the text to this effect at line 200:

"Each ensemble members ancillary file is created by sampling from the normal distribution defined by mean xb and variance $(0.1 \times xb)^2$, where xb = ($\phi a,\phi b,...,\phi o$) with $\phi a,...,\phi o$ taking the values given in table 1, then using each unique set of sampled parameters within equations (1) applied to the HWSD maps of soil properties (see Figure 1) for the whole domain. Although van Genuchten and hydraulic conductivity parameters can be described by logarithmic distributions it is less clear what distribution is best for the PTF parameters optimized here. We therefore made the naive assumption of a normal distribution in the first instance as this gave us good results."

L138-140: Can the authors be more specific about this? There are many studies that have used the COSMIC operator which is available (refer to works by Jim Shuttleworth, Rafael Rosolem, Harrie-Jan Hendricks-Franssen, as examples). Have the authors consider implementing this operator?
Section 2.6: Needs to be expanded as it is very vague and general.
Apologies, we should have added more here and also referenced the works noted above. We have included this additional information at line 167:

"There are many studies translating the cosmic–ray neutron intensity measured at COSMOS probe sites to soil moisture (Baatz et al., 2014; Bogena et al., 2015; Köhli et al., 2015). There have also been efforts to relate modelled soil moisture to cosmic–ray neutron intensity, such as the COsmic-ray Soil Moisture Interaction Code (COSMIC) (Shuttleworth et al., 2013;Rosolem et al., 2014). The COSMOS–UK network use the $N_0$–method described by Baatz et al. (2014) to diagnose values for the soil moisture and then the method of Köhli et al. (2015) to calculate the representative depth for each COSMOS probe measurement. To make a fair comparison between the COSMOS–UK and JULES soil moisture estimates we have constructed a simple variable depth algorithm for JULES which takes a weighted average of the different soil layers of the model given the relative depth of the COSMOS–UK observation. This is defined as [...] where $\theta_D$ is the JULES modelled soil moisture at the COSMOS-UK representative depth (D) and $\theta_{10}$, $\theta_{25}$ and $\theta_{65}$ are the top, second and third layer soil moisture estimates from the JULES model."

Figure 3: Typically, DA are justified as an operational tool for models (in the case of state estimation). This figure here shows the Bayesian optimization approach (prior –> likelihood –> posterior) which is fine. However, I'd be interested to see the time-series of the final soil parameters (produced with the updated pedotransfer function) to check for any inconsistencies in the way a particular parameter change from time to time. I'd expect soil properties to be fairly constant (relatively to the fluxes and states in the JULES model). Also, the authors should consider checking which of the PDFs shown in the figure are expected to be significantly different. One way to do this is for example by checking whether two samples come (or not) from the same probability distribution. This can be easily done with a two-sample Kolmogorov-Smirnov test.
This is indeed an interesting comment and Reviewer #1 had a similar query (point #6), it is important to make the following distinction. Data assimilation can be used to determine the value of state variables and parameters. In our case we are interested in determining the value of the 15 PTF parameters which are fixed in time. We use an implementation of the Iterative Ensemble Smoother. Being a smoother, this method performs data assimilation over a time window (labelled assimilation window). The method uses all observations over the spatial domain and during the time window (here the year 2016) in a single minimisation process. This minimisation is done via an iterative routine. Hence, a single set of estimated PTF parameters are obtained. A smoother usually requires the Jacobian of the evolution model, the so-called tangent linear model (TML) and adjoint model (AM) in the 4D-variational literature. We do not have such a model for JULES. The Ensemble nature of the method allows to replace the role of the TLM/AM by operations involving 4D sample covariances, i.e. covariances defined over space and time and computed from the ensemble.

It is also important to mention that we chose the length of the assimilation window to be the whole length of the experiment. This is of course, only a choice, but we justify it given that the parameters we are looking for do not vary in time, or if they do, this is in time-scales which are considerably longer than the duration of the experiment. In numerical weather prediction, for instance, the situation is different. In such a scenario one is more interested in the actual values of the time-evolving state variables. For instance, operational centres like Met Office and ECMWF use 12-hour windows to perform data assimilation (e.g with a 4D-variational technique), and then cycle this process for subsequent windows. We have included plots of how the soil parameters change after DA when produced with the prior or posterior PTF parameters (see below). We have also included additional description of the DA method at line 181:

"        In order to estimate the identified pedotransfer function parameters we use the LAVENDAR data assimilation framework (Pinnington et al., 2020). This framework utilises a hybrid DA technique similar to that of the Iterative Ensemble KalmanSmoother (IEnKS) (Bocquet and Sakov, 2013). A smoother is different than a filter (e.g. the Ensemble Kalman Filter (Evensen,2003)) in that it uses batches of observations which are taken over a time window of given length and the whole spatial domain, as opposed to just in a time instant. These observations are combined with the model evolution over this window and a minimization process is performed to obtain initial conditions for the state/parameter values. It is possible to run a sequence of smoother steps for successive windows, but our study only uses one year long assimilation window as the parameters we are optimising do not vary in time.

        Using a smoother instead of a filter has advantages (Lorenc and Rawlins, 2005) in that (a) more observations can be used to constrain the problem solution, and (b) information from the model evolution is implicitly used in the search process. However, using a smoother requires computing the Jacobian of the model, the so–called tangent linear model (TLM) and the related adjoint model (AM). The TLM/AM (Courtier et al., 1994). Computing and maintaining the TLM/AM is not a trivial task, and in fact we do not have this for JULES. The IEnKS solves this problem by replacing the role of the TLM/AM by 4–dimensional covariances, i.e. covariances defined over time and space. These covariances are computed as sample estimators of a given ensemble. The iterative nature of the method means that it finds the solution to the minimization problems using inner iterations rather than a single step (hence the variational nature), and this helps when the distributions of the variables/parameters of interest are not Gaussian. We provide details of the method in Appendix A. Furthermore, to understand the variants of the ensemble Kalman Smoother and its position within the hybrid DA methods, the reader is referred to Evensen (2018).

        We show a schematic of how this system works in Figure 3, […]"

[Figure]

Figure 6: It is important to show how the prior and posterior spread compare with the actual RMSE calculated against the actual observation to check for consistencies with the DA setup. Without this analysis shown (for some points and maybe regionally), it is hard to diagnose the DA results. The goal is for the spread to have the same magnitude of the RMSE (not too large, nor too small)

We agree this is a useful check to make. We have added plots of the prior and posterior spread in the model predictions of soil moisture averaged in space to compare to the model RMSE averaged in space (see below, Reviewer #1 also had similar comments at point #6). We can see from this plot that for the prior we have the desired relationship with the ensemble spread being around the same magnitude as the prior RMSE. However, for the posterior we do find an ensemble spread with a slightly lower magnitude than the posterior RMSE. This is perhaps unsurprising as we are conducting just a single assimilation step but using all <28000 observations at once in space and time, so we may find that some of the posterior parameter distributions become too narrow, as with increasing observations we increase the confidence in our posterior, thus tightening the retrieved distributions. If we were to use our posterior optimized parameters in onward experiments we would require some form of ensemble inflation. We have added additional discussion on this in the results section at line 296:

"In Figure 10 we show the RMSE averaged in space for the JULES model prior and posterior mean estimate, when compared to SMAP, alongside the JULES model prior and posterior ensemble spread. At all times the posterior JULES RMSE is lower than that of the prior, showing that the DA system has

found a set of PTF parameters that improve the fit to the SMAP observations through time, this continues into the hindcast period (2017) when judged against observations that were not included in the DA cost function. We find slight peaks in the RMSE values throughout the time period corresponding to wetter conditions, this could be due to slight errors in the precipitation driving data used to force the model. It is optimal to have an ensemble spread that matches the magnitude of the ensemble mean RMSE and this relationship should hold given a large enough ensemble size (Houtekamer and Mitchell, 1998). We can see that this relationship holds for our prior estimates. However, after DA our posterior ensemble spread is slightly lower than that of the ensemble mean RMSE. This is perhaps unsurprising as we are conducting just a single assimilation step using all observations (over 28000) at once in space and time with a relatively small ensemble size (50). This can lead to some of the posterior parameter distributions becoming narrow, as with increasing observations we increase the confidence in our posterior, thus tightening the retrieved distributions and reducing the model ensemble spread. This result suggests that ensemble inflation (Anderson and Anderson, 1999) may be necessary if this ensemble was to be used in subsequent assimilation experiments."

[Figure]

Figure 4: It is not clear to me how RMSE is calculated in percentage. Maybe I missed something. Can the authors made this clear in the captions.
For the spatial plots of error reduction we have just calculated the percentage change between the prior JULES prediction RMSE (compared to SMAP) and the posterior JULES prediction RMSE, both averaged in time. We have defined this as an equation in the manuscript at line 274.

Results section: I found the results section to be presented in a very weak way. It seems to be rushed with the same regional map shown only for different metrics. The section is written almost like a technical report just going from figure to figure with very little in-depth analysis. How does the soil moisture in the region change from time to time (the metrics are only aggregated for the period)? Are the soil properties and consequently soil moisture profiles realistic? What are the impacts on other components of the model? Does 'improving' soil moisture improves other fluxes in JULES? My understanding is that COSMOS-UK also has flux data that can be used (H, LE, G???). The simple exercise of assimilating soil moisture to constrain parameters and/or states and evaluate the impact on soil moisture only does not seem to be particularly novel in my opinion (the DA frameowrk and the use of COSMOS-UK do, but should be explored further). This item is a major issue I have with the current manuscript.

We agree that the results section could be expanded to add more detail on the impact of the DA to other model components. We have had similar comments from other reviewers. We have added plots of soil texture and how the resultant soil parameters change after DA. Also we agree looking at the performance of the models through time would be useful and have included a plot of RMSE through time over the modelled domain (see plot above). Unfortunately flux observations are not routinely available from COSMOS-UK, we do have access to soil temperature observations which we have included to judge the performance of another model component. We have also included other water budget variables (ET, Run off) for the domain and at each of the current 4 COSMOS sites shown so that we can judge the impact of the DA on these variables (please see below for the included plots). We believe that the ability to calibrate pedotransfer functions at a large scale using a considerable amount of satellite data (<28000 observations) in an innovative data assimilation system does present novelty. Especially when it is shown that from this very large scale we are able to improve independent in-situ estimates from the model. However, we do agree that including extra variables will strengthen the paper and have included those stated above along with a strengthening of the analysis text in the Results section (see line 240 for start of results section).

Soil properties:

[Figure]

Impact on water budget variables:

[Figure]

COSMOS-UK site example (Cardington):

[Figure]

Figure 10: There seems to be some systematic biases in the model that suggests non-optimal DA setup (DA requires errors to be around a zero mean). How much that impacts the results? Are there other sites with similar issues (can you expand the discussion)? Have you tried some initial pre-calibration prior to running the DA to reduce/remove the biases?

We have added text addressing this at line 334:
"The large errors in our prior JULES estimates for the COSMOS sites in Figure 13 and 14 could point towards some systematic bias within the model. However, it is important to note that the COSMOS-UK observations are independent of the data assimilation. For the assimilated SMAP observations it may be optimal to have errors centred around zero but for the independent in-situ validation data there will be many competing errors that may make this impossible. There will be errors in the forcing meteorology (here we are using CHESS 1 km forcing data and not observed in-situ meteorology), errors in the model grid and its representativity to the in-situ location, structural model errors (we currently have no ground water model in JULES and some in-situ sites may be more ground water dominated), errors in the vegetation fractions, and many more. At the larger SMAP scale many of these effects will be minimised when looking at the 9 km spatial scale that is more representative of modelled estimates."

Discussion section: I also found the discussion a bit weak. Very little is further discussed and explored. Sometimes the discussion is mainly focused on aspects that can be done in the future. I'd suggest the authors to define 2-5 clear objectives –> questions –> hypotheses that can be presented in more detail in the Results section, and discussed more in-depth in this current section.

We agree the discussion section could be improved and had similar comments from R#1. We have expanded the discussion section and included more literature (see line 343 onwards for discussion section).

At the end of the introduction we have also included the 2 main objectives of this study, line 86:

[revised manuscript text omitted]

---

## Author Response (AR2)

Editor

Your manuscript "Improving Soil Moisture Prediction of a High-Resolution Land Surface Model by Parameterising Pedotransfer Functions through Assimilation of SMAP Satellite Data" has been subjected now to re-review by the original three reviewers. They recommend now moderate revision, minor revision and acceptance of the paper. Apart from handling the minor comments, the major point raised by reviewer #3 is important. It implies that additional verification of the estimates is needed, for example evaluating the impact of soil moisture assimilation on evaporative fluxes. If the authors cannot give such a verification, a motivation should be given why this is not done. Clearly, further independent validation would be of interest. For example, in several studies it was found that soil moisture assimilation hardly improves the characterization of ET. It would be important to document such a finding which allows for further research actions in the future. I recommend therefore moderate revision.

Looking forward to the revised version of your manuscript.

Dear Professor Hendricks-Franssen,

We thank you for providing a decision on our manuscript and the clarification of final points which need addressing. We have managed to acquire access to observations from 3 flux tower sites with which to assess the performance of both latent and sensible heat flux for the experiment, 2 new sections have been added to the paper comprising approximately 40 new lines of text, 3 new figures and 2 new tables. We have also added two additional authors to the manuscript who were responsible for providing the flux tower observations. From the flux tower sites we see an improvement in the modelled heat fluxes after the data assimilation procedure is conducted, giving us further confidence in the results. We thank you again for your handling of this manuscript and hope that this will satisfy the final comments.

Kind Regards,
Ewan Pinnington

**Reviewer 1 (R#1)**

The authors have significantly improved the manuscript to the first submission. The method and results have been well described and meet high scientific standards. Great to see the well-developed section 2.3 SMAP Observations and the Section 2.5 Data Assimilation Framework. Both sections are clarify the questions about methodology. Same holds for the discussion section.
We thank the reviewer for their comments that have helped to polish the final few things in the manuscript.
Only few minor issues /suggestions:
Abstract: Split the last sentence into two.
Line 87: Replace JULES model with 'domain'. Reason: The number of grid points does not depend on the model but on the resolution and the domain.
Corrected.
Line 87: Replace the two points with a single paragraph and three sentences. An enumeration of two points is not worth an enumeration. e.g. 'We defined two objectives. At first...'
Corrected.
Line 119: replace 'van Genuchten soil model' with van Genuchten soil parameters, 'parameterizations scheme', PTF, or something alike. Look for further appearances (e.g. line

120/121). In Line 130, 'soil parameters' was used, according to this suggestion. This is more specific and much clearer to 'soil model'.

Corrected.

Line 134: capital T for 'Table 1'.

Corrected.

Figure 3:Please, clarify the meaning of 5 blue lines. The reader expects 50 lines or an indication of spread. Consider using shading, dashed lines, dotted lines, etc. The messages of the figure are clear, though and the figure is well placed along the methodology section.

We have updated the captions in this figure to represent *N* ensemble members and have indicated that, although 5 are shown in the Figure, the experiments use 50 ensemble members.

Line 345/346: Two grammar mistakes in one sentence. Although these are the only ones I spotted, consider running the script through a grammar check. 'pedotranfer', 'the the'

Corrected.

Line 426: 'have have'

Corrected.

Congratulations to this manuscript.

Thank you again!

**Reviewer 3 (R#3)**

I have read the revised version of the MS entitled "Improving Soil Moisture Prediction of a High–Resolution Land Surface Model by Parameterising Pedotransfer Functions through Assimilation of SMAP Satellite Data" by Pinnington and others. I can say that the authors have made a significant effort to improve the MS and to address all reviewers questions. This has resulted in a significantly better manuscript. However, I do still have concerns about the study, especially its lack of validation and overall impact on JULES predictability of other hydrometeorological variables (explained below). For that reason, I recommend moderate revisions for the paper either by introducing additional experiments/analysis or by carefully and clearly explaining why the decision for not to do so with an explanation for the possible consequences of such changes in the model performance.

We thank the reviewer for their additional comments which have helped us further improve the manuscript. We agree that including some additional validation will help to improve the strength of the manuscript. We have therefore managed to gain access to observations from three flux tower sites located within the experiment domain for comparison to the modelled estimates. We find improvements vs these flux tower observations for sensible and latent heat fluxes. We have added these results to Section 3.3 of the manuscript along with three additional figures (Figure 15-17) and 2 additional tables (Table 4 & 5).

My main point relates to the fact that SMAP data is ingested into the JULES-DA system to improve the estimate of static soil properties as well as then soil moisture dynamics in a region. The authors have shown for some few selected grid-points, a comparison against independent soil moisture from the COSMOS-UK network. However, it is unclear with that information, how one should expect the region to be improved or not. For example:

1. How do we know the updated regional maps of soil properties are realistic? Could the authors compare with independent maps (e.g., Cranfield soil database in the UK, or using the global SoilsGrid database) at least as a baseline?

Whilst we agree that further validation of the retrieved soil properties would be beneficial, we do not believe that comparison to the other datasets mentioned would achieve this. The

technique outlined here updates the parameters of a pedotransfer function. This optimised pedotransfer function is the applied to the Harmonized World Soil Database (HWSD) for the experiment domain. As the soil properties themselves are not updated within the experiment comparing results to the Cranfield or SoilGrids database will only tell us about the differences between these soil property datasets and not the performance of the outlined technique.

2. How do we know the claimed improvements in the soil moisture will ultimately result in improvements to other hydrologically-relevant variables in JULES? My understanding is that a good number of COSMOS-UK stations provide at least sensible heat flux estimates and evapotranspiration can be estimated as a residual given the other surface energy balance components are also provided. Yes, there are some uncertainties in such approach but for, say, daily estimates this should give the authors more/less confidence in their experiment. Additionally, the authors could have also run their experiments trying to estimate streamflow at particular catchments and compare their results against UK's NRFA streamflow data, although this can be a bit more complicated if the region were characterized by groundwater dominated catchments and I believe the operational JULES version doesn't yet include such parameterizations.

As stated above we have now included comparison to the observations from three flux tower sites within the experiment domain. We believe that the observations from these tower sites will be more accurate than those estimated using the residual method at the COSMOS probe locations. At all three sites we find some improvement in the modelled sensible and latent heat fluxes, with the largest improvements corresponding to the site where the data assimilation is having the greatest impact on the model soil moisture trajectory. The reviewer is correct that the current JULES version does not include a ground water parameterisation, the water ways of East Anglia are also quite heavily managed so may impede such a comparison to streamflow data. We therefore leave evaluation of the effect on streamflow for further work, which we are beginning to pursue at present.

Minor comments:
With regards to my question about the thickness of JULES layers against shallower SMAP estimates, I appreciate that the results from the 5-cm thick experiments were similar to the original 10-cm thick simulations. I am fine to accept that. However, the additional explanation that 10-cm thickness was kept to be consistent for the wider community is very weak in my opinion. The increase in horizontal resolution should be thought simultaneously with its vertical resolution, and old parameterizations or model structures need to be frequently "challenged" otherwise understanding of model structures and their limitations are hindered. We appreciate the reviewer's comments here and will keep it in mind in the future to challenge the status-quo! Thank you again.